# 8-Hydroxyquinoline Glycoconjugates: Modifications in the Linker Structure and Their Effect on the Cytotoxicity of the Obtained Compounds

**DOI:** 10.3390/molecules24224181

**Published:** 2019-11-18

**Authors:** Monika Krawczyk, Gabriela Pastuch-Gawołek, Aleksandra Pluta, Karol Erfurt, Adrian Domiński, Piotr Kurcok

**Affiliations:** 1Department of Organic Chemistry, Bioorganic Chemistry and Biotechnology, Silesian University of Technology, B. Krzywoustego 4, 44-100 Gliwice, Poland; gabriela.pastuch@polsl.pl (G.P.-G.); aleksandrapluta93@o2.pl (A.P.); 2Biotechnology Centre, Silesian University of Technology, B. Krzywoustego 8, 44-100 Gliwice, Poland; 3Department of Chemical Organic Technology and Petrochemistry, Silesian University of Technology, B. Krzywoustego 4, 44-100 Gliwice, Poland; karol.erfurt@polsl.pl; 4Centre of Polymer and Carbon Materials, Polish Academy of Sciences, M. Curie-Sklodowskiej 34, 41-819 Zabrze, Poland; adrian.dominski@cmpw-pan.edu.pl (A.D.); piotr.kurcok@cmpw-pan.edu.pl (P.K.)

**Keywords:** quinoline, glycoconjugates, click reaction, 1,3-dipolar cycloaddition, chelators, anticancer properties

## Abstract

Small molecule nitrogen heterocycles are very important structures, widely used in the design of potential pharmaceuticals. Particularly, derivatives of 8-hydroxyquinoline (8-HQ) are successfully used to design promising anti-cancer agents. Conjugating 8-HQ derivatives with sugar derivatives, molecules with better bioavailability, selectivity, and solubility are obtained. In this study, 8-HQ derivatives were functionalized at the 8-OH position and connected with sugar derivatives (D-glucose or D-galactose) substituted with different groups at the anomeric position, using copper(I)-catalyzed 1,3-dipolar azide-alkyne cycloaddition (CuAAC). Glycoconjugates were tested for inhibition of the proliferation of cancer cell lines (HCT 116 and MCF-7) and inhibition of β-1,4-galactosyltransferase activity, which overexpression is associated with cancer progression. All glycoconjugates in protected form have a cytotoxic effect on cancer cells in the tested concentration range. The presence of additional amide groups in the linker structure improves the activity of glycoconjugates, probably due to the ability to chelate metal ions present in many types of cancers. The study of metal complexing properties confirmed that the obtained glycoconjugates are capable of chelating copper ions, which increases their anti-cancer potential.

## 1. Introduction

Cancer is one of the biggest problems in modern medicine and one of the main causes of death in the world. The high toxicity of drugs and the growing resistance of cancer cells to a significant number of pharmaceuticals increasingly limit the possibility of obtaining successful results of anti-cancer therapy. Therefore, it is necessary to search for new, effective chemotherapeutics characterized by low toxicity and high selectivity profile. Analysis of the Food and Drug Administration (FDA) database revealed that of the novel approved drugs for oncology in 2018, 7 are biologicals and 13 are small-molecules drugs. Importantly, all of the newly reported chemotherapeutics contain an *N*-heterocyclic fragment in their structure [1]. Small molecule nitrogen heterocycles are very important structures, widely used in the design of potential pharmaceuticals for several years [2]. Among them, the quinoline scaffold, which is present in many classes of biologically active compounds, deserves special attention [3,4,5,6]. The quinoline scaffold is often used as a privileged structure in the design of a large number of structurally diverse molecules that exhibit promising pharmacological effects. Particularly, derivatives of 8-hydroxyquinoline (8-HQ) are successfully applied to design chemotherapeutics used in the treatment of bacterial [7,8,9], fungal [10,11] and neurodegenerative diseases [12,13]. An important element of 8-HQ biological activity is the ability to chelate metal ions, which makes 8-HQ a promising anti-cancer agent [14,15,16].

Transition metal ions, including Fe^2+^, Cu^2+^, and Zn^2+^ play a significant role in the human organism. They affect the proper course of many key cellular processes. Therefore, it is important to maintain homeostasis. Incorrect metabolism of the above microelements may contribute to the development of many diseases [17,18]. There are several strategies for controlling the proper level of metals within cells and tissues. For this purpose, a common strategy is the use of chelators or ionophores. Chelating agents are successfully used to remove excess metal ions from the body. Whereas, ionophores are used to transport metal ions via cell membrane in both directions [18,19]. Copper is one of the transition metals which complex compounds have been intensively studied in recent years for anticancer applications. As is well known, copper is an essential cofactor for cancer growth and angiogenesis. Numerous studies have shown that an elevated level of copper is directly correlated with cancer progression. Additionally, copper concentration correlated with the age of patients and the stage of cancer [20,21,22,23,24]. Excess of copper ions was found in the serum and tissues of patients with, among others: breast [24,25], prostate [26,27], colorectal [28], lung [29] and brain [30] cancers, compared to healthy people. These observations suggest that copper ions may be one of the selective targets for cancer treatment [20,31,32,33].

Among the known copper chelators include tetrathiomolybdate, trientine, and D-penicillamine primarily used to treat Wilson′s disease [34]. Clinical studies have shown that these compounds are effective in inhibiting the angiogenesis of some types of cancer [35,36,37,38,39,40,41]. The literature also reports on the use of 8-HQ derivatives as copper chelators. Due to the presence of free electron pairs on nitrogen and oxygen atoms of 8-HQ, they have the ability to complex Cu^2+^ ions, forming chelates, which are then removed from the body [14,15,16,42,43]. However, metal ions play an important role in many cellular processes, including those necessary for the proper functioning of the organism. Therefore, traditional chelators can cause a deficiency of copper in healthy tissues. The consequence of the lack of selectivity in directing drugs to a specific site of action may be undesirable side effects resulting from the implemented anticancer therapy. That is why the aim of many researchers is to develop more selective and safer drugs. Therefore, an ideal anti-cancer drug should be selective for cancer cells, thereby alleviating the undesirable toxic effects of chemotherapy.

During designing new drugs, it is very important to know and use the differences between cancer cells and healthy cells. One such difference is the specific metabolism of glucose in cancer cells. It has been proven that cancer cells have an increased demand for glucose compared to healthy cells, which metabolizes it to obtain the energy needed to increase proliferation. This provides the cells with a sufficient amount of nutrients and energy to carry out the processes taking place during the cell cycle. This phenomenon is known as the Warburg effect and arises from mitochondrial metabolic changes. It consists in the fact that cancer cells produce their energy through glycolysis followed by lactic acid fermentation, characteristic of hypoxic conditions, and its level is much higher (more than a hundred times) than in healthy cells, for which the main source of energy is mitochondrial oxidative phosphorylation [44,45]. Increased glycolysis process in cancer cells is associated with the overexpression of GLUT transporters. There are special proteins that mediate the transfer of sugars across cell membranes [46,47,48]. Therefore, sugars are an attractive system for transporting drugs directly to cancer cells. The strategy of conjugating sugar derivatives with biological active aglycons is widely used in research on the synthesis of new drugs, supporting the treatment against various diseases [49]. The chelating functions of glycoconjugates are masked by the presence of a sugar unit until the release of active aglycon in the target cells, occurring as a result of hydrolysis catalyzed by specific β-glycosidases [50]. Consequently, drug interference in the glycolytic pathway should be effective and more selective for cancer cells.

As a result of conjugating 8-HQ derivatives with sugar derivatives, a molecule with better bioavailability, selectivity and solubility can be obtained [50,51,52,53]. Considering the potential therapeutic application of planned connections, this synthesis must be simple, fast, selective and efficient. An example of a reaction that meets these criteria is the reaction of 1,3-dipolar azide-alkyne cycloaddition, which leads to receiving the 1,2,3-triazole ring between joined compounds.

We have recently shown that the 1,2,3-triazole ring is an important element for glycoconjugate activity [54] (Figure 1). Except for the role of a connection between two active groups, it also has other interesting properties. The H-C(5) atom in the triazole ring is a donor for a hydrogen bond, while the free electron pair at the N(3) atom in the triazole can be used as an acceptor in the formation of a hydrogen bond. Their tendency to form hydrogen bonds increases the solubility of such molecules, which favor binding to biomolecular targets [55]. The aim of present work is to extend the library of such combinations. The general structure of the planned glycoconjugates is shown in Figure 2. It was decided to reverse the direction of linking reactive moieties compared to the previously reported glycoconjugates [54], consisting of sugar azides and propargyl derivatives of 8-HQ (Figure 1) and see how this will affect the activity of glycoconjugates. In addition, it was decided to check how the structure modification of the linker of the quinoline glycoconjugates affects their biological properties. It was assumed that an extension of the alkyl chain between the triazole ring and quinoline or sugar part should increase the “flexibility” of the obtained compounds, which may affect the cytotoxicity profile of the molecules. Moreover, insertion into a linker an additional amide or carbamate bonds, the presence of which is observed in many biologically active compounds should improve glycoconjugates chelating ability [56]. Therefore, it was decided to check whether the addition of this structural element to quinoline glycoconjugates will improve their activity compared to previously obtained derivatives.

A multiplicity of structural modifications in building blocks for the synthesis of glycoconjugates and their combination in various configurations will allow determining the relationships between the chemical structure of this group of compounds and their biological activity. This will allow also identifying those specific structural elements that are responsible for demonstrating biological activity. This is important information from the point of view of the mechanisms of action of designed structures.

## 2. Results and Discussion

### 2.1. Synthesis

As mentioned above, the synthetic aim of this work was to create a large library of compounds based on the 8-HQ scaffold with a sugar fragment attached via an appropriate linker. Target glycoconjugates were obtained by modifying commercially available 8-hydroxyquinoline **1** or 8-hydroxyquinaldine **2**, and then connecting them with derivatives of D-glucose or D-galactose substituted with different groups at the anomeric position. A common feature of all glycoconjugates is the presence of a 1,2,3-triazole ring in the linker structure. This improves their cytotoxic activity, probably by increasing the ability to metal ions chelation found in many types of cancers [54,55]. In addition, the used substrates have a relatively small size and are easy to modify, which makes them particularly interesting from the point of view of using them to design potential drugs.

The first part of the synthesis concerned the preparation of the corresponding quinoline derivatives functionalized in the 8-OH position with propargyl or azide groups, that are involved in the *click chemistry* reaction. The path of the syntheses is presented in Scheme 1.

Propargyl quinoline derivatives **3** or **4** were obtained in good yields (88% and 73% respectively), according to the previously published procedure [54,57]. The corresponding compound **1** or **2** was reacted with propargyl bromide in a reaction carried out under basic conditions. The first approach to the synthesis of 8-(2-azidoethoxy)quinolone **5** was the reaction of 8-HQ **1** with 2-bromoethanol to obtain 8-(2-hydroxyethoxy)quinoline. The obtained alcohol was treated with methanesulfonyl or *p*-toluenesulfonyl chloride in the presence of a non-nucleophilic base (TEA) followed by sodium azide in DMF. However, the formation of the expected product **5** was no observed. Also in case of lengthening the alkyl chain of the donor, the desired products **6–10** could not be obtained. Treatment of 8-(2-hydroxyethoxy)quinoline by azidotrimethylsilane (TMSN_3_) in the presence of Lewis acid also did not give the expected products. The application of the Appel reaction conditions, in which the alcohol reacts with carbon tetrachloride in the presence of triphenylphosphine also not allowed to obtain the desired products. Instead, spectroscopic data (HRMS) confirmed the formation of a resonance-stabilized structure of the tricyclic oxazaquinolinium salts **5a–10a** [58]. Finally, to obtain compounds **5–10**, a reaction of quinoline derivatives **1** or **2** with 1-azido-2-bromoethane, 1-azido-3-bromopropane or 1-azido-4-bromobutane as donors of desired groups was carried out. These donors were previously obtained by monoazidation of the corresponding dibromoalkane with NaN_3_ in DMF [59]. The optimal yield of the desired products was obtained by carrying out the reaction at 50 °C, using an equimolar ratio of substrates. Except for the monosubstituted derivatives (50% yields), diazidesubstituted derivatives (25% yield) were also obtained (the yield was estimated by ^1^H NMR).

Sugar derivatives substituted at the anomeric position were a second necessary structural element for the synthesis of glycoconjugates. The synthesis route to the corresponding protected and deprotected derivatives of D-glucose and D-galactose are shown in Scheme 2, Scheme 3 and Scheme 4. The choice of sugar units is dictated by the frequency of their occurrence and their importance for cell metabolism. The procedure for obtaining 2,3,4,6-tetra-*O*-acetyl-β-d-glycopyranosyl amines **19** and **20** has been described in detail in earlier works [51,54,60] (Scheme 2).

Sugar derivatives in which the alkynyl or azide moiety was introduced by the formation of an *O*-glycosidic linkage were prepared by reacting per-*O*-acetylated D-glucose **13** or D-galactose **14** with propargyl alcohol or 2-bromoethanol in the presence of a Lewis acid as a catalyst (Scheme 3) [61,62]. The reactions were carried out in an anhydrous DCM in the presence of boron trifluoride etherate until the complete conversion of the substrate, which was monitored by TLC. The reaction mixture was diluted with dichloromethane and extracted with NaHCO_3_ and brine to wash off the acid. The acetyl neighboring-group participation at the C-2 position of the sugar ensured the formation of an intermediate acyloxonium ion, which for steric reasons could be “attacked” by a nucleophile only from the opposite side, resulting in only products of the β-configuration. This is confirmed by the large coupling constant from the H-1 proton equal *J* = 8.0 Hz observed in the ^1^H NMR spectra for compounds **21**–**22** and **25**–**26**. As a result, propargyl 2,3,4,6-tetra-*O*-acetyl-β-D-glycopyranosides **21** and **22** were obtained in good yields (89% and 83% respectively), whereas 2-bromoethyl 2,3,4,6-tetra-*O*-acetyl-β-D-glycopyranosides **25** and **26** were obtained in slightly lower yields (76% and 71% respectively). 2-Azidoethyl 2,3,4,6-tetra-*O*-acetyl-β-D-glycopyranosides **27** and **28** were obtained by substituting bromine by sodium azide in the reaction carried out in DMF in almost quantitative yield. Confirmation of bromine exchange to the azide moiety was the appearance in the ^13^C NMR spectra of the signal of CH_2_N_3_ carbon with a shift of about δ = 50.55 ppm for **27** and 50.60 ppm for **28**, while the signal of CH_2_Br carbon was observed at about δ = 29.83 ppm for **25** and 29.91 ppm for **26**.

In the next step, sugar derivatives containing an amide or a carbamate moiety at the sugar anomeric position were obtained by several-step synthesis presented in Scheme 4.

Derivatives of 2,3,4,6-tetra-*O*-acetyl-*N*-(β-D-glycopyranosyl)azidoacetamide **33, 34** were obtained in two-steps procedure. 1-Aminosugars **19** or **20** were reacted with chloroacetyl chloride in the presence of TEA, which neutralized the formed of hydrogen chloride. This approach eliminates the use of toxic SnCl_4_ and tin metal to reduce glycopyranosyl azides, which has been described as a method for the synthesis of *N*-glycopyranosyl chloroacetamides [63]. In the second step, the terminal chlorine atom in compounds **31** and **32** was exchanged with an azide group by a nucleophilic substitution reaction with sodium azide in dry DMF. The structures of the obtained products were confirmed on the basis of NMR spectra analysis. In this case, the characteristic carbon signal of CH_2_N_3_ located at δ = 52.59 ppm for **33** and 52.61 ppm for **34** in ^13^C NMR spectra was shifted from δ = 42.25 ppm and 42.26 ppm corresponding to chloroacetamide derivatives **31** and **32**.

In turns, structures **37** and **38** were obtained in the reaction of 1-amino sugars **19** or **20** with propargylic acid. In these reactions activating the non-reactive carboxyl group seems to be necessary. Of the many ways to form an amide bond, initially, it was decided to use the method developed by Kaminski and co-workers [64]. This method involves creating so-called superactive ester 4-(4,6-dimethoxy-1,3,5-triazin-2-yl)-4-methylmorpholinium chloride (DMTMM) in situ in reaction between 2-chloro-4,6-disubstituted-1,3,5-triazines (CDMT) and *N*-methylmorpholine (NMM). It turned out that as a result of this reaction, only traces of expected products were formed. In the next approach, the carbodiimide method was used to the condensation reaction using DCC as a coupling agent [65]. The reaction was carried out in DCM in which the isolated DCU is insoluble and could be separated from the reaction mixture by filtration. As a result, 2,3,4,6-tetra-*O*-acetyl-*N*-(β-d-glycopyranosyl)propiolamides **37** and **38** were obtained in high yields (90% and 77% respectively).

In order to obtain sugar derivatives containing a carbamate group at the sugar anomeric position, reactions of 1-amino sugars **19** and **20** with propargyl chloroformate, which is a reactive acylating agent, were also carried out. This reaction requires the presence of a tertiary amine in the reaction medium to entrap the formed HCl and avoid conversion of the amine into its non-reactive hydrochloride salt [66]. However, the use of TEA gave only moderate yield, while the replacement of TEA with *N,N*-diisopropylethylamine allowed obtaining the desired products **41** and **42** in high yields (81% and 88% respectively).

The ester type protecting groups in the sugar part increase the lipophilicity of glycoconjugates and improved its passive transport inside the cell, where are enzymes capable of hydrolyzing them. However, for some biological studies, it is also necessary to use compounds with unprotected hydroxyl groups. The removal of the acetyl protecting groups from sugars was carried out according to the classic Zemplén protocol under alkaline conditions using a solution of sodium methoxide in methanol (0.2 molar equiv.) [67]. The final step was to neutralize the reaction mixture with the use of Amberlyst-15 ion exchange resin, after which the mixture was filtered to give compounds **23, 24, 29, 30, 35, 36, 39, 40, 43, 44** sufficiently pure for further reactions.

For the preparation of glycoconjugates, the obtained protected or deprotected derivatives of D-glucose or D-galactose were combined with derivatives of 8-HQ using copper(I)-catalyzed 1,3-dipolar azide-alkyne cycloaddition (CuAAC) [68,69]. A general scheme for the synthesis of glycoconjugates is shown in Scheme 5. The reactants were combined in an equimolar ratio in a THF/*i*-PrOH/H_2_O solvent system at room temperature. As the source of copper ions, CuSO_4_·5H_2_O was used. Whereas, sodium ascorbate (NaAsc) was a reducing agent for Cu ions from II for the I oxidation stage. Due to the using Cu(I) as a catalyst, the reaction was carried out at room temperature and only 1,4-disubstituted 1,2,3-triazoles were obtained. It is worth noting that both protected and deprotected sugar derivatives can be used for this reaction. This eliminates the need for a final deprotection of glycoconjugates, which could adversely affect the yields of the unprotected products. The crude products of these reactions were purified by column chromatography. As a result of the CuAAC reaction, glycoconjugates **45**–**108** were obtained with the yields shown in the Table 1, Table 2, Table 3, Table 4 and Table 5. The structure of substrates for CuAAC reactions and all obtained glycoconjugates were confirmed by ^1^H and ^13^C NMR spectroscopy. For glycoconjugates, HRMS analyses were also performed. All these data are included in the Appendix A. The physicochemical properties, such as melting point and optical rotation, were also determined.

### 2.2. Biological Studies

#### 2.2.1. Cytotoxicity Evaluation of Glycoconjugates

Glycoconjugates have been tested for their potential anticancer activity in vitro. For this purpose, the whole range of obtained glycoconjugates and substrates used for their synthesis were subjected to a cytotoxicity screening using MTT assay on cell lines: HCT 116 (colorectal carcinoma cell line) and MCF-7 (human breast adenocarcinoma cell line). In these lines, overexpression of the glucose and galactose transporters was observed [49,70,71,72,73]. For compounds presenting the highest antiproliferative activity within the cancer cell line, cytotoxicity tests on Normal Human Dermal Fibroblast-Neonatal cells (NHDF-Neo) were also be performed, to determine their selectivity index. Tests were conducted for glycoconjugates solutions at concentrations ranging from 0.01 mM to 0.8 mM. The determined IC_50_ values, defined as 50% cell growth inhibition compared to the untreated control, are shown in Table 6 and Table 7). The effect of the length and structure of the linker connecting quinoline with the sugar fragment for the activity of glycoconjugates was investigated.

As part of the experiments, it was checked whether the building blocks necessary to obtain final glycoconjugates are able to limit the proliferation of cancer cells (Table 6). Sugar derivatives appeared to be inactive on the tested cell lines. However, the high toxicity of parent compounds **1** and **2** towards the MCF-7 cancer cell line was observed. A lower IC_50_ value was determined for 8-HQ than for doxorubicin commonly used in cancer treatment. This observation indicates the huge sensitivity of this particular cancer cell line to 8-HQ and gives hope for even more effective therapeutics based on the 8-HQ scaffold. In addition, some of the derivatives of 8-HQ **5**–**10** were toxic to the tested cell lines. However, further experiments have shown that they also cause the death of healthy cells. Therefore, the next studies checked whether the addition of the sugar moiety to aglycon would affect the selectivity of the obtained glycoconjugates.

The results of the cytotoxicity assay indicate that the glycoconjugates with a deprotected sugar unit (right part of Table 7) are mostly unable to inhibit cell proliferation in the tested concentration range. This is probably related to their hindered penetration into the cell through the lipid biological barriers, due to the high hydrophilicity of the compounds, compared to glycoconjugates having acetyl protecting groups. The exception is compound **81**, whose IC_50_ = 112.8 ± 1.6 µM for HCT 116 and 87.9 ± 4.1 µM for MCF-7. For its protected analog **77**, the IC_50_ value twice as high was determined for both tumor cell lines. Most likely, in this case, GLUT transporters had a more important role in transporting into the cell, and passive transport is less important. Unfortunately, this compound is also toxic to healthy cells, so it cannot be considered a selective drug.

All glycoconjugates containing ester protection of hydroxyl groups in the sugar part proved to be active on the tested cancer cell lines in the tested concentration range (left part of Table 7). Their non-polar nature should facilitate the process of crossing through phospholipid bilayer and penetrating into the cell, where then intracellular hydrolytic enzymes are able to remove the acetyl groups. In addition, as opposed to substrates, some of the obtained glycoconjugates showed low cytotoxicity to NHDF-Neo cells, especially glycoconjugates **45**, **46**, **85**, and **86** derived from quinoline **5**. Moreover, for the HCT 116 cell line, all protected glycoconjugates exhibit higher antiproliferative activity relative to aglycons **1** and **2**. This fact confirms the accuracy of the assumption that in this case, the presence of the sugar fragment improves the distribution and absorption of the compound, and thus its activity.

In the beginning, it was decided to check the effect of the length of the linker between the 1,2,3-triazole fragment and the derivative of 8-HQ for glycoconjugate activity (compounds **45**–**68**). The lowest IC_50_ values were obtained for glycoconjugates **53**–**56**, whose alkyl chain between triazole and 8-HQ consisted of 3 carbon atoms. However, further elongation of the alkyl chain in the compounds **61**–**64** resulted in a decrease in the cytotoxic activity of glyconjugates. Therefore, no further chain extension seems necessary. On the other hand, the alkyl chain extension between the 1,2,3-triazole ring and the sugar moiety in compounds **69**–**72** did not significantly affect the antiproliferative activity of glycoconjugates. Noteworthy is compound **71**, which turned out to be more active than the unconjugated quinoline derivative **4** and for which the selectivity index, calculated as the ratio of the IC_50_ value determined for healthy cells to the IC_50_ value determined for tumor cells, equal 5.3 for MCF-7 lines and 2.5 for HCT 116 lines. This observation makes it interesting in the aspect of further, more detailed studies involving different cell lines.

Glycoconjugates **77**–**100** containing an additional amide bond in the linker structure proved to be more active relative to the tested cancer cell lines. Among them, the most promising results were obtained for compounds **78** and **93**. For the MCF-7 cell line, the IC_50_ value equals 39.1 ± 0.9 µM and 95.7 ± 0.02 µM, respectively. However, for HCT 116 not much higher IC_50_ were noted: 119.1 ± 5.6 µM and 137.3 ± 2.1 µM, respectively. Unfortunately, these compounds were also toxic to healthy cells, so they cannot be considered selective drugs. It was noted that compound with 8-HQ fragment usually showed higher cytotoxicity compared to derivatives with 2Me8HQ unit, while the type of sugar moiety did not significantly affect to the glycoconjugate activity. Importantly, compounds **86–88** did not show any ability to inhibit the proliferation of healthy cells while having a moderate ability to inhibit cancer cells.

Recently, an important role in drug design has been played by compounds containing a carbamate moiety in the structure. Carbamates are usually more stable under enzymatic hydrolysis than the corresponding esters and are generally more susceptible to hydrolysis than amides. Carbamate prodrugs have been designed for selective hydrolysis by human carboxylesterases to release active drugs [56,74]. It appeared that glycoconjugates **101** and **103**, based on the D-glucose moiety, showed low cytotoxicity to healthy cells while showing significant antiproliferative activity against cancer cells. For these compounds, a selectivity index for the MCF-7 cell line was 3.9 and 3.3 respectively.

#### 2.2.2. Inhibitory Activity Against β-1,4-GalT

Tested cancer cell lines are characterized by overexpression of the β-1,4-galactosyltransferase (β-1,4-GalT) [75]. This enzyme belongs to the group of glycosyltransferases (GTs). Due to a number of important functions that GTs perform, among others: post-translational protein modifications and synthesis of oligosaccharide chains, they are an important object of research on potential anticancer drugs. A high rate of glycosylation is a common disorder of cancer cell metabolism, and the expression of GTs can be associated with cancer progression. GTs are a group of metal-dependent enzymes, therefore the presence of a species capable of binding divalent metal ions in the molecule of potential inhibitor appears to be necessary to inhibit the activity of these enzymes [75,76]. It seems appropriate to use glycoconjugates derivatives of 8-HQ to coordinate the metal ions present in the active centers of many enzymes, and thus to inhibit their activity.

It was decided to evaluate the obtained glycoconjugates for their inhibitory activity against enzyme from the glycosyltransferases group. Therefore, the experiments will be carried out using commercially available metal-dependent β-1,4-GalT I from bovine milk. To evaluate the activity of tested compounds, concentrations of substrate and product of the enzymatic reaction in the reaction mixtures was determined by RP-HPLC method, which is a modification of the Vidal method [77]. This method uses UDP-Gal, a natural β-1,4-GalT donor type substrate and (6-esculetinyl) β-D glucopyranoside (esculine) as glycosyl fluorescent acceptor. The number of products formed in the reaction with the addition of glycoconjugates as potential enzyme inhibitors was compared with the number of products in reactions carried out under the same conditions without the addition of inhibitors (test reactions). Analyzes were conducted in the linear range of the peak area from the product and substrate concentration. Experiments were conducted for glycoconjugates solutions at concentrations ranging from 0.1 mM to 0.8 mM. For the most active compounds, IC_50_ values were designated. The results are presented in Table 8.

In this study, the influence of the alkyl chain length between the 1,2,3-triazole fragment and the 8-HQ derivative (**49**–**52, 57**–**60, 65**–**68**), as well as between the 1,2,3-triazole fragment and the sugar unit (**73**–**76**) for enzyme inhibition was tested. In addition, the effect of the presence of an additional amide bond in the structure of the glycoconjugate linker (**81**–**84**) was analyzed. Based on previous experience [54], only glycoconjugates with an unprotected sugar part were tested.

Experiments have shown that parent compounds **1**, **2** and substrates **3**–**10** are not able to inhibit the enzyme, which may indicate that the sugar fragment is necessary to obtain inhibitory activity. For glycoconjugates obtained as a result of CuAAC reactions between sugar derivatives containing a propargyl moiety and quinoline derivatives containing an azide moiety, the results indicate that all tested glycoconjugates derivatives of D-glucose showed higher activity compared to their analogs containing the D-galactose unit. However, the type of 8-HQ derivative did not matter in this case. Considering the effect of linker length between the 1,2,3-triazole fragment and the 8-HQ derivative, it can be clearly stated that the extension of the alkyl chain increases the inhibition of β-1,4-GalTI by glycoconjugates. In this group of compounds (**49**–**52, 57**–**60, 65**–**68**), all glycoconjugates based on D-glucose moiety showed the ability to inhibit the enzyme by over 50% compared to the test reaction. However, for compound **67**, having four carbon atoms in the alkyl linker, the lowest IC_50_ value was determined (0.35 mM). It is probable, that an extension of the alkyl chain length between the triazole ring and quinolone, increases the “flexibility” of the molecule, which may have a better fit into the active center of the enzyme. Much lower ability to inhibition of the model enzyme was observed for glycoconjugates that were obtained in the CuAAC reaction of sugar derivatives containing an azide group and quinoline derivatives containing the propargyl moiety. Glycoconjugates with an ethoxy fragment between the 1,2,3-triazole and sugar units (**73**–**76**) were not able to inhibit β-1,4-GalTI by 50%. However, they showed slightly increased activity compared to glycoconjugates **81**–**84**, containing an additional amide fragment in the structure. The amide fragment is in close distance to the aromatic moiety, which creates a high rigidity of the molecule and probably makes it difficult to fit into the active center of the enzyme. Therefore, the addition of an amide fragment is not a good idea for the design of β-1,4-GalT inhibitors.

### 2.3. Study of Metal Complexing Properties

Previously it was mentioned that cancer cells exhibit an increased concentration of copper ions [20,21,22,23,24]. Moreover, it is broadly known that some quinoline derivatives possess metal ions complexing properties [18] what is essential for their anti-proliferative properties [51]. Bivalent metal ions are coordinated by phenol oxygen atom as well as nitrogen atom of 8-HQ ring with the formation of an 8HQ-metal ion complex with stoichiometry 2:1 [78]. In addition, 1,2,3-triazole ring also has a metal-complexation ability [79]. Therefore, it is interesting if the addition of linker containing such fragment having metal ions chelation ability into 8-HQ might significantly increase the metal ion complexation capability. The capability of complexes formation by investigated glycoconjugates and stoichiometry of such complexes was evaluated for compounds **101** and **93** with copper ions by spectroscopic titration experiments using UV-VIS. The UV-VIS spectrum of **101** is shown in Figure 3a. The addition of subsequent copper ion portions to glycoconjugate solution results in a gradual lowering of absorption bands at 242 nm and 310 nm. In addition, increase of the absorption band at 265 nm was observed. Importantly, during titration two well-defined isosbestic points around 250 nm and 286 nm were noted indicating the glycoconjugate-copper ion complex formation. The stoichiometry of obtained complexes was determined with Job′s plot using the changes in absorption band at 265 nm. The plotted graph presenting the difference in absorbance, ΔA = A_x_ − A_0_ as a function of molar fraction [101]/[**101** + Cu^2+^] was shown in Figure 3b. Maximum observed at molar fraction 0.5 in obtained curve indicates a formation of 1:1 complex of glycoconjugate:copper ion, regardless of the type of linker (see Appendix A).

ESI-MS spectra showed that copper ions are complexed by investigated glycoconjugates. The ESI-MS spectrum in positive ions mode of the investigated compound **101** (Figure 4a) revealed the presence of the ion at *m*/*z* = 658.35 (adduct [**101** + H]^+^) corresponding to protonated glycoconjugate molecule and less intensive one corresponding to its sodium adduct, *m/z* = 680.36 [**101** + Na]^+^. Additionally, ions of **101_2_** aggregate (dimer) adducts can be noticed, respectively at *m*/*z* = 1314.23 [**101_2_** + H]^+^, 1337.26 [**101_2_** + Na]^+^ and 1353.19 [**101_2_** + K]^+^. In the ESI-MS of glycoconjugate partially titrated with copper ions (Figure 4b) the most abundant peak at *m*/*z* = 720.15 correspondings to [**101** + Cu(I)]^+^ is visible. The second intensive signal *m*/*z* = 658.2 is derived from glycoconjugate still present in the solution. Moreover the signals at *m*/*z* = 1377.17 and at *m*/*z* = 360.28 corresponding to dimer **101_2_** complex with Cu [**101_2_** + Cu(I)]^+^ and complex [**101** + Cu(II)]^2+^ were observed. The ESI-MS/MS fragmentation spectrum of the ion with *m*/*z* = 1377.17 (see Appendix A) reveals only one peak at *m*/*z* = 720 corresponding to complex [**101** + Cu(I)]^+^. It confirms that ion at *m*/*z* = 1377.17 corresponds to Cu^+^-dimer complex (**101_2_** is present in the initial solution, see Figure 4a). Thus, complex with apparent 2:1 stoichiometry is observed, but in fact, the complex is formed probably only with one of the **101** molecules of the dimer. Moreover, in the signals ascribed to formed complexes characteristic two peaks with the difference of 2 Da are observed. Such spectrum is typical of copper complexes due to existence of copper in two basic isotopic forms (^63^Cu (69,17%) and ^65^Cu (30,83%)) [80]. It may seem surprising that copper(I) complexes are observed in the ESI-MS spectrum since copper(II) was used for titration. This phenomenon can be explained by a reduction reaction proceeding under the conditions of the analysis as a result of charge transfer between the metal complexes and the solvent molecules in the gas-phase [81,82,83].

## 3. Materials and Methods

### 3.1. General Information

NMR spectra were recorded with an Agilent spectrometer at a frequency of 400 MHz using TMS or DSS as the internal standards and CDCl_3_, CD_3_OD, DMSO-d6 or D_2_O as the solvents. NMR solvents were purchased from ACROS Organics (Geel, Belgium). Chemical shifts (*δ*) are expressed in ppm and coupling constants (*J*) in Hz. The following abbreviations were used to explain the observed multiplicities: s: singlet, d: doublet, dd: doublet of doublets, ddd: doublet of doublet of doublets, t: triplet, dd-t: doublet of doublets resembling a triplet (with similar values of coupling constants), m: multiplet, p: pentet (quintet), b: broad. High-resolution mass spectra (HRMS) were recorded with a WATERS LCT Premier XE system using the electrospray-ionization (ESI) technique. Optical rotations were measured with a JASCO P-2000 polarimeter using a sodium lamp (589.3 nm) at room temperature. Melting point measurements were performed on OptiMelt (MPA 100) Stanford Research Systems. Reactions were monitored by thin-layer chromatography (TLC) on precoated plates of silica gel 60 F254 (Merck Millipore, Burlington, MA, USA). The TLC plates were visualized under UV light (λ = 254 nm) or by charring the plates after spraying with 10% solution of sulfuric acid in ethanol. Crude products were purified using column chromatography performed on Silica Gel 60 (70–230 mesh, Fluka, St. Louis, MI, USA), developed using toluene:EtOAc or CHCl_3_:MeOH as solvent systems. All evaporations were performed on a rotary evaporator under diminished pressure at 40 °C. Reversed-phase HPLC analyses were performed using JASCO LC 2000 apparatus equipped with a reverse-phase column (Nucleosil 100 C18.5 μm, 25 × 0.4 cm; mobile phase: H_2_O/MeCN 90:10, flow rate 0.8 mL/min) with a fluorescence detector (FP). Fluorescence for substrate and product was read at 385 nm excitation/540 nm emission. The absorbance on MTT assay was measured spectrophotometrically at the 570 nm wavelength using a plate reader (Epoch, BioTek, USA).

All of the chemicals used in the experiments were purchased from Sigma-Aldrich, ACROS Organics, Fluka and Avantor and were used without purification. 8-Hydroxyquinoline **1**, 8-hydroxyquinaldine **2**, D-glucose **11** and D-galactose **12** are commercially available (Sigma-Aldrich). 8-(2-Propyn-1-yloxy)quinoline **3** [57], 2-methyl-8-(2-propyn-1-yloxy)quinoline **4** [57], 1,2,3,4,6-penta-*O*-acetyl-β-D-glucopyranose **13** [54], 1,2,3,4,6-penta-*O*-acetyl-β-D-galactopyranose **14** [54], 2,3,4,6-tetra-*O*-acetyl-α-D-glucopyranosyl bromide 1**5** [54], 2,3,4,6-tetra-*O*-acetyl-α-D-galactopyranosyl bromide 1**6** [54], 2,3,4,6-tetra-*O*-acetyl-β-D-glucopyranosyl azide 1**7** [54], 2,3,4,6-tetra-*O*-acetyl-β-D-galactopyranosyl azide 1**8** [54], 2,3,4,6-tetra-*O*-acetyl-β-D-glucopyranosyl amine **19** [60], 2,3,4,6-tetra-*O*-acetyl-β-D-galactopyranosyl amine **20** [60], propargyl 2,3,4,6-tetra-*O*-acetyl-β-D-glucopyranoside **21** [61], propargyl 2,3,4,6-tetra-*O*-acetyl-β-D-galactopyranoside **22** [61], 2-bromoethyl 2,3,4,6-tetra-*O*-acetyl-β-D-glucopyranoside **25** [62], 2-bromoethyl 2,3,4,6-tetra-*O*-acetyl-β-D-galactopyranoside **26** [62], 2-azidoethyl 2,3,4,6-tetra-*O*-acetyl-β-D-glucopyranoside **27** [62], 2-azidoethyl 2,3,4,6-tetra-*O*-acetyl-β-D-galactopyranoside **28** [62], 2,3,4,6-tetra-*O*-acetyl-*N*-(β-D-glucopyranosyl)propiolamide **37** [65], 2,3,4,6-tetra-*O*-acetyl-*N*-(β-D-galactopyranosyl)propiolamide **38** [65], 2,3,4,6-tetra-*O*-acetyl-*N*-(β-D-glucopyranosyl)-*O*-propargyl carbamate **41** [66] and 2,3,4,6-tetra-*O*-acetyl-*N*-(β-D-galactopyranosyl)-*O*-propargyl carbamate **42** [66] were prepared according to the respective published procedures. Propargyl β-D-glucopyranoside **23**, propargyl β-D-galactopyranoside **24**, 2-azidoethyl β-D-glucopyranoside **29**, 2-azidoethyl β-D-galactopyranoside **30**, *N*-(β-D-glucopyranosyl)azidoacetamide **35**, *N*-(β-D-galactopyranosyl)azidoacetamide **36**, *N*-(β-D-glucopyranosyl)propiolamide **39**, *N*-(β-D-galactopyranosyl)propiolamide **40**
*N*-(β-D-glucopyranosyl)-*O*-propargyl carbamate **43**, *N*-(β-D-galactopyranosyl)-*O*-propargyl carbamate **44** were obtained by Zemplén protocol [67] by deacetylation of the corresponding sugar derivatives.

### 3.2. Chemistry

#### 3.2.1. General Procedure for the Synthesis of Quinoline Derivatives **5**–**10**

To a solution of 1,2-dibromoethane or 1,3-dibromopropane or 1,4-dibromobutane (23.1 mmol) in dry DMF (10 mL), sodium azide (23.1 mmol, 1.5 g) was added. The reaction mixture was stirred overnight at 50 °C. Afterwards, the reaction mixture was diluted with ether and the organic layer was washed (three times) with H_2_O (3 × 10 mL). The organic layer was dried over anhydrous magnesium sulfate (MgSO_4_), concentrated under vacuum to afford the corresponding azide as a clear oil, which was used in the next reaction without further purification.

Obtained 1-azido-2-bromoethane or 1-azido-3-bromopropane or 1-azido-4-bromobutane (4.0 mmol), was added to a solution of 8-hydroxyquinoline **1** or 8-hydroxyquinaldine **2** (4.0 mmol) in acetone (20 mL), followed by addition of potassium carbonate (10.0 mmol, 1.38 g). The reaction mixture was heated under reflux for 4 h and then at room temperature overnight. After completion, the reaction mixture was filtered off and the filtrate was concentrated under vacuum and purified by column chromatography (toluene:AcOEt; gradient 20:1 to 2:1) to give products **5**–**10**.

8-(2-Azidoethoxy)quinolone **5**: Starting from 1-azido-2-bromoethane and 8-hydroxyquinoline **1**, product was obtained as a brown oil (702.7 mg, 82%); [α]^24^_D_ = −0.6 (c = 1.0, CHCl_3_); ^1^H NMR (400 MHz, CDCl_3_): δ 3.87 (t, 2H, *J* = 5.6 Hz, CH_2_N), 4.43 (t, 2H, *J* = 5.6 Hz, CH_2_O), 7.12 (dd, 1H, *J* = 2.5 Hz, *J* = 6.5 Hz, H-7_chin_), 7.41–7.50 (m, 3H, H-3_chin_, H-5_chin_, H-6_chin_), 8.14 (dd, 1H, *J* = 1.7 Hz, *J* = 8.3 Hz, H-4_chin_), 8.96 (dd, 1H, *J* = 1.7 Hz, *J* = 4.2 Hz, H-2_chin_); ^13^C NMR (100 MHz, CDCl_3_): δ 50.06, 67.64, 109.77, 120.66, 121.70, 126.52, 129.61, 135.95, 140.39, 149.51, 154.19.

8-(3-Azidopropoxy)quinolone **6**: Starting from 1-azido-3-bromopropane and 8-hydroxyquinoline **1**, product was obtained as a brown oil (684.7 mg, 75%); [α]^24^_D_ = −0.4 (c = 1.0, CHCl_3_); ^1^H NMR (400 MHz, CDCl_3_): δ 2.28 (p, 2H, *J* = 6.4 Hz, CH_2_), 3.65 (t, 2H, *J* = 6.6 Hz, CH_2_N), 4.33 (t, 2H, *J* = 6.2 Hz, CH_2_O), 7.09 (dd, 1H, *J* = 1.4 Hz, *J* = 7.5 Hz, H-7_chin_), 7.38–7.49 (m, 3H, H-3_chin_, H-5_chin_, H-6_chin_), 8.13 (dd, 1H, *J* = 1.8 Hz, *J* = 8.3 Hz, H-4_chin_), 8.95 (dd, 1H, *J* = 1.8 Hz, *J* = 4.2 Hz, H-2_chin_); ^13^C NMR (100 MHz, CDCl_3_): δ 28.67, 48.44, 65.70, 109.08, 119.97, 121.62, 126.69, 129.55, 136.04, 140.28, 149.31, 154,50.

8-(4-Azidobutoxy)quinoline **7**: Starting from 1-azido-4-bromobutane and 8-hydroxyquinoline **1**, product was obtained as a brown oil (736.5 mg, 76%); [α]^23^_D_ = −0.4 (c = 1.0, CHCl_3_); ^1^H NMR (400 MHz, CDCl_3_): δ 1.90 (p, 2H, *J* = 7.0 Hz, CH_2_), 2.11 (p, 2H, *J* = 6.5 Hz, CH_2_), 3.42 (t, 2H, *J* = 6.9 Hz, CH_2_N), 4.29 (t, 2H, *J* = 6.5 Hz, CH_2_O), 7.07 (dd, 1H, *J* = 1.2 Hz, *J* = 7.6 Hz, H-7_chin_), 7.37–7.49 (m, 3H, H-3_chin_, H-5_chin_, H-6_chin_), 8.14 (dd, 1H, *J* = 1.7 Hz, *J* = 8.3 Hz, H-4_chin_), 8.95 (dd, 1H, *J* = 1.7 Hz, *J* = 4.2 Hz, H-2_chin_); ^13^C NMR (100 MHz, CDCl_3_): δ 25.86, 26.31, 51.29, 68.26, 108.90, 119.72, 121.58, 126.71, 129.56, 136.06, 140.26, 149.28, 154,59.

2-Methyl-8-(2-azidoethoxy)quinolone **8**: Starting from 1-azido-2-bromoethane and 2-methylo-8-hydroxyquinoline **2**, product was obtained as a brown oil (757.8 mg, 83%); [α]^23^_D_ = −0.4 (c = 1.0, CHCl_3_); ^1^H NMR (400 MHz, CDCl_3_): δ 2.79 (s, 3H, CH_3_), 3.83 (t, 2H, *J* = 5.5 Hz, CH_2_N), 4.43 (t, 2H, *J* = 5.5 Hz, CH_2_O), 7.10 (dd, 1H, *J* = 2.6 Hz, *J* = 6.4 Hz, H-7_chin_), 7.31 (d, 1H, *J* = 8.4 Hz, H-3_chin_), 7.35–7.44 (m, 2H, H-5_chin_, H-6_chin_), 8.03 (d, 1H, *J* = 8.4 Hz, H-4_chin_); ^13^C NMR (100 MHz, CDCl_3_): δ 25.66, 50.19, 68.05, 110.41, 120.62, 122.57, 125.49, 127.82, 136.00, 140.06, 153.78, 158,36.

2-Methyl-8-(3-azidopropoxy)quinolone **9**: Starting from 1-azido-3-bromopropane and 2-methylo-8-hydroxyquinoline **2**, product was obtained as a brown oil (688.1 mg, 71%); [α]^24^_D_ = −0.6 (c = 1.0, CHCl_3_); ^1^H NMR (400 MHz, CDCl_3_): δ 2.28 (p, 2H, *J* = 6.5 Hz, CH_2_), 2.78 (s, 3H, CH_3_), 3.65 (t, 2H, *J* = 6.6 Hz, CH_2_N), 4.33 (t, 2H, *J* = 6.3 Hz, CH_2_O), 7.07 (dd, 1H, *J* = 2.4 Hz, *J* = 6.5 Hz, H-7_chin_), 7.30 (d, 1H, *J* = 8.4 Hz, H-3_chin_), 7.33–7.42 (m, 2H, H-5_chin_, H-6_chin_), 8.01 (d, 1H, *J* = 8.4 Hz, H-4_chin_); ^13^C NMR (100 MHz, CDCl_3_): δ 25.71, 28.64, 48.48, 66.03, 109.73, 119.95, 122.52, 125.65, 127.76, 136.11, 140.00, 154.01, 158,18.

2-Methyl-8-(4-azidobutoxy)quinoline **10**: Starting from 1-azido-4-bromobutane and 2-methylo-8-hydroxyquinoline **2**, product was obtained as a brown oil (707.4 mg, 69%); [α]^22^_D_ = −0.6 (c = 1.0, CHCl_3_); ^1^H NMR (400 MHz, CDCl_3_): δ 1.89 (p, 2H, *J* = 7.0 Hz, CH_2_), 2.10 (p, 2H, *J* = 6.7 Hz, CH_2_), 2.78 (s, 3H, CH_3_), 3.45 (t, 2H, *J* = 6.9 Hz, CH_2_N), 4.27 (t, 2H, *J* = 6.5 Hz, CH_2_O), 7.03 (dd, 1H, *J* = 1.8 Hz, *J* = 7.1 Hz, H-7_chin_), 7.29 (d, 1H, *J* = 8.4 Hz, H-3_chin_), 7.32–7.41 (m, 2H, H-5_chin_, H-6_chin_), 8.00 (d, 1H, *J* = 8.4 Hz, H-4_chin_); ^13^C NMR (100 MHz, CDCl_3_): δ 25.71, 25.89, 26.24, 51.36, 68.46, 109.35, 119.66, 122.49, 125.64, 127.74, 136.08, 139.91, 154.13, 158,13.

#### 3.2.2. General Procedure for the Synthesis of Sugar Derivatives **31** and **32**

To a solution of 2,3,4,6-tetra-*O*-acetyl-β-D-glucopyranosyl amine **19** or 2,3,4,6-tetra-*O*-acetyl-β-D-galactopyranosyl amine **20** (1.7 g, 4.9 mmol) in dry CH_2_CL_2_ (20 mL), triethylamine (2 mL) was added. The reaction mixture was cooled to 0 °C and chloroacetyl chloride was added dropwise (613 μL, 7.7 mmol) then stirring was continued at room temperature. After 1 h, the resulting mixture was diluted with dichloromethane (90 mL) and washed with brine (2 × 60 mL). The combined organic layer was dried over anhydrous MgSO_4_, concentrated under vacuum and purified by column chromatography (toluene:AcOEt; gradient 8:1 to 2:1) to give products **31**–**32**.

2,3,4,6-Tetra-*O*-acetyl-*N*-(β-D-glucopyranosyl)chloroacetamide **31**: Starting from 2,3,4,6-tetra-*O*-acetyl-β-D-glucopyranosyl amine **19**, product was obtained as a white solid (1.41 g, 68%); m.p.: 160–163 °C; [α]^24^_D_ = 8.3 (c = 1.0, CHCl_3_); ^1^H NMR (400 MHz, CDCl_3_): δ 2.03, 2.04, 2.06, 2.09 (4s, 12H, CH_3_CO), 3.84 (ddd, 1H, *J* = 2.1 Hz, *J* = 4.4 Hz, *J* = 10.1 Hz, H-5_glu_), 4.00 and 4.07 (qAB, 2H, *J* = 15.4 Hz, CH_2_Cl), 4.10 (dd, 1H, *J* = 2.1 Hz, *J* = 12.5 Hz, H-6a_glu_), 4.31 (dd, 1H, *J* = 4.4 Hz, *J* = 12.5 Hz, H-6b_glu_), 5.01 (dd-t, 1H, *J* = 9.4 Hz, *J* = 10.0 Hz, H-1_glu_), 5.09 (dd-t, 1H, *J* = 9.4 Hz, *J* = 10.1 Hz, H-4_glu_), 5.21 (dd-t, 1H, *J* = 9.0 Hz, *J* = 9.4 Hz, H-3_glu_), 5.33 (dd-t, 1H, *J* = 9.0 Hz, *J* = 9.4 Hz, H-2_glu_), 7.29 (d, 1H, *J* = 9.0 Hz, CONH); ^13^C NMR (100 MHz, CDCl_3_): δ 20.57, 20.61, 20.72, 20.76, 42.25, 61.56, 68.09, 70.26, 72.55, 73.83, 78.54, 166.81, 169.49, 169.87, 170.58, 170.79; HRMS (ESI-TOF): calcd for C_16_H_22_ClNO_10_Na ([M + Na]^+^): *m*/*z* 446.0830; found: *m*/*z* 446.0834.

2,3,4,6-Tetra-*O*-acetyl-*N*-(β-D-galactopyranosyl)chloroacetamide **32**: Starting from 2,3,4,6-tetra-*O*-acetyl-β-D-galactopyranosyl amine **20**, product was obtained as a white solid (1.66 g, 80%); m.p.: 143-144 °C; [α]^25^_D_ = 21.3 (c = 1.0, CHCl_3_); ^1^H NMR (400 MHz, CDCl_3_): δ 2.01, 2.04, 2.07, 2.16 (4s, 12H, CH_3_CO), 4.01 and 4.08 (qAB, 2H, *J* = 15.5 Hz, CH_2_Cl), 4.04-4.18 (m, 3H, H-5_gal_, H-6a_gal_, H-6b_gal_), 5.11-5.24 (m, 3H, H-1_gal_, H-2_gal_, H-3_gal_), 5.45 (dd, 1H, *J* = 0. 7 Hz, *J* = 3.0 Hz, H-4_gal_), 7.32 (d, 1H, *J* = 6.9 Hz, CONH); ^13^C NMR (100 MHz, CDCl_3_): δ 20.54, 20.61, 20.67, 20.71, 42.26, 61.12, 67.10, 67.97, 70.72, 72.58, 78.84, 166.71, 169.76, 170.01, 170.34, 171.06; HRMS (ESI-TOF): calcd for C_16_H_22_ClNO_10_Na ([M + Na]^+^): *m*/*z* 446.0830; found: *m*/*z* 446.0832.

#### 3.2.3. General Procedure for the Synthesis of Sugar Derivatives **33** and **34**

To a solution of 2,3,4,6-tetra-*O*-acetyl-*N*-(β-D-glucopyranosyl)chloroacetamide **31** or 2,3,4,6-tetra-*O*-acetyl-*N*-(β-D-galactopyranosyl)chloroacetamide **32** (1.0 g, 2.36 mmol) in dry DMF (15 mL), sodium azide (753 mg, 11.59 mmol) was added. The reaction mixture was stirred at room temperature for 24 h. After completion reaction, the solvent was evaporated under reduced pressure, and the residue was diluted with ethyl acetate (50 mL) and extracted with water (30 mL), 0.25 M HCl (20 mL) and brine (20 mL). The combined organic layer was dried over anhydrous MgSO_4_, concentrated under vacuum, to afford the corresponding azide **33** and **34**, which were used for the next reaction without further purification.

2,3,4,6-tetra-*O*-acetyl-*N*-(β-D-glucopyranosyl)azidoacetamide **33**: Starting from 2,3,4,6-tetra-*O*-acetyl-*N*-(β-D-glucopyranosyl)chloroacetamide **31**, product was obtained as a white solid (975.0 mg, 96%); m.p.: 150-153 °C; [α]^24^_D_ = 12.4 (c = 1.0, CHCl_3_); ^1^H NMR (400 MHz, CDCl_3_): δ 2.03, 2.04, 2.06, 2.09 (4s, 12H, CH_3_CO), 3.83 (ddd, 1H, *J* = 2.1 Hz, *J* = 4.4 Hz, *J* = 10.1 Hz, H-5_glu_), 3.95 and 4.00 (qAB, 2H, *J* = 16.9 Hz, CH_2_N), 4.09 (dd, 1H, *J* = 2.1 Hz, *J* = 12.5 Hz, H-6a_glu_), 4.30 (dd, 1H, *J* = 4.4 Hz, *J* = 12.5 Hz, H-6b_glu_), 4.98 (dd-t, 1H, *J* = 9.4 Hz, *J* = 9.8 Hz, H-1_glu_), 5.08 (dd-t, 1H, *J* = 9.4 Hz, *J* = 10.1 Hz, H-4_glu_), 5.22 (dd-t, 1H, *J* = 9.0 Hz, *J* = 9.4 Hz, H-3_glu_), 5.32 (dd-t, 1H, *J* = 9.0 Hz, *J* = 9.8 Hz, H-2_glu_), 7.10 (d, 1H, *J* = 9.1 Hz, CONH); ^13^C NMR (100 MHz, CDCl_3_): δ 20.57, 20.61, 20.72, 20.75, 52.59, 61.56, 68.08, 70.45, 72.57, 73.78, 78.15, 167.41, 169.50, 169.86, 170.57, 170.88; HRMS (ESI-TOF): calcd for C_16_H_22_N_4_O_10_Na ([M + Na]^+^): *m*/*z* 453.1234; found: *m*/*z* 453.1236.

2,3,4,6-tetra-*O*-acetyl-*N*-(β-D-galactopyranosyl)azidoacetamide **34**: Starting from 2,3,4,6-tetra-*O*-acetyl-*N*-(β-D-galactopyranosyl)chloroacetamide **32**, product was obtained as a white solid (873.5 mg, 86%); m.p.: 60–63 °C; [α]^24^_D_ = 25.4 (c = 1.0, CHCl_3_); ^1^H NMR (400 MHz, CDCl_3_): δ 2.01, 2.04, 2.07, 2.16 (4s, 12H, CH_3_CO), 3.96 and 4.01 (qAB, 2H, *J* = 16.8 Hz, CH_2_N), 4.02–4.17 (m, 3H, H-5_gal_, H-6a_gal_, H-6b_gal_), 5.11–5.25 (m, 3H, H-1_gal_, H-2_gal_, H-3_gal_), 5.45 (dd, 1H, *J* = 1.1 Hz, *J* = 2.9 Hz, H-4_gal_), 7.13 (d, 1H, *J* = 8.5 Hz, CONH); ^13^C NMR (100 MHz, CDCl_3_): δ 20.54, 20.60, 20.67, 20.70, 52.61, 61.14, 67.11, 68.17, 70.73, 72.52, 78.43, 167.31, 169.76, 170.00, 170.35, 171.15; HRMS (ESI-TOF): calcd for C_16_H_22_N_4_O_10_Na ([M + Na]^+^): *m*/*z* 453.1234; found: *m*/*z* 453.1227.

#### 3.2.4. Synthesis of Glycoconjugates **45**–**108**

The appropriate derivatives of 8-hydoxyquinoline **3–10** (0.5 mmol) and sugar derivatives **21**–**24**, **27**–**30**, **33**–**44** (0.5 mmol) were dissolved in dry THF (5 mL) and *i*-PrOH (5 mL). To the obtained solution, CuSO_4_·5H_2_O (0.1 mmol, 25.0 mg) dissolved in H_2_O (2.5 mL) and sodium ascorbate (0.2 mmol, 39.6 mg) dissolved in H_2_O (2.5 mL) were added. The reaction mixture was stirred for 24 h at room temperature. The progress of the reaction was monitored on TLC in an CHCl_3_:CH_3_OH eluents system (20:1 for protected or 2:1 for unprotected compounds). After completion, the reaction mixture was concentrated in vacuo and purified using column chromatography (dry loading; toluene:AcOEt, 2:1 and CHCl_3_:MeOH, 100:1 for fully protected glycoconjugates or CHCl_3_:MeOH, gradient: 50:1 to 2:1 for glycoconjugates with unprotected sugar part) to give products **45–108**.

Glycoconjugate **45**: Starting from propargyl 2,3,4,6-tetra-*O*-acetyl-β-D-glucopyranoside **21** and 8-(2-azidoethoxy)quinoline **5**, product was obtained as a yellow solid (219.2 mg, 73%); m.p.: 130–133 °C; [α]^24^_D_ = −36.0 (c = 1.0, CHCl_3_); ^1^H NMR (400 MHz, CDCl_3_): δ 1.84, 1.97, 2.01, 2.07 (4s, 12H, CH_3_CO), 3.66 (ddd, 1H, *J* = 2.3 Hz, *J* = 4.6 Hz, *J* = 9.8 Hz, H-5_glu_), 4.10 (dd, 1H, *J* = 2.3 Hz, *J* = 12.3 Hz, H-6a_glu_), 4.21 (dd, 1H, *J* = 4.6 Hz, *J* = 12.3 Hz, H-6b_glu_), 4.63 (d, 1H, *J* = 8.0 Hz, H-1_glu_), 4.59–4.70 (m, 2H, CH_2_N), 4.80 and 4.88 (qAB, 2H, *J* = 12.7 Hz, CH_2_C), 4.95 (dd, 1H, *J* = 8.0 Hz, *J* = 9.4 Hz H-2_glu_), 5.01 (t, 2H, *J* = 5.2 Hz, CH_2_O), 5.04 (dd-t, 1H, *J* = 9.4 Hz, *J* = 9.8 Hz, H-4_glu_), 5.13 (dd-t, 1H, *J* = 9.4 Hz, *J* = 9.4 Hz, H-3_glu_), 7.05 (dd, 1H, *J* = 2.1 Hz, *J* = 6.4 Hz, H-7_chin_), 7.42–7.55 (m, 3H, H-3_chin_, H-5_chin_, H-6_chin_), 8.21 (d, 1H, *J* = 8.1 Hz, H-4_chin_), 8.27 (s, 1H, H-5_triaz_), 8.99 (d, 1H, *J* = 2.9 Hz, H-2_chin_); ^13^C NMR (100 MHz, CDCl_3_): δ 20.47, 20.48, 20.59, 20.75, 49.75, 61.84, 62.40, 67.89, 68.34, 71.17, 71.82, 72.86, 99.32, 110.31, 121.07, 121.95, 125.10, 126.87, 129.65, 136.68, 139.55, 143.93, 149.26, 153.45, 169.36, 169.40, 170.18, 170.66; HRMS (ESI-TOF): calcd for C_28_H_33_N_4_O_11_ ([M + H]^+^): *m*/*z* 601.2146; found: *m*/*z* 601.2148.

Glycoconjugate **46:** Starting from propargyl 2,3,4,6-tetra-*O*-acetyl-β-D-galactopyranoside **22** and 8-(2-azidoethoxy)quinoline **5**, product was obtained as a yellow solid (222.2 mg, 74%); m.p.: 69–72 °C; [α]^24^_D_ = −30.0 (c = 1.0, CHCl_3_); ^1^H NMR (400 MHz, CDCl_3_): δ 1.85, 1.95, 2.05, 2.13 (4s, 12H, CH_3_CO), 3.83-3.92 (m, 1H, H-5_gal_), 4.10-4.16 (m, 2H, H-6a_gal_, H-6b_gal_), 4.60 (d, 1H, *J* = 8.0 Hz, H-1_gal_), 4.62–4.69 (m, 2H, CH_2_N), 4.79 and 4.90 (qAB, 2H, *J* = 12.6 Hz, CH_2_C), 4.94 (dd, 1H, *J* = 3.4 Hz, *J* = 10.4 Hz H-3_gal_), 4.97-5.05 (m, 2H, CH_2_O), 5.17 (dd, 1H, *J* = 8.0 Hz, *J* = 10.4 Hz, H-2_gal_), 5.35 (dd, 1H, *J* = 0.7 Hz, *J* = 3.4 Hz, H-4_gal_), 6.99–7.12 (m, 1H, H-7_chin_), 7.38–7.58 (m, 3H, H-3_chin_, H-5_chin_, H-6_chin_), 8.12–8.34 (m, 2H, H-4_chin_, H-5_triaz_), 8.99 (bs, 1H, H-2_chin_); ^13^C NMR (100 MHz, DMSO-d6): δ 20.26, 20.28, 20.35, 20.47, 49.18, 61.17, 61.76, 67.25, 67.43, 68.53, 69.91, 70.19, 98.99, 110.74, 120.57, 121.98, 124.93, 126.84, 129.15, 136.26, 139.30, 143.07, 149.00, 153.50, 169.03, 169.41, 169.85, 169.88; HRMS (ESI-TOF): calcd for C_28_H_33_N_4_O_11_ ([M + H]^+^): *m*/*z* 601.2146; found: *m*/*z* 601.2145.

Glycoconjugate **47:** Starting from propargyl 2,3,4,6-tetra-*O*-acetyl-β-D-glucopyranoside **21** and 2-methyl-8-(2-azidoethoxy)quinoline **8**, product was obtained as a yellow solid (307.3 mg, 100%); m.p.: 119-123 °C; [α]^24^_D_ = −32.4 (c = 1.0, CHCl_3_); ^1^H NMR (400 MHz, DMSO-d6): δ 1.83, 1.91, 1.98, 2.02 (4s, 12H, CH_3_CO), 2.69 (s, 3H, CH_3_), 3.93 (ddd, 1H, *J* = 2.4 Hz, *J* = 4.9 Hz, *J* = 10.0 Hz, H-5_glu_), 4.01 (dd, 1H, *J* = 2.4 Hz, *J* = 12.3 Hz, H-6a_glu_), 4.15 (dd, 1H, *J* = 4.9 Hz, *J* = 12.3 Hz, H-6b_glu_), 4.59 (t, 2H, *J* = 4.9 Hz, CH_2_N), 4.65 and 4.81 (qAB, 2H, *J* = 12.3 Hz, CH_2_C), 4.74 (dd, 1H, *J* = 8.0 Hz, *J* = 9.6 Hz H-2_glu_), 4.87 (d, 1H, *J* = 8.0 Hz, H-1_glu_), 4.86–4.95 (m, 3H, CH_2_O, H-4_glu_), 5.20 (dd-t, 1H, *J* = 9.6 Hz, *J* = 9.6 Hz, H-3_glu_), 7.20 (dd, 1H, *J* = 1.2 Hz, *J* = 7.7 Hz, H-7_chin_), 7.39-7.46 (m, 2H, H-3_chin_, H-6_chin_), 7.50 (dd, 1H, *J* = 1.2 Hz, *J* = 8.2 Hz, H-5_chin_), 8.20 (d, 1H, *J* = 8.4 Hz, H-4_chin_), 8.59 (s, 1H, H-5_triaz_); ^13^C NMR (100 MHz, DMSO-d6): δ 20.15, 20.22, 20.34, 20.46, 25.01, 49.18, 61.60, 61.90, 67.58, 68.04, 70.58, 70.80, 72.05, 98.51, 111.21, 120.48, 122.52, 125.45, 125.65, 127.35, 136.04, 139.29, 142.89, 153.20, 157.55, 168.88, 169.20, 169.45, 169.99; HRMS (ESI-TOF): calcd for C_29_H_35_N_4_O_11_ ([M + H]^+^): *m*/*z* 615.2302; found: *m*/*z* 615.2304.

Glycoconjugate **48:** Starting from propargyl 2,3,4,6-tetra-*O*-acetyl-β-D-galactopyranoside **22** and 2-methyl-8-(2-azidoethoxy)quinoline **8**, product was obtained as a yellow solid (218.2 mg, 71%); m.p.: 52–56 °C; [α]^23^_D_ = −20.8 (c = 1.0, CHCl_3_); ^1^H NMR (400 MHz, DMSO-d6): δ 1.83, 1.89, 2.01, 2.11 (4s, 12H, CH_3_CO), 2.69 (s, 3H, CH_3_), 3.99–4.10 (m, 2H, H-5_gal_, H-6a_gal_), 4.13–4.21 (m, 1H, H-6b_gal_), 4.59 (t, 2H, *J* = 5.0 Hz, CH_2_N), 4.64 and 4.81 (qAB, 2H, *J* = 12.3 Hz, CH_2_C), 4.79 (d, 1H, *J* = 8.0 Hz, H-1_gal_), 4.86–4.97 (m, 3H, CH_2_O, H-2_gal_), 5.11 (dd, 1H, *J* = 0.9 Hz, *J* = 10.3 Hz, H-4_gal_), 5.24 (dd, 1H, *J* = 3.6 Hz, *J* = 10.3 Hz, H-3_gal_), 7.20 (dd, 1H, *J* = 1.2 Hz, *J* = 7.7 Hz, H-7_chin_), 7.38-7.47 (m, 2H, H-3_chin_, H-6_chin_), 7.50 (dd, 1H, *J* = 1.2 Hz, *J* = 8.2 Hz, H-5_chin_), 8.21 (d, 1H, *J* = 8.4 Hz, H-4_chin_), 8.60 (s, 1H, H-5_triaz_); ^13^C NMR (100 MHz, DMSO-d6): δ 20.25, 20.28, 20.35, 20.47, 25.01, 49.19, 61.19, 61.76, 67.26, 67.59, 68.53, 69.89, 70.20, 98.94, 111.22, 120.48, 122.52, 125.40, 125.66, 127.35, 136.04, 139.31, 142.97, 153.21, 157.56, 168.98, 169.40, 169.85, 169.89; HRMS (ESI-TOF): calcd for C_29_H_35_N_4_O_11_ ([M + H]^+^): *m*/*z* 615.2302; found: *m*/*z* 615.2303.

Glycoconjugate **49:** Starting from propargyl β-D-glucopyranoside **23** and 8-(2-azidoethoxy)quinoline **5**, product was obtained as a yellow solid (205.4 mg, 95%); m.p.: 42-45 °C; [α]^25^_D_ = −17.2 (c = 1.0, MeOH); ^1^H NMR (400 MHz, DMSO-d6): δ 2.92-3.09 (m, 2H, H-2_glu_, H-5_glu_), 3.10-3.17 (m, 2H, H-3_glu_, H-4_glu_), 3.40–3.50 (m, 1H, H-6a_glu_), 3.66–3.75 (m, 1H, H-6b_glu_), 4.08 (bs, 1H, OH), 4.27 (d, 1H, *J* = 7.8 Hz, H-1_glu_), 4.57 (bs, 1H, OH), 4.60–4.66 (m, 3H, CH_2_N, CH_2_C), 4.83–4.95 (m, 4H, CH_2_O, CH_2_C, OH), 5.02 (bs, 1H, OH), 7.25 (dd, 1H, *J* = 1.5 Hz, *J* = 7.5 Hz, H-7_chin_), 7.47–7.58 (m, 3H, H-3_chin_, H-5_chin_, H-6_chin_), 8.32 (dd, 1H, *J* = 1.7 Hz, *J* = 8.3 Hz, H-4_chin_), 8.40 (s, 1H, H-5_triaz_), 8.89 (dd, 1H, *J* = 1.7 Hz, *J* = 4.2 Hz, H-2_chin_); ^13^C NMR (100 MHz, DMSO-d6): δ 49.12, 61.15, 61.47, 67.36, 70.10, 73.37, 76.70, 76.94, 102.14, 110.57, 120.51, 121.91, 124.96, 126.71, 129.08, 135.85, 139.69, 143.78, 149.27, 153.68; HRMS (ESI-TOF): calcd for C_20_H_25_N_4_O_7_ ([M + H]^+^): *m*/*z* 433.1723; found: *m*/*z* 433.1723.

Glycoconjugate **50:** Starting from propargyl β-D-galactopyranoside **24** and 8-(2-azidoethoxy)quinoline **5**, product was obtained as a yellow solid (144.9 mg, 67%); m.p.: 44–45 °C; [α]^23^_D_ = −12.9 (c = 1.0, MeOH); ^1^H NMR (400 MHz, DMSO-d6): δ 3.21–3.39 (m, 3H, H-2_gal_, H-3_gal_, H-4_gal_), 3.40–3.48 (m, 1H, H-5_gal_), 3.50–3.57 (m, 1H, H-6a_gal_), 3.59–3.66 (m, 1H, H-6b_gal_), 4.21 (d, 1H, *J* = 7.4 Hz, H-1_gal_), 4.61 and 4.83 (qAB, 2H, *J* = 12.2 Hz, CH_2_C), 4.63 (t, 2H, *J* = 5.2 Hz, CH_2_N), 4.90 (t, 2H, *J* = 5.2 Hz, CH_2_O), 7.25 (dd, 1H, *J* = 1.5 Hz, *J* = 7.5 Hz, H-7_chin_), 7.47–7.58 (m, 3H, H-3_chin_, H-5_chin_, H-6_chin_), 8.32 (dd, 1H, *J* = 1.7 Hz, *J* = 8.3 Hz, H-4_chin_), 8.40 (s, 1H, H-5_triaz_), 8.89 (dd, 1H, *J* = 1.7 Hz, *J* = 4.2 Hz, H-2_chin_); ^13^C NMR (100 MHz, DMSO-d6): δ 49.11, 60.52, 61.31, 67.37, 68.18, 70.44, 73.40, 75.32, 102.69, 110.58, 120.51, 121.92, 124.91, 126.70, 129.07, 135.83, 139.70, 143.85, 149.30, 153.69; HRMS (ESI-TOF): calcd for C_20_H_25_N_4_O_7_ ([M + H]^+^): *m*/*z* 433.1723; found: *m*/*z* 433.1725.

Glycoconjugate **51:** Starting from propargyl β-D-glucopyranoside **23** and 2-methyl-8-(2-azidoethoxy)quinoline **8**, product was obtained as a yellow solid (196.4 mg, 88%); m.p.: 120–121 °C; [α]^24^_D_ = −20.0 (c = 1.0, MeOH); ^1^H NMR (400 MHz, DMSO-d6): δ 2.69 (s, 3H, CH_3_), 2.92–3.08 (m, 2H, H-2_glu_, H-5_glu_), 3.09-3.19 (m, 2H, H-3_glu_, H-4_glu_), 3.40–3.49 (m, 1H, H-6a_glu_), 3.66–3.73 (m, 1H, H-6b_glu_), 4.09 (bs, 1H, OH), 4.27 (d, 1H, *J* = 7.8 Hz, H-1_glu_), 4.33 (bs, 1H, OH), 4.53 (t, 1H, *J* = 5.6 Hz, OH), 4.60 (t, 2H, *J* = 5.2 Hz, CH_2_N), 4.63 and 4.87 (qAB, 2H, *J* = 12.1 Hz, CH_2_C), 4.89 (t, 2H, *J* = 5.2 Hz, CH_2_O), 4.99 (d, 1H, *J* = 4.9 Hz, OH), 7.21 (dd, 1H, *J* = 1.3 Hz, *J* = 7.7 Hz, H-7_chin_), 7.39–7.45 (m, 2H, H-3_chin_, H-6_chin_), 7.50 (dd, 1H, *J* = 1.2 Hz, *J* = 8.2 Hz, H-5_chin_), 8.20 (d, 1H, *J* = 8.4 Hz, H-4_chin_), 8.54 (s, 1H, H-5_triaz_); ^13^C NMR (100 MHz, DMSO-d6): δ 25.11, 49.11, 61.14, 61.48, 67.56, 70.08, 73.35, 76.70, 76.93, 102.19, 111.16, 120.45, 122.54, 125.21, 125.66, 127.35, 136.01, 139.28 143.77, 153.21, 157.61; HRMS (ESI-TOF): calcd for C_21_H_27_N_4_O_7_ ([M + H]^+^): *m*/*z* 447.1880; found: *m*/*z* 447.1882.

Glycoconjugate **52:** Starting from propargyl β-D-galactopyranoside **24** and 2-methyl-8-(2-azidoethoxy)quinoline **8**, product was obtained as a yellow solid (174.1 mg, 78%); m.p.: 52–54 °C; [α]^24^_D_ = −15.0 (c = 1.0, MeOH); ^1^H NMR (400 MHz, DMSO-d6): δ 2.69 (s, 3H, CH_3_), 3.21–3.38 (m, 4H, H-2_gal_, H-3_gal_, H-4_gal_, H-5_gal_), 3.49–3.57 (m, 1H, H-6a_gal_), 3.60–3.66 (m, 1H, H-6b_gal_), 4.22 (d, 1H, *J* = 7.3 Hz, H-1_gal_), 4.32 (d, 1H, *J* = 4.5 Hz, OH), 4.58 (t, 1H, *J* = 5.6 Hz, OH), 4.60 (t, 2H, *J* = 5.2 Hz, CH_2_N), 4.60 and 4.87 (qAB, 2H, *J* = 12.0 Hz, CH_2_C), 4.66 (d, 1H, *J* = 5.2 Hz, OH), 4.83 (t, 1H, *J* = 2.2 Hz, OH), 4.89 (t, 2H, *J* = 5.2 Hz, CH_2_O), 7.21 (dd, 1H, *J* = 1.3 Hz, *J* = 7.7 Hz, H-7_chin_), 7.38–7.46 (m, 2H, H-3_chin_, H-6_chin_), 7.50 (dd, 1H, *J* = 1.2 Hz, *J* = 8.2 Hz, H-5_chin_), 8.20 (d, 1H, *J* = 8.4 Hz, H-4_chin_), 8.54 (s, 1H, H-5_triaz_); ^13^C NMR (100 MHz, DMSO-d6): δ 25.10, 49.11, 60.45, 61.28, 67.57, 68.13, 70.41, 73.42, 75.28, 102.72, 111.18, 120.45, 122.53, 125.19, 125.66, 127.35, 136.01, 139.28, 143.79, 153.21, 157.62; HRMS (ESI-TOF): calcd for C_21_H_27_N_4_O_7_ ([M + H]^+^): *m*/*z* 447.1880; found: *m*/*z* 447.1879.

Glycoconjugate **53:** Starting from propargyl 2,3,4,6-tetra-*O*-acetyl-β-D-glucopyranoside **21** and 8-(3-azidopropoxy)quinoline **6**, product was obtained as a yellow oil (307.3 mg, 100%); [α]^25^_D_ = −21.4 (c = 1.0, CHCl_3_); ^1^H NMR (400 MHz, DMSO-d6): δ 1.88, 1.92, 1.98, 2.03 (4s, 12H, CH_3_CO), 2.43 (p, 2H, *J* = 6.5 Hz, CH_2_), 3.99 (ddd, 1H, *J* = 2.4 Hz, *J* = 4.9 Hz, *J* = 10.0 Hz, H-5_glu_), 4.10 (dd, 1H, *J* = 2.4 Hz, *J* = 12.3 Hz, H-6a_glu_), 4.15–4.24 (m, 3H, CH_2_N, H-6b_glu_), 4.66 (t, 2H, *J* = 6.9 Hz, CH_2_O), 4.65 and 4.80 (qAB, 2H, *J* = 12.1 Hz, CH_2_C), 4.76 (dd, 1H, *J* = 8.0 Hz, *J* = 9.6 Hz H-2_glu_), 4.88 (d, 1H, *J* = 8.0 Hz, H-1_glu_), 4.91 (dd-t, 1H, *J* = 9.6 Hz, *J* = 9.7 Hz, H-4_glu_), 5.25 (dd-t, 1H, *J* = 9.6 Hz, *J* = 9.6 Hz, H-3_glu_), 7.19 (dd, 1H, *J* = 1.9 Hz, *J* = 7.1 Hz, H-7_chin_), 7.47–7.53 (m, 2H, H-3_chin_, H-6_chin_), 7.56 (dd, 1H, *J* = 4.0 Hz, *J* = 8.2 Hz, H-5_chin_), 8.24 (s, 1H, H-5_triaz_), 8.33 (dd, 1H, *J* = 1.6 Hz, *J* = 8.2 Hz, H-4_chin_), 8.89 (bs, 1H, H-2_chin_); ^13^C NMR (100 MHz, DMSO-d6): δ 20.23, 20.25, 20.35, 20.47, 29.57, 46.66, 61.64, 61.85, 65.39, 68.11, 70.61, 70.83, 72.03, 98.47, 109.82, 119.93, 121.85, 124.50, 126.77, 129.03, 135.78, 139.79, 142.94, 149.01, 154.20, 168.92, 169.22, 169.48, 170.01; HRMS (ESI-TOF): calcd for C_29_H_35_N_4_O_11_ ([M + H]^+^): *m*/*z* 615.2302; found: *m*/*z* 615.2303.

Glycoconjugate **54:** Starting from propargyl 2,3,4,6-tetra-*O*-acetyl-β-D-galactopyranoside **22** and 8-(3-azidopropoxy)quinoline **6**, product was obtained as a yellow oil (307.3 mg, 100%); [α]^25^_D_ = −16.2 (c = 1.0, CHCl_3_); ^1^H NMR (400 MHz, DMSO-d6): δ 1.89, 1.90, 2.01, 2.11 (4s, 12H, CH_3_CO), 2.43 (p, 2H, *J* = 6.4 Hz, CH_2_), 4.01–4.11 (m, 2H, H-5_gal_, H-6a_gal_), 4.15–4.24 (m, 3H, H-6b_gal_, CH_2_N), 4.64 and 4.79 (qAB, 2H, *J* = 12.4 Hz, CH_2_C), 4.66 (t, 2H, *J* = 6.8 Hz, CH_2_O), 4.79 (d, 1H, *J* = 8.0 Hz, H-1_gal_), 4.92 (dd, 1H, *J* = 8.0 Hz, *J* = 10.3 Hz, H-2_gal_), 5.15 (dd, 1H, *J* = 3.6 Hz, *J* = 10.3 Hz, H-3_gal_), 5.25 (dd, 1H, *J* = 0.9 Hz, *J* = 3.6 Hz, H-4_gal_), 7.20 (dd, 1H, *J* = 1.5 Hz, *J* = 7.4 Hz, H-7_chin_), 7.46–7.61 (m, 3H, H-3_chin_, H-5_chin_, H-6_chin_), 8.24 (s, 1H, H-5_triaz_), 8.31–8.35 (m, 1H, H-4_chin_), 8.90 (bs, 1H, H-2_chin_); ^13^C NMR (100 MHz, DMSO-d6): δ 20.28, 20.29, 20.35, 20.47, 29.58, 46.66, 61.22, 61.72, 65.39, 67.28, 68.57, 69.91, 70.19, 98.89, 109.81, 119.94, 124.47, 125.27, 126.76, 128.16, 128.86, 135.77, 143.00, 148.99, 154.22, 169.03, 169.42, 169.86, 169.89; HRMS (ESI-TOF): calcd for C_29_H_35_N_4_O_11_ ([M + H]^+^): *m*/*z* 615.2302; found: *m*/*z* 615.2302.

Glycoconjugate **55:** Starting from propargyl 2,3,4,6-tetra-*O*-acetyl-β-D-glucopyranoside **21** and 2-methyl-8-(3-azidopropoxy)quinoline **9**, product was obtained as a yellow oil (204.3 mg, 65%); [α]^23^_D_ = −20.2 (c = 1.0, CHCl_3_); ^1^H NMR (400 MHz, DMSO-d6): δ 1.88, 1.92, 1.98, 2.03 (4s, 12H, CH_3_CO), 2.42 (p, 2H, *J* = 6.5 Hz, CH_2_), 2.67 (s, 3H, CH_3_), 3.98 (ddd, 1H, *J* = 2.4 Hz, *J* = 4.9 Hz, *J* = 10.0 Hz, H-5_glu_), 4.04 (dd, 1H, *J* = 2.4 Hz, *J* = 12.4 Hz, H-6a_glu_), 4.15–4.24 (m, 3H, CH_2_N, H-6b_glu_), 4.64 and 4.80 (qAB, 2H, *J* = 12.3 Hz, CH_2_C), 4.65 (t, 2H, *J* = 6.9 Hz, CH_2_O), 4.75 (dd, 1H, *J* = 8.0 Hz, *J* = 9.6 Hz H-2_glu_), 4.87 (d, 1H, *J* = 8.0 Hz, H-1_glu_), 4.91 (dd-t, 1H, *J* = 9.6 Hz, *J* = 9.8 Hz, H-4_glu_), 5.24 (dd-t, 1H, *J* = 9.6 Hz, *J* = 9.6 Hz, H-3_glu_), 7.16 (dd, 1H, *J* = 1.2 Hz, *J* = 7.6 Hz, H-7_chin_), 7.38–7.50 (m, 3H, H-3_chin_, H-5_chin_, H-6_chin_), 8.19 (d, 1H, *J* = 8.4 Hz, H-4_chin_), 8.25 (s, 1H, H-5_triaz_); ^13^C NMR (100 MHz, DMSO-d6): δ 20.23, 20.24, 20.35, 20.48, 25.05, 29.52, 46.64, 61.64, 61.83, 65.57, 68.11, 70.60, 70.83, 72.03, 98.46, 110.37, 119.86, 122.44, 124.47, 125.70, 127.32, 135.98, 139.32, 142.93, 153.67, 157.30, 168.92, 169.23, 169.48, 170.01; HRMS (ESI-TOF): calcd for C_30_H_37_N_4_O_11_ ([M + H]^+^): *m*/*z* 629.2459; found: *m*/*z* 629.2458.

Glycoconjugate **56:** Starting from propargyl 2,3,4,6-tetra-*O*-acetyl-β-D-galactopyranoside **22** and 2-methyl-8-(3-azidopropoxy)quinoline **9**, product was obtained as a yellow oil (242.0 mg, 77%); [α]^24^_D_ = −8.8 (c = 1.0, CHCl_3_); ^1^H NMR (400 MHz, DMSO-d6): δ 1.89, 1.90, 2.01, 2.11 (4s, 12H, CH_3_CO), 2.42 (p, 2H, *J* = 6.3 Hz, CH_2_), 2.67 (s, 3H, CH_3_), 4.01–4.11 (m, 2H, H-5_gal_, H-6a_gal_), 4.15–4.24 (m, 3H, H-6a_gal_, CH_2_N), 4.64 and 4.79 (qAB, 2H, *J* = 12.6 Hz, CH_2_C), 4.66 (t, 2H, *J* = 6.9 Hz, CH_2_O), 4.78 (d, 1H, *J* = 8.0 Hz, H-1_gal_), 4.92 (dd, 1H, *J* = 8.0 Hz, *J* = 10.4 Hz, H-2_gal_), 5.14 (dd, 1H, *J* = 3.6 Hz, *J* = 10.4 Hz, H-3_gal_), 5.25 (dd, 1H, *J* = 0.9 Hz, *J* = 3.6 Hz, H-4_gal_), 7.16 (dd, 1H, *J* = 0.8 Hz, *J* = 7.5 Hz, H-7_chin_), 7.37–7.51 (m, 3H, H-3_chin_, H-5_chin_, H-6_chin_), 8.19 (d, 1H, *J* = 8.4 Hz, H-4_chin_), 8.25 (s, 1H, H-5_triaz_); ^13^C NMR (100 MHz, DMSO-d6): δ18.53, 20.28, 20.29, 20.35, 20.48, 29.53, 46.64, 61.23, 61.70, 65.55, 68.57, 69.91, 70.19, 72.20, 98.88, 110.33, 119.87, 122.45, 124.46, 125.71, 127.31, 135.96, 139.32, 143.02, 153.26, 158.61, 169.03, 169.43, 169.87, 169.90; HRMS (ESI-TOF): calcd for C_30_H_37_N_4_O_11_ ([M + H]^+^): *m*/*z* 629.2459; found: *m*/*z* 629.2455. 

Glycoconjugate **57:** Starting from propargyl β-D-glucopyranoside **23** and 8-(3-azidopropoxy)quinoline **6**, product was obtained as a white solid (223.2 mg, 100%); m.p.: 77–79 °C; [α]^24^_D_ = −19.2 (c = 1.0, MeOH); ^1^H NMR (400 MHz, DMSO-d6): δ 2.42 (p, 2H, *J* = 6.4 Hz, CH_2_), 2.94-3.09 (m, 2H, H-2_glu_, H-5_glu_), 3.10-3.17 (m, 2H, H-3_glu_, H-4_glu_), 3.40–3.50 (m, 1H, H-6a_glu_), 3.67–3.75 (m, 1H, H-6b_glu_), 4.05–4.11 (m, 1H, OH), 4.21 (t, 2H, *J* = 6.1 Hz, CH_2_N), 4.27 (d, 1H, *J* = 7.8 Hz, H-1_glu_), 4.34 (t, 1H, *J* = 5.1 Hz, OH), 4.57 (t, 1H, *J* = 5.9 Hz, OH), 4.64 (t, 1H, *J* = 7.1 Hz, CH_2_O), 4.64 and 4.85 (qAB, 2H, J = 12.2 Hz, CH_2_C), 5.01 (d, 1H, *J* = 4.9 Hz, OH), 7.20 (dd, 1H, *J* = 2.2 Hz, *J* = 6.8 Hz, H-7_chin_), 7.47–7.58 (m, 3H, H-3_chin_, H-5_chin_, H-6_chin_), 8.29 (s, 1H, H-5_triaz_), 8.32 (dd, 1H, *J* = 1.7 Hz, *J* = 8.3 Hz, H-4_chin_), 8.90 (dd, 1H, *J* = 1.6 Hz, *J* = 4.1 Hz, H-2_chin_); ^13^C NMR (100 MHz, DMSO-d6): δ 29.61, 46.61, 61.16, 61.57, 65.43, 70.11, 73.39, 76.69, 76.94, 102.14, 109.84, 119.92, 121.86, 124.42, 126.79, 129.04, 135.81, 139.76, 143.89, 149.07, 154.19; HRMS (ESI-TOF): calcd for C_21_H_27_N_4_O_7_ ([M + H]^+^): *m*/*z* 447.1880; found: *m*/*z* 447.1880.

Glycoconjugate **58:** Starting from propargyl β-D-galactopyranoside **24** and 8-(3-azidopropoxy)quinoline **6**, product was obtained as a yellow solid (169.7 mg, 76%); m.p.: 50–54 °C; [α]^24^_D_ = −11.2 (c = 1.0, MeOH); ^1^H NMR (400 MHz, DMSO-d6): δ 2.42 (p, 2H, *J* = 6.5 Hz, CH_2_), 3.22-3.40 (m, 4H, H-2_gal_, H-3_gal_, H-4_gal_, H-5_gal_), 3.50-3.58 (m, 1H, H-6a_gal_), 3.60–3.66 (m, 1H, H-6b_gal_), 4.20 (t, 2H, *J* = 6.0 Hz, CH_2_N), 4.21 (d, 1H, *J* = 6.3 Hz, H-1_gal_), 4.34 (d, 1H, *J* = 4.4 Hz, OH), 4.57–4.71 (m, 2H, OH), 4.62 and 4.83 (qAB, 2H, *J* = 12.4 Hz, CH_2_C), 4.64 (t, 2H, *J* = 7.0 Hz, CH_2_O), 4.86 (d, 1H, *J* = 5.7 Hz, OH), 7.20 (dd, 1H, *J* = 2.1 Hz, *J* = 6.8 Hz, H-7_chin_), 7.46–7.59 (m, 3H, H-3_chin_, H-5_chin_, H-6_chin_), 8.28 (s, 1H, H-5_triaz_), 8.32 (dd, 1H, *J* = 1.6 Hz, *J* = 8.4 Hz, H-4_chin_), 8.90 (dd, 1H, *J* = 1.7 Hz, *J* = 4.1 Hz, H-2_chin_); ^13^C NMR (100 MHz, DMSO-d6): δ 29.62, 46.61, 60.53, 61.42, 65.44, 68.18, 70.47, 73.39, 75.32, 102.70, 109.85, 119.91, 121.85, 124.36, 126.79, 129.04, 135.79, 139.76, 143.96, 149.08, 154.19; HRMS (ESI-TOF): calcd for C_21_H_27_N_4_O_7_ ([M + H]^+^): *m*/*z* 447.1880; found: *m*/*z* 447.1879. 

Glycoconjugate **59:** Starting from propargyl β-D-glucopyranoside **23** and 2-methyl-8-(3-azidopropoxy)quinoline **9**, product was obtained as a yellow solid (200.3 mg, 87%); m.p.: 47-50 °C; [α]^23^_D_ = −16.6 (c = 1.0, MeOH); ^1^H NMR (400 MHz, DMSO-d6): δ 2.42 (p, 2H, *J* = 6.5 Hz, CH_2_), 2.68 (s, 3H, CH_3_), 2.93–3.09 (m, 2H, H-2_glu_, H-5_glu_), 3.10–3.20 (m, 2H, H-3_glu_, H-4_glu_), 3.40–3.50 (m, 1H, H-6a_glu_), 3.66–3.75 (m, 1H, H-6b_glu_), 4.20 (t, 2H, *J* = 6.2 Hz, CH_2_N), 4.26 (d, 1H, *J* = 7.8 Hz, H-1_glu_), 4.34 (t, 2H, *J* = 5.0 Hz, OH), 4.56 (t, 1H, *J* = 5.8 Hz, OH), 4.63 and 4.84 (qAB, 2H, *J* = 12.2 Hz, CH_2_C), 4.64 (t, 2H, *J* = 6.9 Hz, CH_2_O), 5.00 (d, 1H, *J* = 4.9 Hz, OH), 7.16 (dd, 1H, *J* = 1.3 Hz, *J* = 7.6 Hz, H-7_chin_), 7.38–7.49 (m, 3H, H-3_chin_, H-5_chin_, H-6_chin_), 8.19 (d, 1H, *J* = 8.4 Hz, H-4_chin_), 8.26 (s, 1H, H-5_triaz_); ^13^C NMR (100 MHz, DMSO-d6): δ 25.09, 29.57, 46.60, 61.16, 61.56, 65.57, 70.12, 73.40, 76.70, 76.95, 102.15, 110.35, 119.86, 122.48, 124.40, 125.75, 131.21, 136.01, 139.31, 143.15, 150.40, 153.68; HRMS (ESI-TOF): calcd for C_22_H_29_N_4_O_7_ ([M + H]^+^): *m*/*z* 461.2036; found: *m*/*z* 461.2039.

Glycoconjugate **60:** Starting from propargyl β-D-galactopyranoside **24** and 2-methyl-8-(3-azidopropoxy)quinoline **9**, product was obtained as a brown solid (149.7 mg, 65%); m.p.: 50–54 °C; [α]^24^_D_ = −4.6 (c = 1.0, MeOH); ^1^H NMR (400 MHz, DMSO-d6): δ 2.42 (p, 2H, *J* = 6.5 Hz, CH_2_), 2.67 (s, 3H, CH_3_), 3.22–3.42 (m, 4H, H-2_gal_, H-3_gal_, H-4_gal_, H-5_gal_), 3.49–3.57 (m, 1H, H-6a_gal_), 3.60–3.66 (m, 1H, H-6b_gal_), 4.20 (t, 2H, *J* = 5.9 Hz, CH_2_N), 4.21 (d, 1H, *J* = 7.0 Hz, H-1_gal_), 4.34 (bs, 1H, OH), 4.54-4.72 (m, 4H, CH_2_O, CH_2_C, OH), 4.77-4.89 (m, 2H, CH_2_C, OH), 7.16 (dd, 1H, *J* = 1.2 Hz, *J* = 7.6 Hz, H-7_chin_), 7.37-7.50 (m, 3H, H-3_chin_, H-5_chin_, H-6_chin_), 8.19 (d, 1H, *J* = 8.4 Hz, H-4_chin_), 8.26 (s, 1H, H-5_triaz_); ^13^C NMR (100 MHz, DMSO-d6): δ 25.00, 29.56, 46.58, 60.52, 61.40, 65.59, 68.18, 70.48, 73.40, 75.31, 102.71, 110.44, 119.85, 122.51, 124.32, 125.81, 127.34, 136.18, 139.12, 143.96, 153.58, 157.34; HRMS (ESI-TOF): calcd for C_22_H_29_N_4_O_7_ ([M + H]^+^): *m*/*z* 461.2036; found: *m*/*z* 461.2038.

Glycoconjugate **61:** Starting from propargyl 2,3,4,6-tetra-*O*-acetyl-β-D-glucopyranoside **21** and 8-(4-azidobutoxy)quinoline **7**, product was obtained as a yellow oil (301.7 mg, 96%); [α]^23^_D_ = −17.8 (c = 1.0, CHCl_3_); ^1^H NMR (400 MHz, CDCl_3_): δ 1.61 (bs, 2H, CH_2_), 1.92, 1.98, 2.01, 2.08 (4s, 12H, CH_3_CO), 2.26 (p, 2H, *J* = 7.1 Hz, CH_2_), 3.69 (ddd, 1H, *J* = 2.4 Hz, *J* = 4.7 Hz, *J* = 9.9 Hz, H-5_glu_), 4.12 (dd, 1H, *J* = 2.4 Hz, *J* = 12.3 Hz, H-6a_glu_), 4.24 (dd, 1H, *J* = 4.7 Hz, *J* = 12.3 Hz, H-6a_glu_), 4.28 (t, 2H, *J* = 6.1 Hz, CH_2_N), 4.58 (t, 2H, *J* = 6.9 Hz, CH_2_O), 4.68 (d, 1H, *J* = 8.0 Hz, H-1_glu_), 4.81 and 4.93 (qAB, 2H, *J* = 12.6 Hz, CH_2_C), 5.00 (dd, 1H, *J* = 8.0 Hz, *J* = 9.5 Hz H-2_glu_), 5.08 (dd-t, 1H, *J* = 9.4 Hz, *J* = 9.8 Hz, H-4_glu_), 5.17 (dd-t, 1H, *J* = 9.4 Hz, *J* = 9.4 Hz, H-3_glu_), 7.05 (dd, 1H, *J* = 1.4 Hz, *J* = 7.5 Hz, H-7_chin_), 7.38–7.49 (m, 3H, H-3_chin_, H-5_chin_, H-6_chin_), 7.95 (s, 1H, H-5_triaz_), 8.14 (dd, 1H, *J* = 1.7 Hz, *J* = 8.3 Hz, H-4_chin_), 8.93 (dd, 1H, *J* = 1.7 Hz, *J* = 4.2 Hz, H-2_chin_); ^13^C NMR (100 MHz, CDCl_3_): δ 20.60, 20.67, 20.75, 20.79, 25.61, 27.83, 49.93, 61.85, 62.86, 68.36, 71.24, 71.89, 72.86, 77.22, 99.66, 108.81, 119.91, 121.71, 123.74, 126.69, 129.54, 135.96, 140.35, 143.77, 149.32, 154.59, 169.35, 169.42, 170.19, 170.65; HRMS (ESI-TOF): calcd for C_29_H_35_N_4_O_11_ ([M + H]^+^): *m*/*z* 629.2459; found: *m*/*z* 629.2458.

Glycoconjugate **62:** Starting from propargyl 2,3,4,6-tetra-*O*-acetyl-β-D-galactopyranoside **22** and 8-(4-azidobutoxy)quinoline **7**, product was obtained as a yellow oil (292.3 mg, 93%); [α]^25^_D_ = −12.8 (c = 1.0, CHCl_3_); ^1^H NMR (400 MHz, DMSO-d6): δ 1.78–1.87 (m, 2H, CH_2_), 1.90, 1.90, 2.01, 2.10 (4s, 12H, CH_3_CO), 2.05–2.10 (m, 2H, CH_2_), 4.01-4.11 (m, 2H, H-5_gal_, H-6a_gal_), 4.16-4.25 (m, 3H, H-6b_gal_, CH_2_N), 4.54 (t, 2H, *J* = 7.0 Hz, CH_2_O), 4.64 and 4.80 (qAB, 2H, *J* = 12.4 Hz, CH_2_C), 4.81 (d, 1H, *J* = 8.0 Hz, H-1_gal_), 4.92 (dd, 1H, *J* = 8.0 Hz, *J* = 10.3 Hz, H-2_gal_), 5.15 (dd, 1H, *J* = 3.6 Hz, *J* = 10.3 Hz, H-3_gal_), 5.25 (dd, 1H, *J* = 0.9 Hz, *J* = 3.6 Hz, H-4_gal_), 7.17–7.21 (m, 1H, H-7_chin_), 7.47–7.57 (m, 3H, H-3_chin_, H-5_chin_, H-6_chin_), 8.25 (s, 1H, H-5_triaz_), 8.31 (dd, 1H, *J* = 1.7 Hz, *J* = 8.3 Hz, H-4_chin_), 8.85 (bs, 1H, H-2_chin_); ^13^C NMR (100 MHz, DMSO-d6): δ 20.29, 20.32, 20.35, 20.48, 25.53, 27.04, 49.08, 61.21, 61.81, 67.28, 67.85, 68.58, 69.90, 70.19, 98.97, 109.37, 119.55, 121.79, 124.44, 126.80, 129.00, 135.76, 139.72, 142.87, 148.92, 154.38, 169.05, 169.42, 169.86, 169.89; HRMS (ESI-TOF): calcd for C_30_H_37_N_4_O_11_ ([M + H]^+^): *m*/*z* 629.2459; found: *m*/*z* 629.2459.

Glycoconjugate **63:** Starting from propargyl 2,3,4,6-tetra-*O*-acetyl-β-D-glucopyranoside **21** and 2-methyl-8-(4-azidobutoxy)quinoline **10**, product was obtained as a yellow oil (295.6 mg, 92%); [α]^23^_D_ = −18.8 (c = 1.0, CHCl_3_); ^1^H NMR (400 MHz, DMSO-d6): δ 1.80 (p, 2H, *J* = 6.6 Hz, CH_2_), 1.88, 1.91, 1.97, 2.02 (4s, 12H, CH_3_CO), 2.08 (p, 2H, *J* = 7.4 Hz, CH_2_), 2.63 (s, 3H, CH_3_), 3.97 (ddd, 1H, *J* = 2.4 Hz, *J* = 4.9 Hz, *J* = 10.0 Hz, H-5_glu_), 4.03 (dd, 1H, *J* = 2.4 Hz, *J* = 12.4 Hz, H-6a_glu_), 4.14–4.24 (m, 3H, CH_2_N, H-6b_glu_), 4.59 (t, 2H, *J* = 6.9 Hz, CH_2_O), 4.64 and 4.80 (qAB, 2H, *J* = 12.4 Hz, CH_2_C), 4.75 (dd, 1H, *J* = 8.0 Hz, *J* = 9.7 Hz H-2_glu_), 4.90 (d, 1H, *J* = 8.0 Hz, H-1_glu_), 4.91 (dd-t, 1H, *J* = 9.7 Hz, *J* = 9.7 Hz, H-4_glu_), 5.24 (dd-t, 1H, *J* = 9.6 Hz, *J* = 9.6 Hz, H-3_glu_), 7.15 (dd, 1H, *J* = 1.8 Hz, *J* = 7.2 Hz, H-7_chin_), 7.38–7.47 (m, 3H, H-3_chin_, H-5_chin_, H-6_chin_), 8.18 (d, 1H, *J* = 8.4 Hz, H-4_chin_), 8.34 (s, 1H, H-5_triaz_); ^13^C NMR (100 MHz, DMSO-d6): δ 20.23, 20.30, 20.34, 20.47, 24.93, 25.25, 27.23, 49.08, 61.64, 61.92, 68.09, 68.14, 70.60, 70.82, 72.03, 98.53, 109.64, 119.42, 122.41, 124.62, 125.74, 127.26, 135.96, 139.19, 142.70, 153.89, 157.21, 168.92, 169.21, 169.46, 170.00; HRMS (ESI-TOF): calcd for C_31_H_39_N_4_O_11_ ([M + H]^+^): *m*/*z* 643.2615; found: *m*/*z* 643.2618.

Glycoconjugate **64:** Starting from propargyl 2,3,4,6-tetra-*O*-acetyl-β-D-galactopyranoside **22** and 2-methyl-8-(4-azidobutoxy)quinoline **10**, product was obtained as a yellow oil (321.3 mg, 100%); [α]^24^_D_ = −13.6 (c = 1.0, CHCl_3_); ^1^H NMR (400 MHz, DMSO-d6): δ 1.81 (p, 2H, *J* = 6.6 Hz, CH_2_), 1.89, 1.89, 2.00, 2.10 (4s, 12H, CH_3_CO), 2.03–2.10 (m, 2H, CH_2_), 2.63 (s, 3H, CH_3_), 4.01–4.11 (m, 2H, H-5_gal_, H-6a_gal_), 4.15–4.25 (m, 3H, H-6a_gal_, CH_2_N), 4.59 (t, 2H, *J* = 7.0 Hz, CH_2_O), 4.64 and 4.80 (qAB, 2H, *J* = 12.9 Hz, CH_2_C), 4.81 (d, 1H, *J* = 8.0 Hz, H-1_gal_), 4.92 (dd, 1H, *J* = 8.0 Hz, *J* = 10.3 Hz, H-2_gal_), 5.14 (dd, 1H, *J* = 3.6 Hz, *J* = 10.3 Hz, H-3_gal_), 5.24 (dd, 1H, *J* = 0.9 Hz, *J* = 3.6 Hz, H-4_gal_), 7.15 (dd, 1H, *J* = 1.8 Hz, *J* = 7.2 Hz, H-7_chin_), 7.38–7.47 (m, 3H, H-3_chin_, H-5_chin_, H-6_chin_), 8.18 (d, 1H, *J* = 8.4 Hz, H-4_chin_), 8.33 (s, 1H, H-5_triaz_); ^13^C NMR (100 MHz, DMSO-d6): δ 20.22, 20.28, 20.34, 20.47, 24.93, 25.26, 27.25, 49.08, 61.20, 61.78, 67.28, 68.14, 68.57, 69.89, 70.19, 98.95, 109.64, 119.41, 122.40, 124.57, 125.74, 127.26, 135.96, 139.19, 142.78, 153.89, 157.21, 168.02, 169.41, 169.84, 169.88; HRMS (ESI-TOF): calcd for C_31_H_39_N_4_O_11_ ([M + H]^+^): *m*/*z* 643.2615; found: *m*/*z* 643.2612.

Glycoconjugate **65:** Starting from propargyl β-D-glucopyranoside **23** and 8-(4-azidobutoxy)quinoline **7**, product was obtained as a yellow solid (165.8 mg, 72%); m.p.: 49–51 °C; [α]^24^_D_ = −20.2 (c = 1.0, MeOH); ^1^H NMR (400 MHz, DMSO-d6): δ 1.77-1.89 (m, 2H, CH_2_), 2.03-2.15 (m, 2H, CH_2_), 2.94–3.09 (m, 2H, H-2_glu_, H-5_glu_), 3.10–3.17 (m, 2H, H-3_glu_, H-4_glu_), 3.40–3.50 (m, 1H, H-6a_glu_), 3.67–3.75 (m, 1H, H-6b_glu_), 4.05–4.11 (m, 1H, OH), 4.21 (m, 2H, CH_2_N), 4.27 (d, 1H, *J* = 7.8 Hz, H-1_glu_), 4.34 (t, 1H, *J* = 5.1 Hz, OH), 4.52 (t, 1H, *J* = 6.8 Hz, CH_2_O), 4.57 (t, 1H, *J* = 5.9 Hz, OH), 4.64 and 4.85 (qAB, 2H, J = 12.2 Hz, CH_2_C), 5.01 (d, 1H, *J* = 4.9 Hz, OH), 7.17–7.22 (m, 1H, H-7_chin_), 7.45–7.61 (m, 3H, H-3_chin_, H-5_chin_, H-6_chin_), 8.28 (s, 1H, H-5_triaz_), 8.31–8.35 (m, 1H, H-4_chin_), 8.87 (bs, 1H, H-2_chin_); ^13^C NMR (100 MHz, DMSO-d6): δ 25.61, 26.96, 49.06, 61.16, 61.59, 67.84, 70.11, 73.38, 76.70, 76.95, 102.15, 109.35, 119.64, 121.85, 124.42, 126.84, 128.86, 135.68, 139.79, 143.73, 148.98, 154.35; HRMS (ESI-TOF): calcd for C_22_H_29_N_4_O_7_ ([M + H]^+^): *m*/*z* 461.2036; found: *m*/*z* 461.2036. 

Glycoconjugate **66:** Starting from propargyl β-D-galactopyranoside **24** and 8-(4-azidobutoxy)quinoline **7**, product was obtained as a yellow solid (207.2 mg, 90%); m.p.: 62–64 °C; [α]^24^_D_ = −13.0 (c = 1.0, MeOH); ^1^H NMR (400 MHz, DMSO-d6): δ 1.83 (p, 2H, *J* = 6.3 Hz, CH_2_), 2.08 (p, 2H, *J* = 7.1 Hz, CH_2_), 3.24–3.30 (m, 1H, H-2_gal_), 3.33–3.48 (m, 3H, H-3_gal_, H-4_gal_, H-5_gal_), 3.50–3.57 (m, 1H, H-6a_gal_), 3.61–3.66 (m, 1H, H-6b_gal_), 4.21 (t, 2H, *J* = 6.3 Hz, CH_2_N), 4.22 (d, 1H, *J* = 7.5 Hz, H-1_gal_), 4.31–4.36 (m, 2H, OH), 4.52 (t, 2H, *J* = 7.0 Hz, CH_2_O), 4.62 and 4.83 (qAB, 2H, *J* = 12.1 Hz, CH_2_C), 4.67 (d, 1H, *J* = 5.1 Hz, OH), 4.87 (d, 1H, *J* = 4.6 Hz, OH), 7.17–7.21 (m, 1H, H-7_chin_), 7.46–7.57 (m, 3H, H-3_chin_, H-5_chin_, H-6_chin_), 8.27 (s, 1H, H-5_triaz_), 8.31 (dd, 1H, *J* = 1.7 Hz, *J* = 8.3 Hz, H-4_chin_), 8.87 (dd, 1H, *J* = 1.7 Hz, *J* = 4.1 Hz, H-2_chin_); ^13^C NMR (100 MHz, DMSO-d6): δ 25.59, 26.98, 49.03, 60.52, 61.45, 67.82, 68.18, 70.47, 73.40, 75.32, 102.73, 109.39, 119.55, 121.80, 124.33, 126.80, 129.00, 135.75, 139.71, 143.77, 148.99, 154.38; HRMS (ESI-TOF): calcd for C_22_H_29_N_4_O_7_ ([M + H]^+^): *m*/*z* 461.2036; found: *m*/*z* 461.2035.

Glycoconjugate **67:** Starting from propargyl β-D-glucopyranoside **23** and 2-methyl-8-(4-azidobutoxy)quinoline **10**, product was obtained as an orange solid (180.3 mg, 76%); m.p.: 75–78 °C; [α]^24^_D_ = −17.6 (c = 1.0, MeOH); ^1^H NMR (400 MHz, DMSO-d6): δ 1.82 (p, 2H, *J* = 6.5 Hz, CH_2_), 2.08 (p, 2H, *J* = 7.0 Hz, CH_2_), 2.64 (s, 3H, CH_3_), 2.94–3.09 (m, 2H, H-2_glu_, H-5_glu_), 3.10–3.19 (m, 2H, H-3_glu_, H-4_glu_), 3.41–3.49 (m, 1H, H-6a_glu_), 3.66–3.74 (m, 1H, H-6b_glu_), 4.20 (t, 2H, *J* = 6.3 Hz, CH_2_N), 4.27 (d, 1H, *J* = 7.8 Hz, H-1_glu_), 4.55 (d, 1H, *J* = 5.9 Hz, OH), 4.56 (t, 2H, *J* = 6.9 Hz, CH_2_O), 4.64 and 4.86 (qAB, 2H, *J* = 12.2 Hz, CH_2_C), 4.89 (t, 2H, *J* = 5.2 Hz, OH), 4.91 (t, 1H, *J* = 4.7 Hz, OH), 5.01 (d, 1H, *J* = 4.9 Hz, OH), 7.15 (dd, 1H, *J* = 1.8 Hz, *J* = 7.1 Hz, H-7_chin_), 7.38–7.47 (m, 3H, H-3_chin_, H-5_chin_, H-6_chin_), 8.18 (d, 1H, *J* = 8.4 Hz, H-4_chin_), 8.31 (s, 1H, H-5_triaz_); ^13^C NMR (100 MHz, DMSO-d6): δ 25.40, 25.44, 27.09, 49.07, 61.13, 61.55, 68.07, 70.09, 73.37, 76.69, 76.93, 102.13, 109.62, 119.44, 122.43, 124.51, 125.78, 127.28, 135.98, 139.16, 145.72, 148.08, 153.85; HRMS (ESI-TOF): calcd for C_23_H_31_N_4_O_7_ ([M + H]^+^): *m*/*z* 475.2193; found: *m*/*z* 475.2194.

Glycoconjugate **68:** Starting from propargyl β-D-galactopyranoside **24** and 2-methyl-8-(4-azidobutoxy)quinoline **10**, product was obtained as an orange solid (156.6 mg, 66%); m.p.: 38–41 °C; [α]^23^_D_ = −6.0 (c = 1.0, MeOH); ^1^H NMR (400 MHz, DMSO-d6): δ 1.82 (m, 2H, CH_2_), 2.08 (m, 2H, CH_2_), 2.64 (s, 3H, CH_3_), 3.23–3.43 (m, 4H, H-2_gal_, H-3_gal_, H-4_gal_, H-5_gal_), 3.49–3.58 (m, 1H, H-6a_gal_), 3.61–3.66 (m, 1H, H-6b_gal_), 4.15–4.25 (m 3H, H-1_gal_, CH_2_N), 4.56 (t, 2H, *J* = 6.8 Hz, CH_2_O), 4.62 and 4.83 (qAB, 2H, *J* = 12.1 Hz, CH_2_C), 7.16 (m, 1H, H-7_chin_), 7.36–7.52 (m, 3H, H-3_chin_, H-5_chin_, H-6_chin_), 8.19 (d, 1H, *J* = 8.2 Hz, H-4_chin_), 8.29 (s, 1H, H-5_triaz_); ^13^C NMR (100 MHz, DMSO-d6): δ 24.94, 25.40, 27.10, 49.06, 60.50, 61.43, 68.08, 68.17, 70.47, 73.41, 75.31, 102.73, 109.74, 119.43, 122.45, 124.40, 125.81, 127.28, 136.09, 139.06, 143.74, 153.83, 157.26; HRMS (ESI-TOF): calcd for C_23_H_31_N_4_O_7_ ([M + H]^+^): *m*/*z* 475.2193; found: *m*/*z* 475.2199.

Glycoconjugate **69:** Starting from 2-azidoethyl 2,3,4,6-tetra-*O*-acetyl-β-D-glucopyranoside **27** and 8-(2-propyn-1-yloxy)quinoline **3**, product was obtained as a yellow solid (234.2 mg, 78%); m.p.: 52–55 °C; [α]^22^_D_ = −13.6 (c = 1.0, CHCl_3_); ^1^H NMR (400 MHz, CDCl_3_): δ 1.92, 1.98, 2.02, 2.06 (4s, 12H, CH_3_CO), 3.67 (ddd, 1H, *J* = 2.4 Hz, *J* = 4.7 Hz, *J* = 10.0 Hz, H-5_glu_), 3.93–3.99 (m, 1H, CH_2_O), 4.11 (dd, 1H, *J* = 2.4 Hz, *J* = 12.4 Hz, H-6a_glu_), 4.19–4.24 (m, 2H, H-6b_glu_, CH_2_O_._), 4.48 (d, 1H, *J* = 7.9 Hz, H-1_glu_), 4.49–4.62 (m, 2H, CH_2_N), 4.95 (dd, 1H, *J* = 7.9 Hz, *J* = 9.5 Hz H-2_glu_), 5.04 (dd, 1H, *J* = 9.4 Hz, *J* = 10.0 Hz, H-4_glu_), 5.16 (dd-t, 1H, *J* = 9.4 Hz, *J* = 9.5 Hz, H-3_glu_), 5.54 (s, 2H, CH_2_O_chin_), 7.32–7.50 (m, 4H, H-3_chin_, H-5_chin_, H-6_chin,_ H-7_chin_), 7.84 (s, 1H, H-5_triaz_), 8.14 (d, 1H, *J* = 8.0 Hz, H-4_chin_), 8.94 (bs, 1H, H-2_chin_); ^13^C NMR (100 MHz, CDCl_3_): δ 20.53, 20.55, 20.64, 20.70, 50.06, 61.74, 62.79, 67.58, 68.25, 70.90, 71.99, 72.52, 100.54, 110.09, 120.29, 121.63, 124.62, 126.77, 129.50, 136.05, 140.21, 143.85, 149.26, 153.90, 169.29, 169.35, 170.08, 170.54; HRMS (ESI-TOF): calcd for C_28_H_33_N_4_O_11_ ([M + H]^+^): *m*/*z* 601.2146; found: *m*/*z* 601.2149.

Glycoconjugate **70**: Starting from 2-azidoethyl 2,3,4,6-tetra-*O*-acetyl-β-D-galactopyranoside **28** and 8-(2-propyn-1-yloxy)quinoline **3**, product was obtained as a yellow solid (186.2 mg, 62%); m.p.: 55–58 °C; [α]^23^_D_ = −18.4 (c = 1.0, CHCl_3_); ^1^H NMR (400 MHz, DMSO-d6): δ 1.90, 2.00, 2.09, 2.10 (4s, 12H, CH_3_CO), 3.91–3.99 (m, 1H, CH_2_O), 4.00-4.22 (m, 4H, H-6a_gal_, H-6b_gal_, H-5_gal_, CH_2_O), 4.53–4.66 (m, 2H, CH_2_N), 4.74 (d, 1H, *J* = 8.0 Hz, H-1_gal_), 4.91 (dd, 1H, *J* = 8.0 Hz, *J* = 10.4, H-2_gal_), 5.12 (dd, 1H, *J* = 3.6 Hz, *J* = 10.4 Hz, H-3_gal_), 5.25 (dd, 1H, *J* = 0.8 Hz, *J* = 3.6 Hz, H-4_gal_), 5.35 (s, 2H, CH_2_O_chin_), 7.41 (dd, 1H, *J* = 3.4 Hz, *J* = 5.6 Hz, H-7_chin_), 7.50–7.57 (m, 3H, H-3_chin_, H-5_chin_, H-6_chin_), 8.15 (s, 1H, H-5_triaz_), 8.31 (dd, 1H, *J* = 1.6 Hz, *J* = 8.3 Hz, H-4_chin_), 8.83 (bs, 1H, H-2_chin_); ^13^C NMR (100 MHz, DMSO-d6): δ 20.28, 20.32, 20.34, 20.47, 49.34, 61.21, 61.88, 67.26, 68.31, 69.98, 70.08, 79.14, 99.61, 110.10, 120.05, 121.82, 124.97, 126.72, 129.03, 135.75, 139.75, 142.46, 148.92, 153.85, 169.13, 169.43, 169.86, 169.88; HRMS (ESI-TOF): calcd for C_28_H_33_N_4_O_11_ ([M + H]^+^): *m*/*z* 601.2146; found: *m*/*z* 601.2140.

Glycoconjugate **71**: Starting from 2-azidoethyl 2,3,4,6-tetra-*O*-acetyl-β-D-glucopyranoside **27** and 2-methyl-8-(2-propyn-1-yloxy)quinoline **4**, product was obtained as a pink solid (298.1 mg, 97%); m.p.: 50–53 °C; [α]^24^_D_ = −13.0 (c = 1.0, CHCl_3_); ^1^H NMR (400 MHz, DMSO-d6): δ 1.89, 1.92, 1.98, 2.01 (4s, 12H, CH_3_CO), 2.63 (s, 3H, CH_3_), 3.91–4.00 (m, 2H, H-5_glu_, CH_2_O), 4.04 (dd, 1H, *J* = 2.5 Hz, *J* = 12.4 Hz, H-6a_glu_), 4.09–4.13 (m, 1H, CH_2_O), 4.17 (dd, 1H, *J* = 5.0 Hz, *J* = 12.4 Hz, H-6b_glu_), 4.52–4.66 (m, 2H, CH_2_N), 4.75 (dd, 1H, *J* = 8.1 Hz, *J* = 9.7 Hz, H-2_glu_), 4.85 (d, 1H, *J* = 8.1 Hz, H-1_glu_), 4.90 (dd-t, 1H, *J* = 9.4 Hz, *J* = 9.8 Hz, H-4_glu_), 5.23 (dd-t, 1H, *J* = 9.4 Hz, *J* = 9.7 Hz, H-3_glu_), 5.34 (s, 2H, CH_2_O_chin_), 7.37 (dd, 1H, *J* = 1.7 Hz, *J* = 7.4 Hz, H-7_chin_), 7.41 (d, 1H, *J* = 8.4 Hz, H-3_chin_), 7.43 (dd, 1H, *J* = 7.4 Hz, *J* = 8.1 Hz, H-6_chin_), 7.47 (dd, 1H, *J* = 1.7 Hz, *J* = 8.1 Hz, H-5_chin_), 8.16 (s, 1H, H-5_triaz_), 8.18 (d, 1H, *J* = 8.4 Hz, H-4_chin_); ^13^C NMR (100 MHz, DMSO-d6): δ 20.19, 20.20, 20.31, 20.43, 24.85, 49.27, 61.60, 61.71, 67.39, 68.07, 70.51, 71.87, 79.12, 99.12, 110.26, 119.83, 122.40, 125.05, 125.60, 127.29, 135.90, 139.17, 142.50, 153.34, 157.19, 168.98, 169.19, 169.44, 169.97; HRMS (ESI-TOF): calcd for C_29_H_35_N_4_O_11_ ([M + H]^+^): *m*/*z* 615.2302; found: *m*/*z* 615.2303.

Glycoconjugate **72**: Starting from 2-azidoethyl 2,3,4,6-tetra-*O*-acetyl-β-D-galactopyranoside **28** and 2-methyl-8-(2-propyn-1-yloxy)quinoline **4**, product was obtained as a pink solid (236.6 mg, 77%); m.p.: 60–63 °C; [α]^24^_D_ = −13.6 (c = 1.0, CHCl_3_); ^1^H NMR (400 MHz, DMSO-d6): δ 1.89, 1.90, 2.00, 2.10 (4s, 12H, CH_3_CO), 2.62 (s, 3H, CH_3_), 3.95 (m, 1H, CH_2_O), 4.00–4.08 (m, 2H, H-6a_gal_, H-6b_gal_), 4.08–4.21 (m, 2H, CH_2_O, H-5_gal_), 4.52–4.66 (m, 2H, CH_2_N), 4.74 (d, 1H, *J* = 8.0 Hz, H-1_gal_), 4.92 (dd, 1H, *J* = 8.0 Hz, *J* = 10.4, H-2_gal_), 5.12 (dd, 1H, *J* = 3.6 Hz, *J* = 10.4 Hz H-3_gal_), 5.25 (dd, 1H, *J* = 0.8 Hz, *J* = 3.6 Hz, H-4_gal_), 5.34 (s, 2H, CH_2_O_chin_), 7.37 (dd, 1H, *J* = 1.6 Hz, *J* = 7.4 Hz, H-7_chin_), 7.41 (d, 1H, *J* = 8.4 Hz, H-3_chin_), 7.43 (dd, 1H, *J* = 7.4 Hz, *J* = 8.2 Hz, H-6_chin_), 7.47 (dd, 1H, *J* = 1.6 Hz, *J* = 8.2 Hz, H-5_chin_), 8.15 (s, 1H, H-5_triaz_), 8.18 (d, 1H, *J* = 8.4 Hz, H-4_chin_); ^13^C NMR (100 MHz, DMSO-d6): δ 20.30, 20.37, 20.50, 21.05, 24.89, 49.36, 61.23, 61.79, 67.29, 68.31, 70.01, 70.11, 79.17, 99.65, 110.33, 119.90, 122.46, 125.05, 125.67, 127.35, 135.96, 139.23, 142.58, 153.40, 157.25, 169.18, 169.46, 169.89, 169.9k2; HRMS (ESI-TOF): calcd for C_29_H_35_N_4_O_11_ ([M + H]^+^): *m*/*z* 615.2302; found: *m*/*z* 615.2305.

Glycoconjugate **73**: Starting from 2-azidoethyl β-D-glucopyranoside **29** and 8-(2-propyn-1-yloxy)quinoline **3**, product was obtained as a beige solid (134.1 mg, 62%); m.p.: 60–63 °C; [α]^23^_D_ = −9.6 (c = 1.0, MeOH); ^1^H NMR (400 MHz, DMSO-d6): δ 2.95-3.08 (m, 1H, H-5_glu_), 3.10–3.19 (m, 3H, H-2_glu_, H-3_glu_, H-4_glu_), 3.40-3.48 (m, 1H, H-6a_glu_), 3.65–3.72 (m, 1H, H-6b_glu_), 3.91–3.98 (m, 1H, CH_2_O), 4.07-4.14 (m, 1H, CH_2_O), 4.26 (d, 1H, *J* = 7.8 Hz, H-1_glu_), 4.52–4.56 (m, 1H, OH), 4.60–4.66 (m, 2H, CH_2_N), 4.88–4.96 (m, 2H, OH), 5.10–5.14 (d, 1H, *J* = 4.7 Hz, OH), 5.34 (s, 2H, CH_2_O_chin_), 7.39–7.45 (m, 1H, H-7_chin_), 7.50–7.57 (m, 3H, H-3_chin_, H-5_chin_, H-6_chin_), 8.32 (dd, 1H, *J* = 1.8 Hz, *J* = 8.1 Hz, H-4_chin_), 8.38 (s, 1H, H-5_triaz_), 8.83 (dd, 1H, *J* = 1.8 Hz, *J* = 4.1 Hz, H-2_chin_); ^13^C NMR (100 MHz, DMSO-d6): δ 49.74, 61.06, 61.82 67.36, 70.01, 73.27, 76.59, 76.96, 102.95, 109.96, 119.95, 121.81, 125.69, 126.73, 129.03, 135.78, 139.67, 142.32, 148.96, 153.84; HRMS (ESI-TOF): calcd for C_20_H_25_N_4_O_7_ ([M + H]^+^): *m*/*z* 433.1723; found: *m*/*z* 433.1719.

Glycoconjugate **74:** Starting from 2-azidoethyl β-D-galactopyranoside **30** and 8-(2-propyn-1-yloxy)quinoline **3**, product was obtained as a brown oil (118.9 mg, 55%); [α]^25^_D_ = 10.8 (c = 1.0, MeOH); ^1^H NMR (400 MHz, D_2_O): δ 3.44-3.54 (m, 1H, H-2_gal_), 3.57–3.74 (m, 4H, H-3_gal_, H-5_gal_, H-6a_gal_, H-6b_gal_), 3.91 (d, 1H, *J* = 3.2 Hz, H-4_gal_), 4.11-4.21 (m, 1H, CH_2_O), 4.30–4.40 (m, 2H, CH_2_O, H-1_gal_), 4.72-4.78 (m, 2H, CH_2_N_triaz_), 5.66 (s, 2H, CH_2_O_chin_), 7.78 (dd, 1H, *J* = 2.3 Hz, *J* = 6.4 Hz, H-7_chin_), 7.85-7.97 (m, 2H, H-3_chin_, H-5_chin_), 8.06–8.16 (m, 1H, H-6_chin_), 8.35 (s, 1H, H-5_triaz_), 9.05 (d, 1H, *J* = 4.7 Hz, H-2_chin_), 9.14 (d, 1H, *J* = 8.1 Hz, H-4_chin_); ^13^C NMR (100 MHz, D_2_O): δ 50.49, 60.83, 62.27, 67.96, 68.45, 70.52, 72.54, 75.05, 102.97, 114.63, 120.83, 122.17, 126.24, 129.37, 129.92, 130.37, 142.07, 143.05, 147.30, 147.5; HRMS (ESI-TOF): calcd for C_20_H_25_N_4_O_7_ ([M + H]^+^): *m*/*z* 433.1723; found: *m*/*z* 433.1720.

Glycoconjugate **75**: Starting from 2-azidoethyl β-D-glucopyranoside **29** and 2-methyl-8-(2-propyn-1-yloxy)quinoline **4**, product was obtained as a brown oil (129.5 mg, 58%); [α]^24^_D_ = 1.6 (c = 1.0, MeOH); ^1^H NMR (400 MHz, D_2_O): δ 3.01 (s, 3H, CH_3_), 3.18 (dd-t, 1H, *J* = 8.2 Hz, *J* = 8.3 Hz, H-4_glu_), 3.29 (dd-t, 1H, *J* = 8.6 Hz, *J* = 9.0 Hz, H-2_glu_), 3.38–3.47 (m, 2H, H-3_glu_, H-5_glu_), 3.63 (dd, 1H, *J* = 4.7 Hz, *J* = 12.3 Hz, H-6a_glu_), 3.81–3.90 (m, 1H, H-6b_glu_), 4.10–4.22 (m, 1H, CH_2_O), 4.27–4.37 (m, 1H, CH_2_O), 4.41 (d, 1H, *J* = 7.8 Hz, H-1_glu_), 4.68–4.77 (m, 2H, CH_2_N), 5.64 (s, 2H, CH_2_O_chin_), 7.64–7.96 (m, 4H, H-3_chin_, H-5_chin_, H-6_chin_, H-7_chin_), 8.30 (s, 1H, H-5_triaz_), 8.91 (d, 1H, *J* = 8.3 Hz, H-4_chin_); ^13^C NMR (100 MHz, D_2_O): δ 19.99, 50.45, 60.60, 62.10, 67.96, 69.49, 72.88, 75.53, 75.82, 102.32, 114.65, 120.85, 124.20, 126.25, 128.08, 129.08, 129.44, 146.30, 146.98, 155.63, 157.49; HRMS (ESI-TOF): calcd for C_21_H_27_N_4_O_7_ ([M + H]^+^): *m*/*z* 447.1880; found: *m*/*z* 447.1880.

Glycoconjugate **76**: Starting from 2-azidoethyl β-D-galactopyranoside **30** and 2-methyl-8-(2-propyn-1-yloxy)quinoline **4**, product was obtained as a pink solid (131.7 mg, 59%); m.p.: 102-105 °C; [α]^23^_D_ = 4.8 (c = 1.0, MeOH); ^1^H NMR (400 MHz, DMSO-d6): δ 2.63 (s, 3H, CH_3_), 3.33–3.39 (m, 2H, H-2_gal_, H-3_gal_), 3.40–3.48 (m, 1H, H-5_gal_), 3.49–3.55 (m, 2H, H-6a_gal_, H-6b_gal_), 3.63 (dd-t, 1H, *J* = 0.8 Hz, *J* = 3.6 Hz, H-4_gal_), 3.88–3.96 (m, 1H, CH_2_O), 4.06–4.14 (m, 1H, CH_2_O), 4.19 (d, 1H, *J* = 7.3 Hz, H-1_gal_), 4.35 (d, 1H, *J* = 4.6 Hz, OH), 4.53–4.64 (m, 3H, CH_2_N, OH), 4.69 (d, 1H, *J* = 5.0 Hz, OH), 4.95 (d, 1H, *J* = 4.4 Hz, OH), 5.33 (s, 2H, CH_2_O_chin_), 7.38 (dd, 1H, *J* = 2.0 Hz, *J* = 7.2 Hz, H-7_chin_), 7.41 (d, 1H, *J* = 8.4 Hz, H-3_chin_), 7.44 (dd, 1H, *J* = 7.2 Hz, *J* = 7.8 Hz, H-6_chin_), 7.47 (dd, 1H, *J* = 1.8 Hz, *J* = 7.8 Hz, H-5_chin_), 8.18 (d, 1H, *J* = 8.4 Hz, H-4_chin_), 8.37 (s, 1H, H-5_triaz_); ^13^C NMR (100 MHz, DMSO-d6): δ 24.89, 49.75, 60.46, 61.66, 67.16, 68.14, 70.35, 73.29, 75.39, 103.53, 110.20, 119.77, 122.42, 125.65, 125.69, 127.30, 135.93, 139.16, 142.39, 153.37, 157.23; HRMS (ESI-TOF): calcd for C_21_H_27_N_4_O_7_ ([M + H]^+^): *m*/*z* 447.1880; found: *m*/*z* 447.1881.

Glycoconjugate **77**: Starting from 2,3,4,6-tetra-*O*-acetyl-*N*-(β-D-glucopyranosyl)azidoacetamide **33** and 8-(2-propyn-1-yloxy)quinoline **3**, product was obtained as a light yellow solid (205.5 mg, 67%); m.p.: 97–100 °C; [α]^25^_D_ = 4.4 (c = 0.6, CHCl_3_); ^1^H NMR (400 MHz, DMSO-d6): δ 1.94, 1.97, 1.99, 2.00 (4s, 12H, CH_3_CO), 3.99 (m, 1H, H-6a_glu_), 4.10–4.18 (m, 2H, H-5_glu_, H-6b_glu_), 4.87 (dd-t, 1H, *J* = 9.4 Hz, *J* = 9.4 Hz, H-4_glu_), 4.93 (dd-t, 1H, *J* = 9.7 Hz, *J* = 9.7 Hz, H-2_glu_), 5.16 and 5.22 (qAB, 2H, *J* = 16.5 Hz, CH_2_CO), 5.34–5.41 (m, 3H, H-1_glu_, CH_2_O_chin_), 5.45 (dd-t, 1H, *J* = 9.4 Hz, *J* = 9.4 Hz, H-3_glu_), 7.41 (dd, 1H, *J* = 3.6 Hz, *J* = 5.4 Hz, H-7_chin_), 7.49–7.57 (m, 3H, H-3_chin_, H-5_chin_, H-6_chin_), 8.24 (s, 1H, H-5_triaz_), 8.32 (dd, 1H, *J* = 1.7 Hz, *J* = 8.3 Hz, H-4_chin_), 8.83 (d, 1H, *J* = 2.7 Hz, H-2_chin_), 9.25 (d, 1H, *J* = 9.3 Hz, CONH); ^13^C NMR (100 MHz, DMSO-d6): δ 20.27, 20.30, 20.33, 20.48, 51.40, 61.73, 67.72, 70.58, 72.13, 72.64, 76.85, 79.12, 109.94, 119.95, 121.81, 126.38, 126.69, 129.02, 135.73, 139.70, 142.37, 148.94, 153.82, 166.17, 169.13, 169.25, 169.44, 169.94; HRMS (ESI-TOF): calcd for C_28_H_32_N_5_O_11_ ([M + H]^+^): *m*/*z* 614.2098; found: *m*/*z* 614.2095.

Glycoconjugate **78**: Starting from 2,3,4,6-tetra-*O*-acetyl-*N*-(β-D-galactopyranosyl)azidoacetamide **34** and 8-(2-propyn-1-yloxy)quinoline **3**, product was obtained as a yellow solid (227.0 mg, 74%); m.p.: 100–103 °C; [α]^22^_D_ = 14.8 (c = 1.0, CHCl_3_); ^1^H NMR (400 MHz, DMSO-d6): δ 1.92, 1.98, 1.99, 2.12 (4s, 12H, CH_3_CO), 3.98 (dd, 1H, *J* = 6.7 Hz, *J* = 11.3 Hz, H-6a_gal_), 4.05 (dd, 1H, *J* = 5.9 Hz, *J* = 11.3 Hz, H-6b_gal_), 4.35 (m, 1H, H-5_gal_), 5.08 (dd, 1H, *J* = 9.0 Hz, *J* = 9.4 Hz, H-2_gal_), 5.13 and 5.21 (qAB, 2H, *J* = 16.5 Hz, CH_2_CO), 5.27-5.44 (m, 5H, H-1_gal_, H-3_gal_, H-4_gal_, CH_2_O_chin_), 7.42 (dd, 1H, *J* = 3.2 Hz, *J* = 5.8 Hz, H-7_chin_), 7.50–7.58 (m, 3H, H-3_chin_, H-5_chin_, H-6_chin_), 8.24 (s, 1H, H-5_triaz_), 8.28–8.35 (m, 2H, H-4_chin_, H-2_chin_), 9.34 (d, 1H, *J* = 9.5 Hz, CONH); ^13^C NMR (100 MHz, DMSO-d6): δ 20.33, 20.38, 20.48, 20.72, 51.37, 61.42, 61.75, 67.54, 68.24, 70.66, 71.43, 77.22, 109.95, 119.98, 121.88, 124.11, 126.39, 126.70, 135.73, 139.72, 142.38, 148.91, 153.85, 166.09, 169.31, 169.37, 169.78, 169.85; HRMS (ESI-TOF): calcd for C_28_H_32_N_5_O_11_ ([M + H]^+^): *m*/*z* 614.2098; found: *m*/*z* 614.2091.

Glycoconjugate **79**: Starting from 2,3,4,6-tetra-*O*-acetyl-*N*-(β-D-glucopyranosyl)azidoacetamide **33** and 2-methyl-8-(2-propyn-1-yloxy)quinoline **4**, product was obtained as a pink solid (254.2 mg, 81%); m.p.: 93–96 °C; [α]^27^_D_ = 5.2 (c = 1.0, CHCl_3_); ^1^H NMR (400 MHz, DMSO-d6): δ 1.94, 1.97, 1.99, 2.00 (4s, 12H, CH_3_CO), 2.63 (s, 3H, CH_3_), 3.94–4.03 (m, 1H, H-6a_glu_), 4.06–4.18 (m, 2H, H-5_glu_, H-6b_glu_), 4.87 (dd-t, *J* = 9.4 Hz, *J* = 10.4 Hz, H-4_glu_), 4.92 (dd-t, *J* = 9.7 Hz, *J* = 9.8 Hz, H-2_glu_), 5.15 and 5.21 (qAB, 2H, *J* = 16.5 Hz, CH_2_CO), 5.33–5.48 (m, 4H, H-1_glu_, H-3_glu_, CH_2_O_chin_), 7.38 (dd, 1H, *J* = 1.6 Hz, *J* = 7.5 Hz, H-7_chin_), 7.41 (d, 1H, *J* = 8.4 Hz, H-3_chin_), 7.43 (dd, 1H, *J* = 7.5 Hz, *J* = 8.0 Hz, H-6_chin_), 7.47 (dd, 1H, *J* = 1.6 Hz, *J* = 8.0 Hz, H-5_chin_), 8.18 (d, 1H, *J* = 8.4 Hz, H-4_chin_), 8.24 (s, 1H, H-5_triaz_), 9.25 (d, 1H, *J* = 9.3 Hz, CONH); ^13^C NMR (100 MHz, DMSO-d6): δ 20.21, 20.24, 20.27, 20.41, 24.85, 51.34, 61.53, 67.65, 70.51, 72.06, 72.57, 76.78, 79.06, 110.09, 119.72, 122.38, 125.54, 126.38, 127.27, 135.85, 139.11, 142.37, 153.27, 157.15, 166.10, 169.06, 169.19, 169.37, 169.87; HRMS (ESI-TOF): calcd for C_29_H_34_N_5_O_11_ ([M + H]^+^): *m*/*z* 628.2255; found: *m*/*z* 628.2256.

Glycoconjugate **80**: Starting from 2,3,4,6-tetra-*O*-acetyl-*N*-(β-D-galactopyranosyl)azidoacetamide **34** and 2-methyl-8-(2-propyn-1-yloxy)quinoline **4**, product was obtained as a pink solid (244.8 mg, 78%); m.p.: 98–101 °C; [α]^27^_D_ = 11.6 (c = 1.0, CHCl_3_); ^1^H NMR (400 MHz, DMSO-d6): δ 1.92, 1.98, 1.99, 2.12 (4s, 12H, CH_3_CO), 2.63 (s, 3H, CH_3_), 3.98 (dd, 1H, *J* = 6.6 Hz, *J* = 11.3 Hz, H-6a_gal_), 4.06 (dd, 1H, *J* = 5.9 Hz, *J* = 11.3 Hz, H-6b_gal_), 4.35 (m, 1H, H-5_gal_,), 5.07 (dd-t, 1H, *J* = 9.4 Hz, *J* = 9.4 Hz, H-2_gal_), 5.12 and 5.20 (qAB, 2H, *J* = 16.5 Hz, CH_2_CO), 5.26–5.44 (m, 5H, H-1_gal_, H-3_gal_, H-4_gal_, CH_2_O_chin_), 7.32–7.54 (m, 4H, H-3_chin_, H-5_chin_, H-6_chin_, H-7_chin_)8.21 (d, 1H, *J* = 7.9 Hz, H-4_chin_), 8.24 (s, 1H, H-5_triaz_), 9.33 (d, 1H, *J* = 9.5 Hz, CONH); ^13^C NMR (100 MHz, DMSO-d6): δ 20.23, 20.26, 20.31, 20.40, 24.84, 51.30, 61.54, 67.47, 68.16, 70.59, 71.35, 77.14, 79.06, 110.10, 119.72, 122.36, 125.54, 126.38, 127.24, 135.85, 139.11, 142.36, 153.27, 157.15, 166.02, 169.23, 169.29, 169.71, 169.77; HRMS (ESI-TOF): calcd for C_29_H_34_N_5_O_11_ ([M + H]^+^): *m*/*z* 628.2255; found: *m*/*z* 628.2253.

Glycoconjugate **81**: Starting from *N*-(β-D-glucopyranosyl)azidoacetamide **35** and 8-(2-propyn-1-yloxy)quinoline **3**, product was obtained as a brown solid (133.6 mg, 60%); m.p.: 160-163 °C; [α]^25^_D_ = 16.7 (c = 2.0, DMSO); ^1^H NMR (400 MHz, DMSO-d6): δ 3.04–3.15 (m, 3H, H-2_glu_, H-4_glu_, H-5_glu_), 3.21 (dd-t, 1H, *J* = 8.4 Hz, *J* = 8.6 Hz, H-3_glu_), 3.42 (m, 1H, H-6a_glu_), 3.63 (m, 1H, H-6b_glu_), 4.00–4.50 (m, 4H, OH), 4.71 (dd-t, 1H, *J* = 8.9 Hz, *J* = 9.0 Hz, H-1_glu_), 5.17 and 5.22 (qAB, 2H, *J* = 16.5 Hz, CH_2_CO), 5.57 (s, 2H, CH_2_O_chin_), 7.60–8.10 (m, 5H, H-3_chin_, H-4_chin_, H-5_chin_, H-6_chin_, H-7_chin_), 8.39 (s, 1H, H-5_triaz_), 8.96 (d, 1H, *J* = 7.4 Hz, H-2_chin_), 9.04 (d, 1H, *J* = 8.8 Hz, CONH); ^13^C NMR (100 MHz, DMSO-d6): δ 51.55, 60.72, 62.36, 69.74, 72.44, 77.18, 78.65, 79.63, 112.94, 120.36, 122.56, 126.79, 128.94, 132.86, 136.25, 136.88, 141.50, 146.25, 151.30, 165.67; HRMS (ESI-TOF): calcd for C_20_H_24_N_5_O_7_ ([M + H]^+^): *m*/*z* 446.1676; found: *m*/*z* 446.1679.

Glycoconjugate **82**: Starting from *N*-(β-D-galactopyranosyl)azidoacetamide **36** and 8-(2-propyn-1-yloxy)quinoline **3**, product was obtained as a brown solid (129.2 mg, 58%); m.p.: 170-173 °C; [α]^25^_D_ = 14.5 (c = 2.0, DMSO); ^1^H NMR (400 MHz, DMSO-d6): δ 3.34 (dd, 1H, *J* = 3.2 Hz, *J* = 9.4 Hz, H-3_gal_), 3.36–3.46 (m, 3H, H-2_gal_, H-5_gal_, H-6a_gal_), 3.50 (dd, 1H, *J* = 5.8 Hz, *J* = 10.5 Hz, H-6b_gal_), 3.69 (d, 1H, *J* = 3.0 Hz, H-4_gal_), 4.09–4.42 (m, 4H, OH), 4.69 (dd-t, 1H *J* = 8.9 Hz, *J* = 9.0 Hz, H-1_gal_), 5.18 (s, 2H, CH_2_CO), 5.60 (s, 2H, CH_2_O_chin_), 7.56–8.16 (m, 4H, H-3_chin_, H-5_chin_, H-6_chin_, H-7_chin_), 8.41 (s, 1H, H-5_triaz_), 8.85–9.20 (m, 3H, H-2_chin_, H-4_chin_, CONH); ^13^C NMR (100 MHz, DMSO-d6): δ 51.62, 60.31, 68.02, 69.66, 73.85, 76.75, 79.08, 80.07, 112.53, 120.38, 126.83, 126.96, 129.01, 130.10, 141.66, 142.90, 146.07, 150.24, 152.06, 165.58; HRMS (ESI-TOF): calcd for C_20_H_24_N_5_O_7_ ([M + H]^+^): *m*/*z* 446.1676; found: *m*/*z* 446.1670.

Glycoconjugate **83**: Starting from *N*-(β-D-glucopyranosyl)azidoacetamide **35** and 2-methyl-8-(2-propyn-1-yloxy)quinoline **4**, product was obtained as a brown solid (140.1 mg, 61%); m.p.: 135–138 °C; [α]^26^_D_ = −2.8 (c = 1.0, MeOH); ^1^H NMR (400 MHz, DMSO-d6): δ 2.92 (s, 3H, CH_3_), 3.04–3.24 (m, 3H, H-2_glu_, H-4_glu_, H-5_glu_), 3.39–3.47 (m, 2H, H-3_glu_, H-6a_glu_,), 3.62 (dd, 1H, *J* = 1.6 Hz, *J* = 12.1 Hz, H-6b_glu_), 4.69 (dd-t, 1H, *J* = 8.9 Hz, *J* = 9.0 Hz, H-1_glu_), 5.15 and 5.20 (qAB, 2H, *J* = 16.5 Hz, CH_2_CO), 5.60 (s, 2H, CH_2_O_chin_), 7.73–7.93 (m, 4H, H-3_chin_, H-5_chin_, H-6_chin_, H-7_chin_), 8.34 (s, 1H, H-5_triaz_), 8.89 (d, 1H, *J* = 8.3 Hz, H-4_chin_), 9.04 (d, 1H, *J* = 8.8 Hz, CONH); ^13^C NMR (100 MHz, DMSO-d6): δ 18.45, 51.52, 60.71, 62.31, 69.74, 72.42, 77.18, 78.65, 79.62, 113.85, 120.30, 124.31, 126.87, 127.70, 128.48, 141.31, 143.79, 148.61, 158.29, 163.67, 165.65; HRMS (ESI-TOF): calcd for C_21_H_26_N_5_O_7_ ([M + H]^+^): *m*/*z* 460.1832; found: *m*/*z* 460.1830.

Glycoconjugate **84**: Starting from *N*-(β-D-galactopyranosyl)azidoacetamide **36** and 2-methyl-8-(2-propyn-1-yloxy)quinoline **4**, product was obtained as a brown solid (179.2 mg, 78%); m.p.: 145–148 °C; [α]^24^_D_ = 5.0 (c = 1.0, MeOH); ^1^H NMR (400 MHz, DMSO-d6): δ 2.88 (s, 3H, CH_3_), 3.34 (dd, 1H, *J* = 3.2 Hz, *J* = 9.4 Hz, H-3_gal_), 3.37-3.54 (m, 4H, H-2_gal_, H-5_gal_, H-6a_gal_, H-6b_gal_), 3.69 (d, 1H, *J* = 3.0 Hz, H-4_gal_), 3.95–4.31 (m, 4H, OH), 4.67 (dd-t, 1H, *J* = 8.9 Hz, *J* = 9.0 Hz, H-1_gal_), 5.17 (s, 2H, CH_2_CO), 5.59 (s, 2H, CH_2_O_chin_), 7.62–7.98 (m, 4H, H-3_chin_, H-5_chin_, H-6_chin_, H-7_chin_), 8.34 (s, 1H, H-5_triaz_), 8.84 (bs, 1H, H-4_chin_), 9.01 (d, 1H, *J* = 9.0 Hz, CONH); ^13^C NMR (100 MHz, DMSO-d6): δ 18.45, 51.55, 60.30, 62.27, 68.01, 69.65, 73.85, 76.74, 80.06, 113.46, 120.27, 124.12, 126.87, 128.23, 131.53, 141.40, 142.91, 149.12, 158.22, 162.36, 165.59; HRMS (ESI-TOF): calcd for C_21_H_26_N_5_O_7_ ([M + H]^+^): *m*/*z* 460.1832; found: *m*/*z* 460.1831.

Glycoconjugate **85:** Starting from 2,3,4,6-tetra-*O*-acetyl-*N*-(β-D-glucopyranosyl)propiolamide **37** and 8-(2-azidoethoxy)quinoline **5**, product was obtained as a yellow solid (220.9 mg, 72%); m.p.: 89–94 °C; [α]^23^_D_ = −18.0 (c = 1.0, CHCl_3_); ^1^H NMR (400 MHz, DMSO-d6): δ 1.86, 1.94, 1.98, 2.00 (4s, 12H, CH_3_CO), 3.98 (dd, 1H, *J* = 3.9 Hz, *J* = 14.0 Hz, H-6a_glu_), 4.07–4.18 (m, 2H, H-5_glu_, H-6b_glu_), 4.66 (t, 2H, *J* = 5.1 Hz, CH_2_N), 4.90 (dd-t, 1H, *J* = 9.6 Hz, *J* = 9.6 Hz, H-4_glu_), 4.98 (t, 2H, *J* = 5.1 Hz, CH_2_O), 5.20 (dd-t, 1H, *J* = 9.3 Hz, *J* = 9.4 Hz, H-2_glu_), 5.38 (dd-t, 1H, *J* = 9.5 Hz, *J* = 9.5 Hz, H-1_glu_), 5.59 (dd-t, 1H, *J* = 9.3 Hz, *J* = 9.4 Hz, H-3_glu_), 7.25 (dd, 1H, *J* = 1.4 Hz, *J* = 7.5 Hz, H-7_chin_), 7.47–7.59 (m, 3H, H-3_chin_, H-5_chin_, H-6_chin_), 8.32 (dd, 1H, *J* = 1.7 Hz, *J* = 8.4 Hz, H-4_chin_), 8.87 (dd, 1H, *J* = 1.5 Hz, *J* = 4.1 Hz, H-2_chin_), 8.98 (s, 1H, H-5_triaz_), 9.20 (d, 1H, *J* = 9.5 Hz, NHCO); ^13^C NMR (100 MHz, DMSO-d6): δ 20.30, 20.31, 20.37, 20.50, 49.56, 61.85, 67.07, 67.85, 70.49, 72.04, 73.03, 76.79, 110.69, 120.60, 121.93, 126.69, 128.20, 129.08, 135.86, 139.71, 141.83, 149.25, 153.60, 160.03, 169.02, 169.32, 169.54, 169.97; HRMS (ESI-TOF): calcd for C_28_H_32_N_5_O_11_ ([M + H]^+^): *m*/*z* 614.2098; found: *m*/*z* 614.2097.

Glycoconjugate **86:** Starting from 2,3,4,6-tetra-*O*-acetyl-*N*-(β-D-galactopyranosyl)propiolamide **38** and 8-(2-azidoethoxy)quinoline **5**, product was obtained as a white solid (285.3 mg, 93%); m.p.: 187–189 °C; [α]^24^_D_ = −6.8 (c = 1.0, CHCl_3_); ^1^H NMR (400 MHz, CDCl_3_): δ 1.96, 2.00, 2.01, 2.17 (4s, 12H, CH_3_CO), 4.03–4.17 (m, 3H, H-5_gal_, H-6a_gal_, H-6b_gal_), 4.63 (t, 2H, *J* = 4.9 Hz, CH_2_N), 5.02 (t, 2H, *J* = 4.9 Hz, CH_2_O), 5.15 (dd, 1H, *J* = 3.3 Hz, *J* = 10.1 Hz, H-3_gal_), 5.30 (dd-t, 1H, *J* = 9.3 Hz, *J* = 10.1 Hz, H-2_gal_), 5.41 (dd-t, 1H, *J* = 9.3 Hz, *J* = 9.5 Hz, H-1_gal_), 5.45 (dd, 1H, *J* = 0.6 Hz, *J* = 3.3 Hz, H-4_gal_), 7.04 (dd, 1H, *J* = 1.7 Hz, *J* = 7.2 Hz, H-7_chin_), 7.40–7.50 (m, 3H, H-3_chin_, H-5_chin_, H-6_chin_), 7.83 (d, 1H, *J* = 9.5 Hz, NHCO); 8.14 (dd, 1H, *J* = 1.7 Hz, *J* = 8.3 Hz, H-4_chin_), 8.95 (s, 1H, H-5_triaz_), 9.00 (dd, 1H, *J* = 1.7 Hz, *J* = 4.2 Hz. H-2_chin_); ^13^C NMR (100 MHz, CDCl_3_): δ 20.57, 20.63, 20.64, 20.67, 50.14, 61.27, 67.24, 67.53, 68.08, 71.17, 72.32, 78.15, 110.28, 121.34, 122.02, 126.42, 128.22, 129.64, 135.98, 140.36, 142.35, 149.81, 153.67, 160.32, 169.90, 170.15, 170.35, 170.46; HRMS (ESI-TOF): calcd for C_28_H_32_N_5_O_11_ ([M + H]^+^): *m*/*z* 614.2098; found: *m*/*z* 614.2095.

Glycoconjugate **87:** Starting from 2,3,4,6-tetra-*O*-acetyl-*N*-(β-D-glucopyranosyl)propiolamide **37** and 2-methyl-8-(2-azidoethoxy)quinoline **8**, product was obtained as a yellow solid (207.1 mg, 66%); m.p.: 167–172 °C; [α]^24^_D_ = −14.8 (c = 1.0, CHCl_3_); ^1^H NMR (400 MHz, DMSO-d6): δ 1.85, 1.93, 1.98, 2.00 (4s, 12H, CH_3_CO), 2.72 (s, 3H, CH_3_), 3.98 (dd, 1H, *J* = 4.5 Hz, *J* = 14.3 Hz, H-6a_glu_), 4.07–4.18 (m, 2H, H-5_glu_, H-6b_glu_), 4.54–4.67 (m, 2H, CH_2_N), 4.90 (dd-t, 1H, *J* = 9.4 Hz, *J* = 9.5 Hz, H-4_glu_), 4.98 (t, 2H, *J* = 5.0 Hz, CH_2_O), 5.21 (dd-t, 1H, *J* = 9.3 Hz, *J* = 9.4 Hz, H-2_glu_),5.38 (dd-t, 1H, *J* = 9.5 Hz, *J* = 9.5 Hz, H-1_glu_), 5.60 (dd-t, 1H, *J* = 9.3 Hz, *J* = 9.4 Hz, H-3_glu_), 7.21 (dd, 1H, *J* = 1.2 Hz, *J* = 7.7 Hz, H-7_chin_), 7.38–7.46 (m, 2H, H-3_chin_, H-6_chin_), 7.50 (dd, 1H, *J* = 1.1 Hz, *J* = 8.2 Hz, H-5_chin_), 8.20 (d, 1H, *J* = 8.5, H-4_chin_), 9.17 (d, 1H, *J* = 9.5 Hz, NHCO), 9.20 (s, 1H, H-5_triaz_); ^13^C NMR (100 MHz, DMSO-d6): δ 20.29, 20.31, 20.37, 20.50, 25.00, 49.59, 61.86, 67.17, 67.85, 70.52, 72.01, 73.01, 76.75, 111.06, 120.46, 122.58, 125.63, 127.30, 128.62, 136.02, 139.28, 141.93, 153.06, 157.80, 160.08, 169.01, 169.32, 169.53, 169.97; HRMS (ESI-TOF): calcd for C_29_H_33_N_5_O_11_ ([M + H]^+^): *m*/*z* 628.2255; found: *m*/*z* 628.2252.

Glycoconjugate **88:** Starting from 2,3,4,6-tetra-*O*-acetyl-*N*-(β-D-galactopyranosyl)propiolamide **38** and 2-methyl-8-(2-azidoethoxy)quinoline **8**, product was obtained as a white solid (257.3 mg, 82%); m.p.: 107–109 °C; [α]^25^_D_ = −7.2 (c = 1.0, CHCl_3_); ^1^H NMR (400 MHz, CDCl_3_): δ 1.95, 2.00, 2.02, 2.17 (4s, 12H, CH_3_CO), 2.88 (s, 3H, CH_3_), 4.03–4.17 (m, 3H, H-5_gal_, H-6a_gal_, H-6b_gal_), 4.59 (t, 2H, *J* = 4.9 Hz, CH_2_N), 5.00 (t, 2H, *J* = 4.8 Hz, CH_2_O), 5.15 (dd, 1H, *J* = 3.4 Hz, *J* = 10.2 Hz, H-3_gal_), 5.30 (dd-t, 1H, *J* = 9.4 Hz, *J* = 10.2 Hz, H-2_gal_), 5.42 (dd-t, 1H, *J* = 9.4 Hz, *J* = 9.5 Hz, H-1_gal_), 5.45 (dd, 1H, *J* = 0.7 Hz, *J* = 3.4 Hz, H-4_gal_), 7.02 (dd, 1H, *J* = 1.3 Hz, *J* = 7.5 Hz, H-7_chin_), 7.32–7.39 (m, 2H, H-3_chin_, H-6_chin_), 7.42 (dd, 1H, *J* = 1.3 Hz, *J* = 8.2 Hz, H-5_chin_), 7.84 (d, 1H, *J* = 9.6 Hz, NHCO), 8.02 (d, 1H, *J* = 8.4 Hz, H-4_chin_), 9.26 (s, 1H, H-5_triaz_); ^13^C NMR (100 MHz, CDCl_3_): δ 20.57, 20.61, 20.64, 20.66, 25.62, 50.18, 61.23, 67.23, 67.45, 68.08, 71.18, 72.25, 78.11, 110.54, 121.14, 122.90, 125.38, 127.79, 128.65, 136.04, 139.94, 142.40, 153.15, 159.03, 160.36, 169.90, 170.15, 170.34, 170.41; HRMS (ESI-TOF): calcd for C_29_H_34_N_5_O_11_ ([M + H]^+^): *m*/*z* 628.2255; found: *m*/*z* 628.2254.

Glycoconjugate **89:** Starting from *N*-(β-D-glucopyranosyl)propiolamide **39** and 8-(2-azidoethoxy)quinoline **5**, product was obtained as a white solid (133.6 mg, 60%); m.p.: 169–171 °C; [α]^28^_D_ = 5.0 (c = 1.0, DMSO); ^1^H NMR (400 MHz, DMSO-d6): δ 3.05-3.11 (m, 1H, H-5_glu_), 3.13–3.18 (m, 2H, H-2_glu_, H-4_glu_), 3.33–3.37 (m, 1H, H-3_glu_), 3.38–3.44 (m, 1H, H-6a_glu_), 3.60–3.67 (m, 1H, H-6b_glu_), 4.48 (t, 1H, *J* = 5.9 Hz, 6-OH), 4.66 (t, 2H, *J* = 4.6 Hz, CH_2_N), 4.83–4.93 (m, 3H, H-1_glu_, OH), 4.95–5.01 (m, 3H, CH_2_O, OH), 7.25 (d, 1H, *J* = 7.5 Hz, H-7_chin_), 7.47–7.62 (m, 3H, H-3_chin_, H-5_chin_, H-6_chin_), 8.32 (d, 1H, *J* = 10.4 Hz, H-4_chin_), 8.68 (d, 1H, *J* = 9.1 Hz, NHCO), 8.88 (bs, 1H, H-2_chin_), 8.94 (s, 1H, H-5_triaz_); ^13^C NMR (100 MHz, DMSO-d6): δ 49.52, 60.97, 67.11, 69.97, 71.83, 77.44, 78.70, 79.51, 110.64, 120.60, 121.96, 126.69, 127.68, 129.08, 135.84, 139.71, 142.53, 149.29, 153.63, 160.08; HRMS (ESI-TOF): calcd for C_20_H_24_N_5_O_7_ ([M + H]^+^): *m*/*z* 446.1676; found: *m*/*z* 446.1678.

Glycoconjugate **90:** Starting from *N*-(β-D-galactopyranosyl)propiolamide **40** and 8-(2-azidoethoxy)quinoline **5**, product was obtained as a white solid (189.3 mg, 85%); m.p.: 162–164 °C; [α]^25^_D_ = 24.8 (c = 1.0, MeOH); ^1^H NMR (400 MHz, DMSO-d6): δ 3.34-3.54 (m, 4H, H-2_gal_, H-3_gal_, H-4_gal_ H-5_gal_), 3.57–3.65 (m, 1H, H-6a_gal_), 3.66–3.72 (m, 1H, H-6b_gal_), 4.29 (d, 1H, *J* = 4.9 Hz, OH), 4.56 (t, 1H, *J* = 5.4, 6-OH), 4.66 (t, 2H, *J* = 5.1 Hz, CH_2_N), 4.74 (d, 1H, *J* = 5.5 Hz, OH), 4.79 (d, 1H, *J* = 5.5 Hz, OH), 4.87 (dd-t, 1H, *J* = 9.0 Hz, *J* = 9.1 Hz, H-1_gal_), 4.98 (t, 2H, *J* = 5.0 Hz, CH_2_O), 7.26 (dd, 1H, *J* = 1.4 Hz, *J* = 7.5 Hz, H-7_chin_), 7.47–7.59 (m, 3H, H-3_chin_, H-5_chin_, H-6_chin_), 8.32 (dd, 1H, *J* = 1.5 Hz, *J* = 8.2 Hz, H-4_chin_), 8.48 (d, 1H, *J* = 9.1 Hz, NHCO), 8.88 (dd, 1H, *J* = 1.7 Hz, *J* = 4.1 Hz, H-2_chin_), 8.95 (s, 1H, H-5_triaz_); ^13^C NMR (100 MHz, DMSO-d6): δ 49.52, 60.42, 67.13, 68.37, 69.33, 74.03, 76.82, 79.93, 110.69, 120.60, 121.93, 126.68, 127.61, 129.07, 135.84, 139.72, 142.47, 149.29, 153.62, 159.97; HRMS (ESI-TOF): calcd for C_20_H_24_N_5_O_7_ ([M + H]^+^): *m*/*z* 446.1676; found: *m*/*z* 446.1675.

Glycoconjugate **91:** Starting from *N*-(β-D-glucopyranosyl)propiolamide **39** and 2-methyl-8-(2-azidoethoxy)quinoline **8**, product was obtained as a white solid (144.7 mg, 63%); m.p.: 159–161 °C; [α]^25^_D_ = −3.6 (c = 1.0, MeOH); ^1^H NMR (400 MHz, DMSO-d6): δ 2.72 (s, 3H, CH_3_), 3.03–3.12 (m, 1H, H-5_glu_), 3.13–3.27 (m, 2H, H-2_glu_, H-4_glu_), 3.34–3.46 (m, 2H, H-3_glu_, H-6a_glu_), 3.60–3.68 (m, 1H, H-6b_glu_), 4.47 (t, 1H, *J* = 5.9 Hz, 6-OH), 4.55–4.67 (m, 2H, CH_2_N), 4.86–4.93 (m, 3H, H-1_glu_, OH), 4.94–5.02 (m, 3H, CH_2_O, OH), 7.21 (dd, 1H, *J* = 1.2 Hz, *J* = 7.7 Hz, H-7_chin_), 7.39–7.47 (m, 2H, H-3_chin_, H-6_chin_), 7.50 (dd, 1H, *J* = 1.1 Hz, *J* = 8.2 Hz, H-5_chin_), 8.20 (d, 1H, *J* = 8.4 Hz, H-4_chin_), 8.63 (d, 1H, *J* = 9.1 Hz, NHCO), 9.15 (s, 1H, H-5_triaz_); ^13^C NMR (100 MHz, DMSO-d6): δ 25.07, 49.57, 60.99, 67.27, 69.97, 71.85, 77.42, 78.68, 79.47, 111.02, 120.47, 122.62, 125.64, 127.31, 128.11, 136.02, 139.28, 142.63, 153.11, 157.85, 160.12; HRMS (ESI-TOF): calcd for C_20_H_24_N_5_O_7_ ([M + H]^+^): *m*/*z* 460.1832; found: *m*/*z* 460.1830.

Glycoconjugate **92:** Starting from *N*-(β-D-galactopyranosyl)propiolamide **40** and 2-methyl-8-(2-azidoethoxy)quinoline **8**, product was obtained as a white solid (186.1 mg, 81%); m.p.: 148–151 °C; [α]^28^_D_ = 23.0 (c = 1.0, DMSO); ^1^H NMR (400 MHz, DMSO-d6): δ 2.72 (s, 3H, CH_3_), 3.34–3.55 (m, 4H, H-2_gal_, H-3_gal_, H-4_gal_ H-5_gal_), 3.57–3.66 (m, 1H, H-6a_gal_), 3.66–3.72 (m, 1H, H-6b_gal_), 4.28 (d, 1H, *J* = 5.0 Hz, OH), 4.52–4.67 (m, 3H, CH_2_N, 6-OH), 4.73 (d, 1H, *J* = 5.5 Hz, OH), 4.79 (d, 1H, *J* = 5.5 Hz, OH), 4.88 (dd-t, 1H, *J* = 9.0 Hz, *J* = 9.1 Hz, H-1_gal_), 4.98 (t, 2H, *J* = 5.0 Hz, CH_2_O), 7.21 (dd, 1H, *J* = 1.2 Hz, *J* = 7.7 Hz, H-7_chin_), 7.39–7.46 (m, 2H, H-3_chin_, H-6_chin_), 7.50 (dd, 1H, *J* = 1.1 Hz, *J* = 8.2 Hz, H-5_chin_), 8.20 (d, 1H, *J* = 8.4 Hz, H-4_chin_), 8.43 (d, 1H, *J* = 9.1 Hz, NHCO), 9.16 (s, 1H, H-5_triaz_); ^13^C NMR (100 MHz, DMSO-d6): δ 25.04, 49.56, 60.40, 67.28, 68.37, 69.36, 74.02, 76.78, 79.89, 111.08, 120.47, 122.59, 125.62, 127.30, 128.04, 136.00, 139.29, 142.57, 153.11, 157.83, 160.00; HRMS (ESI-TOF): calcd for C_21_H_26_N_5_O_7_ ([M + H]^+^): *m*/*z* 460.1832; found: *m*/*z* 460.1836.

Glycoconjugate **93:** Starting from 2,3,4,6-tetra-*O*-acetyl-*N*-(β-D-glucopyranosyl)propiolamide **37** and 8-(3-azidopropoxy)quinoline **6**, product was obtained as a yellow solid (210.2 mg, 67%); m.p.: 178–180 °C; [α]^24^_D_ = −18.6 (c = 1.0, CHCl_3_); ^1^H NMR (400 MHz, DMSO-d6): δ 1.89, 1.94, 1.99, 2.00 (4s, 12H, CH_3_CO), 2.40–2.50 (m, 2H, CH_2_), 4.00 (dd, 1H, *J* = 4.4 Hz, *J* = 14.2 Hz, H-6a_glu_), 4.06–4.18 (m, 2H, H-5_glu_, H-6b_glu_), 4.21 (t, 2H, *J* = 6.0 Hz, CH_2_N), 4.72 (t, 2H, *J* = 6.8 Hz, CH_2_O), 4.91 (dd-t, 1H, *J* = 9.5 Hz, *J* = 9.5 Hz, H-4_glu_), 5.21 (dd-t, 1H, *J* = 9.3 Hz, *J* = 9.4 Hz, H-2_glu_), 5.39 (dd-t, 1H, *J* = 9.5 Hz, *J* = 9.5 Hz, H-1_glu_), 5.61 (dd-t, 1H, *J* = 9.3 Hz, *J* = 9.4 Hz, H-3_glu_), 7.20 (dd, 1H, *J* = 1.8 Hz, *J* = 7.1 Hz, H-7_chin_), 7.46–7.59 (m, 3H, H-3_chin_, H-5_chin_, H-6_chin_), 8.32 (dd, 1H, *J* = 1.5 Hz, *J* = 8.1 Hz, H-4_chin_), 8.90 (dd, 1H, *J* = 1.7 Hz, *J* = 4.0 Hz, H-2_chin_), 8.90 (s, 1H, H-5_triaz_), 9.17 (d, 1H, *J* = 9.5 Hz, NHCO); ^13^C NMR (100 MHz, DMSO-d6): δ 20.32, 20.33, 20.38, 20.52, 29.36, 47.24, 61.85, 65.52, 67.84, 70.52, 72.04, 73.02, 76.80, 109.92, 119.97, 121.84, 126.77, 127.67, 129.03, 135.79, 139.77, 141.83, 149.07, 154.18, 160.10, 169.04, 169.32, 169.54, 169.98; HRMS (ESI-TOF): calcd for C_29_H_34_N_5_O_11_ ([M + H]^+^): *m*/*z* 628.2255; found: *m*/*z* 628.2252.

Glycoconjugate **94:** Starting from 2,3,4,6-tetra-*O*-acetyl-*N*-(β-D-galactopyranosyl)propiolamide **38** and 8-(3-azidopropoxy)quinoline **6**, product was obtained as a white solid (263.6 mg, 84%); m.p.: 98–100 °C; [α]^25^_D_ = −5.6 (c = 1.0, CHCl_3_); ^1^H NMR (400 MHz, CDCl_3_): δ 1.99, 2.00, 2.03, 2.17 (4s, 12H, CH_3_CO), 2.62 (p, 2H, *J* = 6.2 Hz, CH_2_), 4.04–4.18 (m, 3H, H-5_gal_, H-6a_gal_, H-6b_gal_), 4.24 (t, 2H, *J* = 5.8 Hz, CH_2_N), 4.81 (t, 2H, *J* = 6.6 Hz, CH_2_O), 5.17 (dd, 1H, *J* = 3.3 Hz, *J* = 10.2 Hz, H-3_gal_), 5.31 (dd-t, 1H, *J* = 9.4 Hz, *J* = 10.2 Hz, H-2_gal_), 5.42 (dd-t, 1H, *J* = 9.4 Hz, *J* = 9.5 Hz, H-1_gal_), 5.46 (dd, 1H, *J* = 0.6 Hz, *J* = 3.3 Hz, H-4_gal_), 7.06 (dd, 1H, *J* = 1.7 Hz, *J* = 7.2 Hz, H-7_chin_), 7.41–7.50 (m, 3H, H-3_chin_, H-5_chin_, H-6_chin_), 7.85 (d, 1H, *J* = 9.5 Hz, NHCO), 8.16 (dd, 1H, *J* = 1.7 Hz, *J* = 8.3 Hz, H-4_chin_), 8.52 (s, 1H, H-5_triaz_), 9.01 (dd, 1H, *J* = 1.7 Hz, *J* = 4.2 Hz. H-2_chin_); ^13^C NMR (100 MHz, CDCl_3_): δ 20.53, 20.57, 20.64, 20.68, 29.67, 47.73, 61.28, 65.36, 67.24, 68.11, 71.16, 72.33, 78.16, 109.94, 120.67, 121.82, 126.62, 127.18, 129.61, 136.09, 140.39, 142.04, 149.55, 154.23, 160.36, 169.90, 170.15, 170.36, 170.45; HRMS (ESI-TOF): calcd for C_29_H_34_N_5_O_11_ ([M + H]^+^): *m*/*z* 628.2255; found: *m*/*z* 628.2253.

Glycoconjugate **95:** Starting from 2,3,4,6-tetra-*O*-acetyl-*N*-(β-D-glucopyranosyl)propiolamide **37** and 2-methyl-8-(3-azidopropoxy)quinoline **9**, product was obtained as a beige solid (195.7 mg, 61%); m.p.: 197–198 °C; [α]^24^_D_ = −18.0 (c = 1.0, CHCl_3_); ^1^H NMR (400 MHz, DMSO-d6): δ 1.88, 1.94, 1.99, 2.00 (4s, 12H, CH_3_CO), 2.41–2.49 (m, 2H, CH_2_), 2.67 (s, 3H, CH_3_), 3.96–4.02 (m, 1H, H-6a_glu_), 4.05–4.17 (m, 2H, H-5_glu_, H-6b_glu_), 4.20 (t, 2H, *J* = 6.2 Hz, CH_2_N), 4.71 (t, 2H, *J* = 6.9 Hz, CH_2_O), 4.91 (dd-t, 1H, *J* = 9.5 Hz, *J* = 9.5 Hz, H-4_glu_), 5.20 (dd-t, 1H, *J* = 9.3 Hz, *J* = 9.4 Hz, H-2_glu_), 5.38 (dd-t, 1H, *J* = 9.5 Hz, *J* = 9.5 Hz, H-1_glu_), 5.59 (dd-t, 1H, *J* = 9.3 Hz, *J* = 9.4 Hz, H-3_glu_), 7.17 (dd, 1H, *J* = 1.3 Hz, *J* = 7.6 Hz, H-7_chin_), 7.36–7.44 (m, 2H, H-3_chin_, H-6_chin_), 7.47 (dd, 1H, *J* = 1.2 Hz, *J* = 8.2 Hz, H-5_chin_), 8.19 (d, 1H, *J* = 8.4 Hz, H-4_chin_), 8.83 (s, 1H, H-5_triaz_), 9.17 (d, 1H, *J* = 9.5 Hz, NHCO); ^13^C NMR (100 MHz, DMSO-d6): δ 20.31, 20.37, 20.51, 20.74, 25.03, 29.25, 47.14, 61.84, 65.56, 67.83, 70.51, 72.02, 73.01, 76.78, 110.54, 119.91, 122.44, 125.68, 127.32, 127.58, 135.97, 139.34, 141.77, 153.64, 157.35, 160.06, 169.02, 169.31, 169.53, 169.97; HRMS (ESI-TOF): calcd for C_30_H_36_N_5_O_11_ ([M + H]^+^): *m*/*z* 642.2411; found: *m*/*z* 642.2408.

Glycoconjugate **96:** Starting from 2,3,4,6-tetra-*O*-acetyl-*N*-(β-D-galactopyranosyl)propiolamide **38** and 2-methyl-8-(3-azidopropoxy)quinoline **9**, product was obtained as a white solid (202.1 mg, 63%); m.p.: 121–123 °C; [α]^25^_D_ = −4.6 (c = 1.0, CHCl_3_); ^1^H NMR (400 MHz, CDCl_3_): δ 1.98, 2.00, 2.03, 2.17 (4s, 12H, CH_3_CO), 2.60 (p, 2H, *J* = 6.4 Hz, CH_2_), 2.80 (s, 3H, CH_3_), 4.04–4.17 (m, 3H, H-5_gal_, H-6a_gal_, H-6b_gal_), 4.24 (t, 2H, *J* = 5.9 Hz, CH_2_N), 4.82 (t, 2H, *J* = 6.7 Hz, CH_2_O), 5.16 (dd, 1H, *J* = 3.4 Hz, *J* = 10.2 Hz, H-3_gal_), 5.30 (dd-t, 1H, *J* = 9.4 Hz, *J* = 10.1 Hz, H-2_gal_), 5.40 (dd-t, 1H, *J* = 9.4 Hz, *J* = 9.5 Hz, H-1_gal_), 5.46 (dd, 1H, *J* = 0.6 Hz, *J* = 3.3 Hz, H-4_gal_), 7.04 (dd, 1H, *J* = 1.8 Hz, *J* = 7.1 Hz, H-7_chin_), 7.31–7.42 (m, 3H, H-3_chin_, H-5_chin_, H-6_chin_), 7.84 (d, 1H, *J* = 9.5 Hz, NHCO), 8.03 (d, 1H, *J* = 8.4 Hz, H-4_chin_), 8.38 (s, 1H, H-5_triaz_); ^13^C NMR (100 MHz, CDCl_3_): δ 20.57, 20.63, 20.64, 20.68, 25.65, 29.58, 47.58, 61.27, 65.24, 67.23, 68.10, 71.15, 72.32, 78.14, 110.56, 120.60, 122.68, 125.58, 127.06, 127.85, 136.19, 140.02, 141.93, 153.59, 158.44, 160.30, 169.90, 170.14, 170.36, 170.44; HRMS (ESI-TOF): calcd for C_30_H_36_N_5_O_11_ ([M + H]^+^): *m*/*z* 642.2411; found: *m*/*z* 642.2409.

Glycoconjugate **97:** Starting from *N*-(β-D-glucopyranosyl)propiolamide **39** and 8-(3-azidopropoxy)quinoline **6**, product was obtained as a white solid (158.5 mg, 69%); m.p.: 185–189 °C; [α]^28^_D_ = 6.0 (c = 1.0, DMSO); ^1^H NMR (400 MHz, DMSO-d6): δ 2.46 (p, 2H, *J* = 6.5 Hz, CH_2_), 3.06–3.11 (m, 1H, H-5_glu_), 3.15-3.19 (m, 1H, H-2_glu_), 3.20–3.25 (m, 1H, H-4_glu_), 3.33–3.37 (m, 1H, H-3_glu_), 3.39–3.45 (m, 1H, H-6a_glu_), 3.63–3.68 (m, 1H, H-6b_glu_), 4.19 (t, 2H, *J* = 6.0 Hz, OH), 4.49 (t, 1H, *J* = 5.9 Hz, 6-OH), 4.72 (t, 2H, *J* = 6.3 Hz, CH_2_N), 4.84–4.94 (m, 3H, H-1_glu_, CH_2_O), 5.00 (d, 1H, *J* = 4.6 Hz, OH), 7.20 (dd, 1H, *J* = 1.5 Hz, *J* = 7.4 Hz, H-7_chin_), 7.47-7.62 (m, 3H, H-3_chin_, H-5_chin_, H-6_chin_), 8.32 (dd, 1H, *J* = 1.5 Hz, *J* = 8.0 Hz, H-4_chin_), 8.66 (d, 1H, *J* = 9.1 Hz, NHCO), 8.86 (s, 1H, H-5_triaz_), 8.91 (dd, 1H, *J* = 1.6 Hz, *J* = 4.1 Hz, H-2_chin_); ^13^C NMR (100 MHz, DMSO-d6): δ 26.42, 44.13, 58.00, 62.40, 66.98, 68.88, 74.44, 75.69, 76.48, 106.91, 116.97, 118.88, 123.77, 124.17, 132.78, 136.66, 139.53, 142.49, 145.37, 146.13, 157.11; HRMS (ESI-TOF): calcd for C_21_H_26_N_5_O_7_ ([M + H]^+^): *m*/*z* 460.1832; found: *m*/*z* 460.1832.

Glycoconjugate **98:** Starting from *N*-(β-D-galactopyranosyl)propiolamide **40** and 8-(3-azidopropoxy)quinoline **6**, product was obtained as a white solid (170.0 mg, 74%); m.p.: 129–132 °C; [α]^28^_D_ = 22.0 (c = 1.0, DMSO); ^1^H NMR (400 MHz, DMSO-d6): δ 2.41–2.49 (m, 2H, CH_2_), 3.35–3.56 (m, 4H, H-2_gal_, H-3_gal_, H-4_gal_ H-5_gal_), 3.58–3.66 (m, 1H, H-6a_gal_), 3.68–3.75 (m, 1H, H-6b_gal_), 4.20 (t, 2H, *J* = 6.0 Hz, OH), 4.30 (d, 1H, *J* = 4.9 Hz, OH), 4.58 (t, 1H, *J* = 5.4 Hz, 6-OH), 4.68–4.77 (m, 3H, CH_2_N, CH_2_O), 4.81 (d, 1H, *J* = 5.5 Hz, CH_2_O), 4.89 (dd-t, 1H, *J* = 9.0 Hz, *J* = 9.1 Hz, H-1_gal_), 7.20 (dd, 1H, *J* = 1.8 Hz, *J* = 7.1 Hz, H-7_chin_), 7.46–7.60 (m, 3H, H-3_chin_, H-5_chin_, H-6_chin_), 8.32 (dd, 1H, *J* = 1.6 Hz, *J* = 8.3 Hz, H-4_chin_), 8.47 (d, 1H, *J* = 9.1 Hz, NHCO), 8.87 (s, 1H, H-5_triaz_), 8.91 (bs, 1H, H-2_chin_); ^13^C NMR (100 MHz, DMSO-d6): δ 29.41, 47.14, 60.43, 65.43, 68.38, 69.37, 74.04, 76.80, 79.92, 109.90, 119.97, 121.86, 126.77, 127.09, 129.03, 135.78, 139.77, 142.48, 149.12, 154.18, 160.01; HRMS (ESI-TOF): calcd for C_21_H_26_N_5_O_7_ ([M + H]^+^): *m*/*z* 460.1832; found: *m*/*z* 460.1835.

Glycoconjugate **99:** Starting from *N*-(β-D-glucopyranosyl)propiolamide **39** and 2-methyl-8-(3-azidopropoxy)quinoline **9**, product was obtained as a brown solid (137.3 mg, 58%); m.p.: 167–169 °C; [α]^27^_D_ = −7.0 (c = 1.0, MeOH); ^1^H NMR (400 MHz, DMSO-d6): δ 2.46 (p, 2H, *J* = 6.5 Hz, CH_2_), 2.68 (s, 3H, CH_3_), 3.05–3.12 (m, 1H, H-5_glu_), 3.14–3.19 (m, 1H, H-2_glu_), 3.20–3.25 (m, 1H, H-4_glu_), 3.34–3.38 (m, 1H, H-3_glu_), 3.39–3.46 (m, 1H, H-6a_glu_), 3.61–3.68 (m, 1H, H-6b_glu_), 4.18 (t, 2H, *J* = 6.1 Hz, OH), 4.49 (m, 1H, 6-OH), 4.72 (t, 2H, *J* = 6.9 Hz, CH_2_N), 4.87–4.94 (m, 3H, H-1_glu_, CH_2_O), 5.00 (bs, 1H, OH), 7.17 (d, 1H, *J* = 7.5 Hz, H-7_chin_), 7.38–7.50 (m, 3H, H-3_chin_, H-5_chin_, H-6_chin_), 8.19 (d, 1H, *J* = 8.4 Hz, H-4_chin_), 8.66 (d, 1H, *J* = 9.1 Hz, NHCO), 8.80 (s, 1H, H-5_triaz_); ^13^C NMR (100 MHz, DMSO-d6): δ 25.06, 29.32, 47.02, 60.99, 65.43, 69.98, 71.87, 77.44, 78.69, 79.49, 110.51, 119.92, 122.48, 125.71, 127.10, 127.33, 136.01, 142.48, 146.29, 153.63, 157.39, 160.08; HRMS (ESI-TOF): calcd for C_22_H_28_N_5_O_7_ ([M + H]^+^): *m*/*z* 474.1989; found: *m*/*z* 474.1987.

Glycoconjugate **100:** Starting from *N*-(β-D-galactopyranosyl)propiolamide **40** and 2-methyl-8-(3-azidopropoxy)quinoline **9**, product was obtained as a brown solid (175.2 mg, 74%); m.p.: 77–80 °C; [α]^26^_D_ = 8.0 (c = 1.0, MeOH); ^1^H NMR (400 MHz, DMSO-d6): δ 2.41–2.49 (m, 2H, CH_2_), 2.67 (s, 3H, CH_3_), 3.35–3.54 (m, 4H, H-2_gal_, H-3_gal_, H-4_gal_ H-5_gal_), 3.57–3.64 (m, 1H, H-6a_gal_), 3.66–3.72 (m, 1H, H-6b_gal_), 4.15–4.22 (m, 2H, OH), 4.30 (d, 1H, *J* = 4.9 Hz, OH), 4.37–4.44 (m, 1H, OH), 4.52–4.60 (m, 2H, CH_2_N), 4.66–4.78 (m, 3H, CH_2_N, CH_2_O), 4.81 (d, 1H, *J* = 5.5 Hz, CH_2_O), 4.87 (dd-t, 1H, *J* = 9.0 Hz, *J* = 9.1 Hz, H-1_gal_), 7.17 (dd, 1H, *J* = 1.2 Hz, *J* = 7.6 Hz, H-7_chin_), 7.37–7.50 (m, 3H, H-3_chin_, H-5_chin_, H-6_chin_), 8.19 (d, 1H, *J* = 8.4 Hz, H-4_chin_), 8.46 (d, 1H, *J* = 9.1 Hz, NHCO), 8.80 (s, 1H, H-5_triaz_); ^13^C NMR (100 MHz, DMSO-d6): δ 24.96, 29.23, 46.95, 60.33, 65.36, 68.28, 69.28, 73.94, 76.71, 79.83, 110.43, 119.82, 122.38, 125.62, 126.92, 127.24, 135.92, 139.22, 142.34, 153.54, 157.29, 159.88; HRMS (ESI-TOF): calcd for C_22_H_28_N_5_O_7_ ([M + H]^+^): *m*/*z* 474.1989; found: *m*/*z* 474.1987.

Glycoconjugate **101:** Starting from 2,3,4,6-tetra-*O*-acetyl-*N*-(β-D-glucopyranosyl)-*O*-propargyl carbamate **41** and 8-(3-azidopropoxy)quinoline **6**, product was obtained as a white solid (309.1 mg, 94%); m.p.: 60–65 °C; [α]^25^_D_ = −2.6 (c = 1.0, CHCl_3_); ^1^H NMR (400 MHz, CDCl_3_): δ 2.00, 2.01, 2.02, 2.07 (4s, 12H, CH_3_CO), 2.62 (p, 2H, *J* = 6.4 Hz, CH_2_), 3.73–3.83 (m, 1H, H-5_glu_), 4.03–4.13 (m, 1H, H-6a_glu_), 4.24 (t, 2H, *J* = 5.9 Hz, CH_2_N), 4.26–4.34 (m, 1H, H-6b_glu_), 4.74 (t, 2H, *J* = 6.7 Hz, CH_2_O), 4.90 (dd-t, 1H, *J* = 9.5 Hz, *J* = 9.5 Hz, H-4_glu_), 5.00 (dd-t, 1H, *J* = 9.5 Hz, *J* = 9.6 Hz, H-3_glu_), 5.06 (dd-t, 1H, *J* = 9.7 Hz, *J* = 9.7 Hz, H-2_glu_), 5.16 and 5.20 (qAB, 2H, *J* = 12.7 Hz, CH_2_OCO), 5.59 (d, 1H, *J* = 9.5 Hz, H-1_glu_), 7.04 (dd, 1H, *J* = 2.6 Hz, *J* = 6.3 Hz, H-7_chin_), 7.40–7.50 (m, 3H, H-3_chin_, H-5_chin_, H-6_chin_), 7.78 (s, 1H, H-5_triaz_), 8.16 (dd, 1H, *J* = 1.7 Hz, *J* = 8.3 Hz, H-4_chin_), 8.96 (dd, 1H, *J* = 1.7 Hz, *J* = 4.2 Hz, H-2_chin_); ^13^C NMR (100 MHz, CDCl_3_): δ 20.56, 20.57, 20.63, 20.72, 29.66, 47.32, 58.81, 61.58, 65.26, 68.08, 70.11, 72.80, 73.36, 80.82, 109.52, 120.38, 121.74, 124.61, 126.68, 129.58, 136.07, 140.35, 142.38, 149.42, 154.23, 155.23, 169.48, 169.90, 170.55, 170.59; HRMS (ESI-TOF): calcd for C_30_H_36_N_5_O_12_ ([M + H]^+^): *m*/*z* 658.2360; found: *m*/*z* 658.2360.

Glycoconjugate **102:** Starting from 2,3,4,6-tetra-*O*-acetyl-*N*-(β-D-galactopyranosyl)-*O*-propargyl carbamate **42** and 8-(3-azidopropoxy)quinoline **6**, product was obtained as a white solid (295.9 mg, 90%); m.p.: 66–69 °C; [α]^25^_D_ = −7.0 (c = 1.0, CHCl_3_); ^1^H NMR (400 MHz, CDCl_3_): δ 1.98, 2.02, 2.03, 2.13 (4s, 12H, CH_3_CO), 2.62 (p, 2H, *J* = 6.4 Hz, CH_2_), 3.96–4.03 (m, 1H, H-6a_gal_), 4.04–4.16 (m, 2H, H-5_gal_, H-6a_gal_), 4.25 (t, 2H, *J* = 5.9 Hz, CH_2_N), 4.74 (t, 2H, *J* = 6.7 Hz, CH_2_O), 4.97 (dd-t, 1H, *J* = 8.3 Hz, *J* = 8.8 Hz, H-2_gal_), 5.03–5.13 (m, 2H, H-3_gal_, H-4_gal_), 5.16 and 5.20 (qAB, 2H, *J* = 12.8 Hz, CH_2_OCO), 5.58 (d, 1H, *J* = 9.5 Hz, H-1_gal_), 7.05 (dd, 1H, *J* = 2.5 Hz, *J* = 6.4 Hz, H-7_chin_), 7.41–7.49 (m, 3H, H-3_chin_, H-5_chin_, H-6_chin_), 7.78 (s, 1H, H-5_triaz_), 8.16 (dd, 1H, *J* = 1.7 Hz, *J* = 8.3 Hz, H-4_chin_), 8.96 (dd, 1H, *J* = 1.7 Hz, *J* = 4.2 Hz, H-2_chin_); ^13^C NMR (100 MHz, CDCl_3_): δ 20.52, 20.60, 20.67, 20.71, 29.67, 47.33, 58.79, 61.07, 65.29, 67.10, 67.87, 70.92, 72.08, 81.16, 109.56, 120.39, 121.75, 124.57, 126.68, 129.59, 136.07, 140.37, 142.48, 149.43, 154.25, 155.25, 169.79, 170.04, 170.33, 170.83; HRMS (ESI-TOF): calcd for C_30_H_36_N_5_O_12_ ([M + H]^+^): *m*/*z* 658.2360; found: *m*/*z* 658.2360.

Glycoconjugate **103:** Starting from 2,3,4,6-tetra-*O*-acetyl-*N*-(β-D-glucopyranosyl)-*O*-propargyl carbamate **41** and 2-methyl-8-(3-azidopropoxy)quinoline **9**, product was obtained as a white solid (228.4 mg, 68%); m.p.: 56–59 °C; [α]^26^_D_ = −3.0 (c = 1.0, CHCl_3_); ^1^H NMR (400 MHz, CDCl_3_): δ 2.01, 2.01, 2.03, 2.07 (4s, 12H, CH_3_CO), 2.59 (p, 2H, *J* = 6.4 Hz, CH_2_), 2.79 (s, 3H, CH_3_), 3.73–3.83 (m, 1H, H-5_glu_), 4.01–4.13 (m, 1H, H-6a_glul_), 4.23 (t, 2H, *J* = 5.9 Hz, CH_2_N), 4.26–4.34 (m, 1H, H-6b_glu_), 4.75 (t, 2H, *J* = 6.7 Hz, CH_2_O), 4.89 (dd-t, 1H, *J* = 9.4 Hz, *J* = 9.5 Hz, H-4_glu_), 4.99 (dd-t, 1H, *J* = 9.5 Hz, *J* = 9.7 Hz, H-3_glu_), 5.06 (dd-t, 1H, *J* = 9.7 Hz, *J* = 9.8 Hz, H-2_glu_), 5.13–5.23 (m, 2H, *J* = 12.7 Hz, CH_2_OCO), 5.53 (d, 1H, *J* = 9.5 Hz, H-1_glu_), 7.03 (dd, 1H, *J* = 2.3 Hz, *J* = 6.6 Hz, H-7_chin_), 7.30–7.43 (m, 3H, H-3_chin_, H-5_chin_, H-6_chin_), 7.83 (s, 1H, H-5_triaz_), 8.04 (dd, 1H, *J* = 1.7 Hz, *J* = 8.4 Hz, H-4_chin_); ^13^C NMR (100 MHz, CDCl_3_): δ 20.58, 20.59, 20.66, 20.74, 25.69, 29.62, 47.25, 58.79, 61.54, 65.32, 68.01, 70.03, 72.75, 73.30, 80.78, 110.04, 120.31, 122.64, 124.70, 125.66, 127.78, 136.21, 139.88, 142.28, 153.62, 155.20, 158.27, 169.50, 169.93, 170.58, 170.63; HRMS (ESI-TOF): calcd for C_31_H_38_N_5_O_12_ ([M + H]^+^): *m*/*z* 672.2517; found: *m*/*z* 672.2516.

Glycoconjugate **104:** Starting from 2,3,4,6-tetra-*O*-acetyl-*N*-(β-D-galactopyranosyl)-*O*-propargyl carbamate **42** and 2-methyl-8-(3-azidopropoxy)quinoline **9**, product was obtained as a white solid (214.9 mg, 64%); m.p.: 62–65 °C; [α]^26^_D_ = 6.8 (c = 1.0, CHCl_3_); ^1^H NMR (400 MHz, CDCl_3_): δ 1.98, 2.02, 2.03, 2.14 (4s, 12H, CH_3_CO), 2.59 (p, 2H, *J* = 6.4 Hz, CH_2_), 2.79 (s, 3H, CH_3_), 3.96–4.03 (m, 1H, H-6a_gal_), 4.04–4.16 (m, 2H, H-5_gal_, H-6a_gal_), 4.23 (t, 2H, *J* = 5.9 Hz, CH_2_N), 4.76 (t, 2H, *J* = 6.7 Hz, CH_2_O), 4.96 (dd-t, 1H, *J* = 8.3 Hz, *J* = 8.8 Hz, H-2_gal_), 5.03–5.13 (m, 2H, H-3_gal_, H-4_gal_), 5.14–5.23 (m, 2H, CH_2_OCO), 5.53 (d, 1H, *J* = 9.3 Hz, H-1_gal_), 7.03 (dd, 1H, *J* = 2.5 Hz, *J* = 6.5 Hz, H-7_chin_), 7.31–7.43 (m, 3H, H-3_chin_, H-5_chin_, H-6_chin_), 7.83 (s, 1H, H-5_triaz_), 8.04 (d, 1H, *J* = 8.4 Hz, H-4_chin_); ^13^C NMR (100 MHz, CDCl_3_): δ 20.53, 20.61, 20.67, 20.72, 25.69, 29.68, 47.28, 58.80, 61.07, 65.45, 67.10, 67.85, 70.93, 72.08, 81.15, 110.24, 120.37, 122.63, 124.62, 125.67, 127.82, 136.19, 139.99, 142.36, 153.71, 155.23, 158.28, 169.80, 170.05, 170.34, 170.83; HRMS (ESI-TOF): calcd for C_31_H_38_N_5_O_12_ ([M + H]^+^): *m*/*z* 672.2517; found: *m*/*z* 672.2518.

Glycoconjugate **105:** Starting from *N*-(β-D-glucopyranosyl)-*O*-propargyl carbamate **43** and 8-(3-azidopropoxy)quinoline **6**, product was obtained as a white solid (203.1 mg, 83%); m.p.: 73–75 °C; [α]^27^_D_ = 0.2 (c = 1.0, MeOH); ^1^H NMR (400 MHz, DMSO-d6): δ 2.42 (p, 2H, *J* = 6.4 Hz, CH_2_), 2.96–3.11 (m, 3H, H-2_glu_, H-4_glu_, H-5_glu_), 3.15–3.20 (m, 1H, H-3_glu_), 3.36–3.44 (m, 1H, H-6a_glu_), 3.58–3.68 (m, 1H, H-6b_glu_), 4.09 (d, 1H, *J* = 5.1 Hz, OH), 4.20 (t, 2H, *J* = 6.0 Hz, CH_2_N), 4.45–4.54 (m, 2H, OH), 4.65 (t, 2H, *J* = 6.9 Hz, CH_2_O), 4.86 (m, 2H, CH_2_OCO), 4.95 (m, 1H, H-1_glu_), 5.07 (s, 1H, OH), 7.20 (dd, 1H, *J* = 2.2 Hz, *J* = 6.8 Hz, H-7_chin_), 7.47–7.59 (m, 3H, H-3_chin_, H-5_chin_, H-6_chin_), 7.89 (d, 1H, *J* = 9.2 Hz, NHCO), 8.31 (s, 1H, H-5_triaz_), 8.33 (dd, 1H, *J* = 1.7 Hz, *J* = 8.3 Hz, H-4_chin_), 8.89 (dd, 1H, *J* = 1.7 Hz, *J* = 4.1 Hz, H-2_chin_); ^13^C NMR (100 MHz, DMSO-d6): δ 29.64, 46.70, 57.07, 60.95, 65.43, 69.92, 72.02, 77.56, 78.38, 82.44, 109.85, 119.94, 121.89, 124.98, 126.82, 129.06, 135.84, 139.77, 142.42, 149.09, 154.19, 155.71; HRMS (ESI-TOF): calcd for C_22_H_28_N_5_O_8_ ([M + H]^+^): *m*/*z* 490.1938; found: *m*/*z* 490.1937.

Glycoconjugate **106:** Starting from *N*-(β-D-galactopyranosyl)-*O*-propargyl carbamate **44** and 8-(3-azidopropoxy)quinoline **6**, product was obtained as a white solid (200.7 mg, 82%); m.p.: 119–121 °C; [α]^28^_D_ = 19.0 (c = 1.0, DMSO); ^1^H NMR (400 MHz, DMSO-d6): δ 2.42 (p, 2H, *J* = 6.5 Hz, CH_2_), 3.33–3.53 (m, 4H, H-2_gal_, H-3_gal_, H-4_gal_ H-5_gal_), 3.63-3.69 (m, 1H, H-6a_gal_), 4.08 (dd, 1H, *J* = 5.3 Hz, *J* = 10.5 Hz, H-6b_gal_), 4.20 (t, 2H, *J* = 6.1 Hz, CH_2_N), 4.32 (d, 1H, *J* = 3.8 Hz, OH), 4.46 (dd-t, 1H, *J* = 9.0 Hz, *J* = 9.0 Hz, H-1_gal_), 4.55 (t, 1H, *J* = 5.6 Hz, OH), 4.61–4.72 (m, 4H, CH_2_O, CH_2_OCO), 5.06 (s, 2H, OH), 7.20 (dd, 1H, *J* = 2.2 Hz, *J* = 6.8 Hz, H-7_chin_), 7.47–7.59 (m, 3H, H-3_chin_, H-5_chin_, H-6_chin_), 7.84 (d, 1H, *J* = 9.2 Hz, NHCO), 8.29 (s, 1H, H-5_triaz_), 8.32 (dd, 1H, *J* = 1.7 Hz, *J* = 8.2 Hz, H-4_chin_), 8.89 (dd, 1H, *J* = 1.7 Hz, *J* = 4.1 Hz, H-2_chin_); ^13^C NMR (100 MHz, DMSO-d6): δ 29.62, 46.68, 57.03, 60.47, 65.44, 68.19, 69.27, 74.21, 76.53, 82.92, 109.86, 119.93, 121.86, 124.92, 126.80, 129.04, 135.80, 139.77, 142.47, 149.06, 154.18, 155.76; HRMS (ESI-TOF): calcd for C_22_H_28_N_5_O_8_ ([M + H]^+^): *m*/*z* 490.1938; found: *m*/*z* 490.1935.

Glycoconjugate **107:** Starting from *N*-(β-D-glucopyranosyl)-*O*-propargyl carbamate **43** and 2-methyl-8-(3-azidopropoxy)quinoline **9**, product was obtained as a white solid (201.4 mg, 80%); m.p.: 57–60 °C; [α]^27^_D_ = −0.2 (c = 1.0, MeOH); ^1^H NMR (400 MHz, DMSO-d6): δ 2.41 (p, 2H, *J* = 6.5 Hz, CH_2_), 2.67 (s, 3H, CH_3_), 2.96–3.11 (m, 3H, H-2_glu_, H-4_glu_, H-5_glu_), 3.12–3.20 (m, 1H, H-3_glu_), 3.36–3.44 (m, 1H, H-6a_glu_), 3.59–3.68 (m, 1H, H-6b_glu_), 4.09 (d, 1H, *J* = 5.3 Hz, OH), 4.19 (t, 2H, *J* = 6.2 Hz, CH_2_N), 4.44–4.53 (m, 2H, OH), 4.61–4.69 (m, 2H, CH_2_O), 4.86 (m, 2H, CH_2_OCO), 4.94 (d, 1H, *J* = 4.7 Hz, H-1_glu_), 5.06 (s, 1H, OH), 7.16 (dd, 1H, *J* = 1.1 Hz, *J* = 7.5 Hz, H-7_chin_), 7.38–7.50 (m, 3H, H-3_chin_, H-5_chin_, H-6_chin_), 7.88 (d, 1H, *J* = 9.2 Hz, NHCO), 8.02 (d, 1H, *J* = 9.2 Hz, H-2_chin_), 8.19 (d, 1H, *J* = 8.4 Hz, H-4_chin_), 8.30 (s, 1H, H-5_triaz_); ^13^C NMR (100 MHz, DMSO-d6): δ 25.08, 29.57, 46.68, 57.06, 60.94, 65.58, 69.92, 72.02, 77.54, 78.37, 82.43, 110.36, 119.86, 122.48, 124.94, 125.75, 127.34, 136.01, 139.32, 142.39, 153.68, 155.70, 157.35; HRMS (ESI-TOF): calcd for C_23_H_30_N_5_O_8_ ([M + H]^+^): *m*/*z* 504.2094; found: *m*/*z* 504.2090.

Glycoconjugate **108:** Starting from *N*-(β-D-galactopyranosyl)-*O*-propargyl carbamate **44** and 2-methyl-8-(3-azidopropoxy)quinoline **9**, product was obtained as a white solid (196.4 mg, 78%); m.p.: 94–98 °C; [α]^26^_D_ = 3.0 (c = 1.0, H_2_O); ^1^H NMR (400 MHz, DMSO-d6): δ 2.41 (p, 2H, *J* = 6.6 Hz, CH_2_), 2.67 (s, 3H, CH_3_), 3.33–3.53 (m, 4H, H-2_gal_, H-3_gal_, H-4_gal_ H-5_gal_), 3.63–3.69 (m, 1H, H-6a_gal_), 4.08 (dd, 1H, *J* = 5.3 Hz, *J* = 10.5 Hz, H-6b_gal_), 4.19 (t, 2H, *J* = 6.2 Hz, CH_2_N), 4.32 (d, 1H, *J* = 3.8 Hz, OH), 4.46 (m, 1H, H-1_gal_), 4.55 (t, 1H, *J* = 5.6 Hz, OH), 4.61–4.73 (m, 4H, CH_2_O, CH_2_OCO), 5.06 (s, 2H, OH), 7.16 (dd, 1H, *J* = 1.3 Hz, *J* = 7.6 Hz, H-7_chin_), 7.38–7.49 (m, 3H, H-3_chin_, H-5_chin_, H-6_chin_), 7.83 (d, 1H, *J* = 9.2 Hz, NHCO), 8.19 (d, 1H, *J* = 8.4 Hz, H-4_chin_), 8.29 (s, 1H, H-5_triaz_), ^13^C NMR (100 MHz, DMSO-d6): δ 25.07, 29.56, 46.66, 57.03, 60.47, 65.60, 68.18, 69.27, 74.21, 76.52, 82.92, 110.39, 119.86, 122.46, 124.88, 125.73, 127.33, 135.98, 139.33, 142.45, 153.68, 155.76, 157.32; HRMS (ESI-TOF): calcd for C_23_H_30_N_5_O_8_ ([M + H]^+^): *m*/*z* 504.2094; found: *m*/*z* 504.2097.

### 3.3. Biological Assays

#### 3.3.1. Cell Lines

The culture media were purchased from EuroClone, HyClone and Pan Biotech. Fetal bovine serum (FBS) was delivered by Eurx, Poland and Antibiotic Antimycotic Solution (100×) by Sigma-Aldrich, Germany. The human colon adenocarcinoma cell line HCT 116 was obtained from American Type Culture Collection (ATCC, Manassas, VA, USA). The human cell line MCF-7 was obtained from collections at the Maria Sklodowska-Curie Memorial Cancer Center and Institute of Oncology, branch in Gliwice, Poland, as kindly gift from Monika Pietrowska and prof. Wiesława Widłak. The Normal Human Dermal Fibroblasts-Neonatal, NHDF-Neo were purchased from LONZA (Cat. No. CC-2509, NHDF-Neo, Dermal Fibroblasts, Neonatal, Lonza, Poland). The culture media consisted of RPMI 1640 or DMEM+F12 medium, supplemented with 10% fetal bovine serum and standard antibiotics.

#### 3.3.2. MTT Assay

A lifespan of the cells was assessed with an MTT (3-[4–dimethylthiazol-2-yl]-2,5-diphenyltetrazolium bromide) test (Sigma-Aldrich). The cells at concentration 1 × 10^4^ (HCT 116, NHDF-Neo) or 5 × 10^3^ (MCF-7) per well were seeded into 96-well plates. The cell cultures were incubated for 24 h at 37 °C in a humidified atmosphere of 5% CO_2_. After this time, the culture medium was removed, replaced with solution of the tested compounds in medium and incubated for further 24 h or 72 h. After that, medium was removed and the MTT solution (50 µL, 0.5 mg/mL in PBS) was added. After 3 h of incubation, the MTT solution was removed and the acquired formazan was dissolved in DMSO. Finally, the absorbance at the 570 nm wavelength was measured spectrophotometrically with the plate reader. The experiment was conducted in three independent iterations with four technical repetitions. Tests were conducted at concentrations tested compounds range from 0.01 mM to 0.8 mM solutions. For the most active compounds IC_50_ values were calculated using CalcuSyn.

#### 3.3.3. Bovine Milk β-1,4-Galactosyltransferase I Assay

Bovine milk β-1,4-Galactosyltransferase I was purchased from Sigma-Aldrich. The β-1,4-GalT activity was assayed using UDP-Gal, a natural β-1,4-GalT glycosyl donor type substrate, and (6-esculetinyl) β-D-glucopyranoside (esculine) as glycosyl fluorescent acceptor. The reaction mixtures contained reagents in the following final concentrations: 50 mM citrate buffer (pH 5.4), 100 mM MnCl_2_, 20 mg/mL BSA, 2 mM esculine, 0.4 mM UDP-Gal and 10 μL MeOH or methanolic solution of potential inhibitors **49–52**, **57–60**, **65–68**, **73–76**, **81–84** at 0.8 mM concentration. Assays were performed in a total volume of 200 µL. The enzymatic reactions were started by the addition of 8 mU β-1,4-GalT and incubated at 30 °C for 60 min. After that, the reaction mixture was diluted with water to a volume of 500 μL and then was placed in a thermoblock set at 90 °C for 3 min. After protein denaturation, the solutions were centrifuged at 10 °C for 30 min at 10 000 rpm. The supernatant was filtered using syringe filters (M.E. Cellulose filter, Teknokroma^®^, 0.2 µm × 13 mm). The filtrate was injected into RP-HPLC system. The percentage of inhibition was evaluated from the fluorescence intensity of the peaks referring to product (6-esculetinyl) 4′-*O*-β-D-galactopyranosyl-β-D-glucopyranoside). For the most active compounds, IC_50_ values were determined by the same procedure using the reaction mixtures containing inhibitor in the concentrations of 0.1, 0.2, 0.4, and 0.8 mM and calculated using CalcuSyn software.

### 3.4. Metal Complexing Studies

Absorption measurements were carried out at room temperature, using a UV/VIS/NIR spectrophotometer JASCO V-570 (Tokyo, Japan) with the software Spectra Manager Program. Absorption spectra of selected compounds, in the presence of copper (II) sulfate pentahydrate in HPLC grade methanol, were recorded in spectra range of 200–450 nm. The stoichiometry of complexes were determined using Job′s method, by titrating the particular compound methanol solution with increasing concentrations of copper salt. The graph was plotted as the difference in absorbance of the absorption band at 265 nm, ΔA = A_x_ − A_0_ as a function of molar fraction of glycoconjugate. The maximum in the plot corresponds to the stoichiometry of complex formed.

Electrospray mass spectrometry (ESI-MS^n^) was performed using a Thermo Scientific LCQ Fleet ion trap spectrometer (Thermo Fisher Scientific Inc., CA, USA). All analyses were performed in methanol. The solutions were introduced into the ESI source by continuous infusion at 4 µL·min^−1^ using the syringe pump. The ESI source was set to an operating voltage of 4 kV and the capillary heater was set to 160 °C. Nitrogen was used as the nebulizing gas and helium was used as the collision and damping gas in the ion trap.

## 4. Conclusions

In summary, a wide range of new glycoconjugates derivatives of 8-HQ, in which the sugar part was connected to the quinoline derivative via linker containing a 1,2,3-triazole fragment was obtained and characterized. The effect of modifying the structure of the mentioned linker on the biological activity in vitro exhibited by glycoconjugates was investigated. These modifications concerned the extension of the alkyl chain between 1,2,3-triazole ring and quinoline or sugar moiety, as well as the introduction of an additional amide or carbamate group into the linker structure. The obtained glycoconjugates contained the sugar unit with a binding of the β-configuration at the anomeric center since such orientation is preferred for binding to the GLUT transporters [84]. The new glycoconjugates structures have been designed in such a way to take advantage of some unique features of cancer cells, such as the high concentration of Cu^2+^ ions and overexpression of some proteins, such as β-1,4-galactosyltransferase or GLUT transporters to improving their selectivity and minimizing side effects of their application on healthy tissues.

Glycoconjugates were tested for inhibition of the proliferation of selected cancer cell lines and inhibition of β-1,4-Gal activity, in which overexpression is associated with cancer progression. All glycoconjugates in protected form have a cytotoxic effect on cancer cells in the tested concentration range. Tested glycoconjugates were more active on the HCT 116 cell line than parent quinolines. Therefore, it can be assumed that the presence of a sugar unit and an additional metal ion binding motif improves the cytotoxic activity of glycoconjugates. MCF-7 cell line appears to be more sensitive to quinoline derivatives. Most of the glycoconjugates showed lower activity than their parent compounds against the breast cancer cell line, however the addition of a sugar unit allowed to improve their selectivity (as can be seen from the example of compounds **63**, **85** and **86**). Glycoconjugates containing an additional amide bond in the linker structure proved to be more active relative to the cell lines tested. Among them, derivatives whose structure was based on the 8-HQ fragment showed better cytotoxicity compared to derivatives with 2Me8HQ unit. In turn, the type of sugar unit did not significantly affect the activity of synthesized glycoconjugates. It can be concluded that the presence of amide groups improves the activity of glycoconjugates, probably through the improvement of metal ions chelation. Interestingly, glycoconjugates with an amide moiety in the structure did not affect the inhibition of β-1,4-Gal activity. On the other hand, it was noted, that extension the alkyl chain between triazole and quinoline part increases the ability of glycoconjugates to inhibit enzyme activity, which is probably associated with increased “flexibility” of the molecule. However, it cannot be ruled out any mechanism of action at this step.

Although the IC_50_ values determined do not predestine the use of the obtained compounds as potential drugs, it is possible to notice some regularities in their structure that improve the activity of the obtained molecules. In the course of the research, it was found that the activity of the obtained glycoconjugates depends on the presence of the sugar moiety, the presence of protecting groups in the sugar unit as well as the type and length of the linker connecting the sugar parts and quinoline moiety. Sugar improves the solubility, bioavailability, and selectivity of potential drugs, whereas the heteroaromatic fragment improves the activity of glycoconjugate, probably due to the ability to chelate metals ions present in many types of cancer cells. The study of metal complexing properties confirmed that the obtained glycoconjugates are capable of chelating copper ions. Moreover, the addition of the 1,2,3-triazole fragment improved the ability to form such complexes. The determined stoichiometry of the glycoconjugate complex indicates that the tested compounds chelate copper ions in a 1:1 molar ratio, while 8-HQ forms chelates in a 2:1 ratio.

One of the important strategies for improving the therapeutic efficacy of anticancer drugs is the synthesis of new derivatives with a modified structure. The obtained results encourage further structural optimization of quinoline glycoconjugates. A huge library of compounds will allow the identification of those specific structural elements that are responsible for demonstrating biological activity.

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
