# Peer review of "8-Hydroxyquinoline Glycoconjugates: Modifications in the Linker Structure and Their Effect on the Cytotoxicity of the Obtained Compounds"

_molecules, 2019, doi:10.3390/molecules24224181_

Round 1
Reviewer 1 Report
This paper describes a clear and logical synthesis and biological evaluation of a huge number of glycoconjugates. The synthetic protocol is routine, but that is also never straightforward, and the authors did a thorough investigation to overcome some initial difficulties. The biological testing is also thorough. Unfortunately the results were not outstandingly encouraging, but nevertheless the "negative" results make a useful contribution to future workers in this area. The paper is written in an expansive style, with a long and full introduction, and a detailed discussion of the results. While it could be abbreviated, I do not suggest this, because many of the readers will be more biological than chemical, so the work will be easier to follow in this form. The very large number of new compounds have been well characterised. The manuscript would benefit from some light English editing. I recommend acceptance for publication.
Author Response
This paper describes a clear and logical synthesis and biological evaluation of a huge number of glycoconjugates. The synthetic protocol is routine, but that is also never straightforward, and the authors did a thorough investigation to overcome some initial difficulties. The biological testing is also thorough. Unfortunately the results were not outstandingly encouraging, but nevertheless the "negative" results make a useful contribution to future workers in this area. The paper is written in an expansive style, with a long and full introduction, and a detailed discussion of the results. While it could be abbreviated, I do not suggest this, because many of the readers will be more biological than chemical, so the work will be easier to follow in this form. The very large number of new compounds have been well characterised. The manuscript would benefit from some light English editing. I recommend acceptance for publication.
Response: Thank you very much for your review. In fact, we tried to write the publication so that it could be understood by both chemists and biologists. We have made efforts in order to improve English language and style.
Reviewer 2 Report
This paper describes the synthesis of derivatives of 8-hydroxyquinoline as glycoconjugates with protected and deprotected derivatives of D-glucose and D-galactose connected by different linker containing a 1,2,3-triazole ring. Moreover, authors tested synthesized glycoconjugates for their potential anticancer activity in vitro.
The research activities illustrated by the authors are well described and clear from the point of view of the synthesis and description of the anticancer tests.
The variability of the synthesized structure of quinoline glycoconjugates is broad and consistent.
The aim of work, as stated by authors, is to extend the library of quinoline glycoconjugate combinations, and from this point of view it does not present great elements of innovation.
Before publication the authors must correct the "supplementary" in fact the structures of the compounds 6-9 in NMR spectra are wrong (Fig.s from S7 to S15). They should also explain what p means in the description of NMR spectra; it seems strange that many methylene quintets are described with two J coupling constants and indicated with a letter p (pag. 17 r. 540, r. 548; pag. 18 r. 563, r. 571, pag. 21 r. 743; pag. 22 r. 758 … and so on)
Author Response
This paper describes the synthesis of derivatives of 8-hydroxyquinoline as glycoconjugates with protected and deprotected derivatives of D-glucose and D-galactose connected by different linker containing a 1,2,3-triazole ring. Moreover, authors tested synthesized glycoconjugates for their potential anticancer activity in vitro.
The research activities illustrated by the authors are well described and clear from the point of view of the synthesis and description of the anticancer tests.
The variability of the synthesized structure of quinoline glycoconjugates is broad and consistent.
The aim of work, as stated by authors, is to extend the library of quinoline glycoconjugate combinations, and from this point of view it does not present great elements of innovation.
Before publication the authors must correct the "supplementary" in fact the structures of the compounds 6-9 in NMR spectra are wrong (Fig.s from S7 to S15). They should also explain what p means in the description of NMR spectra; it seems strange that many methylene quintets are described with two J coupling constants and indicated with a letter p (pag. 17 r. 540, r. 548; pag. 18 r. 563, r. 571, pag. 21 r. 743; pag. 22 r. 758 … and so on).
Response: Thank you very much for your careful reading of the manuscript and supporting info. We have made revisions according to your comments and suggestions, as described below.
We agree with the Reviewer’s comment, that the structures of the compounds 6-9 in NMR spectra included in the supporting information have been confused, which has been corrected; “p” in the description of NMR spectra is an abbreviation for the word “pentet”. This term is used interchangeably in the literature with the term “quintet”. Because this was not entirely clear, now in General information we have added a description of the abbreviations used to describe the signal's multiplicity on NMR spectra; methylene quintets were described with two coupling constants because, in reality, this signal should be a triplet of triplets (tt) resulting from coupling with two groups of protons with a very similar value of coupling constants. Therefore, this signal is sometimes described as “tt~p” (triplet of triplets resembling a pentet). Unfortunately, coupling constants derived from methylene groups were incorrectly noted, which has been corrected.